# *ECHO*: Toward Contextual Seq2Seq Paradigms in Large EEG Models

**Chenyu Liu**[1]*, **Yuqiu Deng**[1]*, **Tianyu Liu**[2], **Jinan Zhou**[3], **Xinliang Zhou**[1], **Ziyu Jia**[4,5]†, **Yi Ding**[1]†

[1]College of Computing and Data Science, Nanyang Technological University, Singapore
[2]School of Mechanical Engineering, Xi'an Jiaotong University, Xi'an, China
[3]Nutanix, CA, USA
[4]Institute of Automation, Chinese Academy of Sciences, Beijing, China
[5]Key Laboratory of Social Computing and Cognitive Intelligence (Dalian University of Technology),
 Ministry of Education
`chenyu003@ntu.edu.sg, genuineyukeo@gmail.com, liuty@stu.xjtu.edu.cn,`
`jinan.zhou@nutanix.com, xinliang001@ntu.edu.sg, jia.ziyu@outlook.com,`
`ding.yi@ntu.edu.sg`

## Abstract

Electroencephalography (EEG), with its broad range of applications, necessitates models that can generalize effectively across various tasks and datasets. Large EEG Models (LEMs) address this by pretraining encoder-centric architectures on large-scale unlabeled data to extract universal representations. While effective, these models lack decoders of comparable capacity, limiting the full utilization of the learned features. To address this issue, we introduce ECHO, a novel decoder-centric LEM paradigm that reformulates EEG modeling as sequence-to-sequence learning. ECHO captures layered relationships among signals, labels, and tasks within sequence space, while incorporating discrete support samples to construct contextual cues. This design equips ECHO with in-context learning, enabling dynamic adaptation to heterogeneous tasks without parameter updates. Extensive experiments across multiple datasets demonstrate that, even with basic model components, ECHO consistently outperforms state-of-the-art single-task LEMs in multi-task settings, showing superior generalization and adaptability.

## 1 Introduction

Electroencephalography (EEG), owing to its portability and cost-effectiveness, has become the most widely used neural recording modality. Leveraging these advantages, EEG has been broadly applied to emotion recognition (Liu et al., 2024b), motor imagery (Ding et al., 2025), and diverse cognitive paradigms, which in turn demands models capable of maintaining generalization across heterogeneous tasks. Following the trend of large-scale models, researchers have proposed a series of Large EEG Models (LEMs) that place a pretrained encoder architecture at the core (Zhou et al., 2025a). These models are typically trained on large collections of unlabeled EEG data with self-supervised objectives such as masked reconstruction (Jiang et al., 2024b) or contrastive prediction (Wang et al., 2024a), thereby producing latent representations with strong generalization capacity that have demonstrated remarkable transferability across tasks.

Although these pretrained EEG encoders have been shown to learn high-quality representations, a major limitation remains: ***they lack decoders of comparable capacity to transform such representations into usable predictions.*** As illustrated in Figure 1 **a** (top), their decoding paradigm typically relies on a lightweight classifier (orders of magnitude smaller than the pretrained encoder) combined with additional fine-tuning to adapt the representations for downstream tasks. In other words, the model's success on downstream tasks largely depends on whether the encoder can "bend" its representations during fine-tuning to accommodate a limited-capacity decoder.

---

*Equal contribution
†Corresponding author

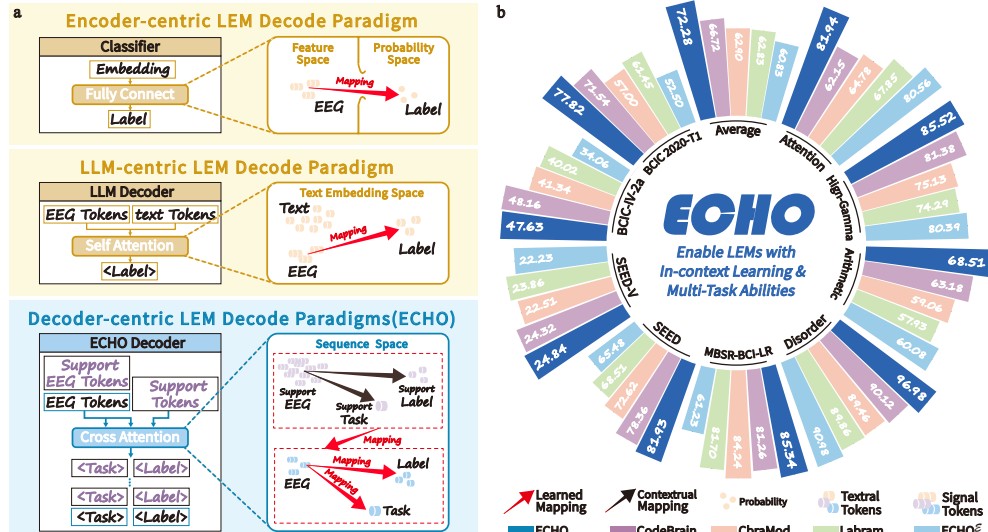

Figure 1: **a**, Top: Encoder-centric LEMs learn a direct mapping from EEG features to labels. Middle: LLM-centric LEMs follow the same scheme but shift it into the text embedding space. Bottom: ECHO extends such mapping by modeling various mappings within the sequence space. **b**, Performance comparison. ECHO$^{\mathcal{E}}$ indicates ECHO that do not adopt a decoder-centric paradigm. Detailed experimental settings and results are provided in Section 4 and Table 2.

Such adaptation to small-scale downstream data is inherently high-risk. On the one hand, the encoder may sacrifice its pretrained general knowledge to meet the fine-tuning demands of the decoder, leading to knowledge forgetting and degraded generalization (Guan et al., 2025). On the other hand, when the decoder itself is insufficient to reliably extract task-discriminative information, reliance on limited labeled data amplifies training uncertainty and makes the model sensitive to noise patterns (Hao et al., 2025). Consequently, **current paradigm remains constrained by decoder bottlenecks, preventing LEMs from realizing their generalization potential.**

While some recent studies have explored incorporating large language models (LLMs) as decoders, this paradigm remains fundamentally constrained: *it does not move beyond the "EEG-to-label" mapping, but merely shifts it into the text embedding space.* As illustrated in Figure 1 **a** (middle), this approach requires the LEM encoder and the LLM decoder to align by projecting EEG tokens and labels into a shared text embedding space, where the mapping is performed under the constraints of textual prompts (e.g., restricting the label types (Jiang et al., 2024a)).

The inductive biases of language models cannot be reliably transferred to time series EEG task (Tan et al., 2024). This stems from fundamental structural differences between EEG and language or vision. EEG relies on the precise localization of critical temporal dynamics, which are inherently misaligned with the static semantic patterns of text or images (Jing et al., 2024; Queen et al., 2023). As a result, projecting EEG directly into the text embedding space often drives models to exploit superficial correlations, such as mapping noise patterns to semantic labels (Wang & Ma, 2025), while diluting or even corrupting task-relevant information in the shared space (Almudévar et al., 2025). Ultimately, **text fails to serve as a genuine semantic bridge across modalities and instead functions merely as a surrogate label space, leaving LEMs without the reasoning and in-context learning (ICL) capabilities expected from LLMs.**

To overcome the limitations of existing paradigms, we propose a decoder-centric sequence-to-sequence (Seq2Seq) approach that enables LEMs to jointly model multi-task EEG representations while leveraging discrete samples as contextual support. As illustrated in Figure 1 **a** (bottom), the input is structured as a sequence comprising target EEG samples together with supporting EEG instances and their associated task and label tokens. The model performs next-token prediction, establishing associations between support samples and the target based on their mapped relationships. This process guides the generation of an output sequence that integrates both label and task tokens, thereby achieving multi-task learning within a unified framework. In summary, we refer to this new paradigm as **ECHO**: a decoder-centric framework and sequence-based learning method that preserves task-discriminative capacity while equipping LEMs with ICL.

To validate the effectiveness of our approach, we adopt off-the-shelf model components to avoid conflating our paradigm with architectural enhancements. We conduct extensive experiments across multiple EEG datasets, showing that ECHO consistently outperforms the latest single-task LEM baselines, even in multi-task settings (see Figure 1 **b**). ECHO can infer both the target task and its specific paradigms (e.g., identifying motor imagery and distinguishing its variants) without explicit prompts. Moreover, ECHO demonstrates ICL ability, adapting to new tasks and environments under the guidance of support samples. These results highlight the critical role of ECHO in advancing cross-task generalization and complex scenario modeling, while also providing insights for unlocking the full potential of existing LEMs.

## 2 PRELIMINARY

### 2.1 MULTI-TASK LEARNING FOR LEMS

Given heterogeneous EEG datasets, each dataset is represented as $\mathcal{D} = (\boldsymbol{X}, \boldsymbol{Y}, t)$, where $\boldsymbol{X} \in \mathbb{R}^{N \times T \times C}$ denotes the EEG inputs with $N$ samples, each represented by $T$ time steps and $C$ channels; $\boldsymbol{Y} \in \mathbb{R}^{N \times |\mathcal{Y}_d|}$ denotes the corresponding dataset-specific labels, where $|\mathcal{Y}_d|$ depends on the dataset; and $t$ is a task identifier specifying the experimental paradigm. The objective of LEMs is to learn generalizable representations, while performing conditional mappings $f(\boldsymbol{X} \mid t) \to \boldsymbol{Y}$. Based on this definition, the proposed decoder-centric paradigm differs from the two existing ones.

**Encoder-centric LEMs:** The conditional mapping from EEG to task-specific label is modeled as:

$$f(\boldsymbol{X} \mid t) = \mathcal{C}(\mathcal{E}(\boldsymbol{X}; \theta_d); \phi_d) \;\; \to \;\; \boldsymbol{Y}, \tag{1}$$

where $\mathcal{E}_{\theta_d}(\,\cdot\,; \theta_d)$ and $\mathcal{C}_{\phi_d}(\,\cdot\,; \phi_d)$ denote the encoder and classifier with parameters $\theta_d, \phi_d$ fine-tuned on dataset $\mathcal{D}$. Consequently, this paradigm fails to generalize across datasets.

**LLM-centric LEMs:** The mapping incorporates both EEG and auxiliary textual prompts, with the decoder instantiated as an LLM:

$$f(\boldsymbol{X} \mid t) = \mathcal{D}_{\text{LLM}}(\mathcal{E}(\boldsymbol{X}), \texttt{<|text|>}) \to \texttt{<|y|>}, \tag{2}$$

where $\mathcal{E}(\boldsymbol{X})$ encodes the EEG tokens, $\texttt{<|text|>}$ denotes textual tokens, and $\mathcal{D}_{\text{LLM}}(\cdot)$ is the LLM decoder that operates in the text embedding space. The output $\texttt{<|y|>}$ is a textual label. Thus, the key distinction from the encoder-centric paradigm lies in shifting the mapping into the text embedding space.

**Decoder-centric LEMs:** The proposed paradigm represents both inputs and outputs as structured sequences, guiding LEMs to perform multi-task EEG learning and contextual modeling within a unified decoding framework:

$$\begin{aligned} \mathbf{S}_{\text{in}} &= \{\texttt{<|special|>}, \{\mathcal{E}(\boldsymbol{X}_s)\}_{s=1}^S, \mathcal{E}(\boldsymbol{X}), \texttt{<|support|>}\}, \\ \mathbf{S}_{\text{out}} &= \{\texttt{<|support|>}, \texttt{<|task|>}, \texttt{<|y|>}, \texttt{<|special|>}\}, \end{aligned} \tag{3}$$

$$f(\boldsymbol{X} \mid t) = \mathcal{D}(\mathbf{S}_{\text{in}}) \to \mathbf{S}_{\text{out}}, \tag{4}$$

where $\texttt{<|special|>}$ denotes special tokens, such as start or delimiter symbols, which provide structural cues for sequence decoding. $\{\mathcal{E}(\boldsymbol{X}_s)\}_{s=1}^S$ and $\mathcal{E}(\boldsymbol{X})$ represent the collection of support EEG tokens and the target EEG token. $\texttt{<|support|>}$ refers to task and label tokens for support samples. Through next-token prediction, $\mathcal{D}(\cdot)$ infers the task token $\texttt{<|task|>}$ and label token $\texttt{<|y|>}$ of the target sample by leveraging the mapping relationships established from the support samples. Therefore, under this Seq2Seq learning scheme, the decoder is required to learn mappings beyond label prediction (see Section 3.2 for details).

### 2.2 TECHNICAL CHALLENGES

While decoder-centric LEMs hold promise for advancing a new framework for LEMs, their implementation introduces three key technical challenges. We detail these challenges below and present the corresponding technical contributions in Section 3.

**C1: Inconsistency of EEG channels.** The number of EEG channels $C$ and their ordering $\pi(C)$ are not standardized across datasets, posing significant challenges for generalization. Existing LEMs

attempt to mitigate sensitivity to channel order during training through positional encoding strategies (e.g., asymmetric conditional positional encoding (Wang et al., 2024b)). However, at inference, the model still requires channel configurations to exactly match those seen during training. In multi-dataset settings, particularly in cross-dataset scenarios, encountering unseen channel arrangements disrupts spatial alignment and substantially degrades performance. To address this issue, we adopt a channel alignment preprocessing strategy (see Section 3.1).

**C2: Heterogeneity of sequence components.** The input and output sequences often consist of heterogeneous tokens, which creates difficulties for modeling. EEG requires capturing fine-grained temporal evolution, while discrete symbols encode semantic or task-control logic. Directly mixing these heterogeneous elements makes it difficult to balance continuous and discrete information. Furthermore, EEG samples may serve different functional roles (e.g., context vs. prediction targets), which the model must distinguish despite their homogeneous form. To address this, we propose a hybrid positional encoding mechanism (see Section 3.2).

**C3: Absence of symbolic structure in EEG.** Unlike language, EEG lacks discrete symbolic structure, making it difficult for LEMs to acquire ICL naturally. In language models, autoregressive pretraining over diverse discrete contexts (e.g., documents, dialogues) enables next-token prediction to act as implicit function fitting, allowing examples to be reused at inference. EEG models, however, are trained on continuous temporal dynamics that require strict temporal coherence, preventing flexible context transfer across tasks or samples. To address this, we propose a Seq2Seq-based in-context training approach (see Section 3.3).

## 3 METHOD

In this section, we present the methodology of decoder-centric LEMs and the technical contributions that address the aforementioned challenges. Section 3.1 introduces the overall architecture design of ECHO. Section 3.2 explains how ECHO operates under the Seq2Seq formulation. Section 3.3 describes the training objectives and optimization strategies that endow LEMs with ICL. Each section begins with an ***Intuition*** subsection that outlines the rationale behind the technical design.

### 3.1 MODEL ARCHITECTURE

***Intuition:*** The design of ECHO follows a core principle that employs simple and established architectural components to highlight the impact of the paradigm shift itself.

**(a)** For channel unification, we intentionally adopt a straightforward yet general preprocessing block to avoid confounding the core contribution of ECHO (stronger alternatives, such as edge-learning strategies used in GNN-based EEG models (Liu et al., 2024a), can replace it). Because EEG acquisition adheres to standardized electrode systems with recorded channel information, channels can be normalized at the preprocessing stage (Figure 2 **a** (left)).

**(b)** For the model components, we build on established networks: the encoder block (Figure 2 **a** (middle)) is a simplified deep ConvNet (Schirrmeister et al., 2017) with a tokenizer, and the decoder block (Figure 2 **a** (right)) is a transformer decoder with pre-activation residual blocks (Child et al., 2019). This emphasizes that ECHO is not bound to specific modules but represents a paradigm that can flexibly integrate stronger architectures to further improve performance.

**Preprocess Block:** To resolve **C1**, we establish a standardized template channel set based on prior neuroscience knowledge, where the number of channels $\overline{\mathcal{C}}$ and their ordering $\pi(\overline{\mathcal{C}})$ are predefined and fixed. For each standardized channel $\overline{c}$, we define a mapping set $\mathcal{M}_{\overline{c}}$, which contains all possible aliases or adjacent variants of that channel under different electrode systems (the template set is provided in Section 4.1). Given an EEG collection $\boldsymbol{X}$ with channel number $C$ and corresponding channel names, we map each channel name in $\boldsymbol{X}$ to the appropriate $\mathcal{M}_{\overline{c}}$ following the ordering $\pi(\overline{\mathcal{C}})$, yielding the matched channel subset $\boldsymbol{X}_{\overline{c}} \subseteq \boldsymbol{X}$. Then, the alignment process is computed as:

$$\overline{\boldsymbol{X}} = \left\{ \frac{1}{|\boldsymbol{X}_{\overline{c}}| + 1} \sum_{x \in \boldsymbol{X}_{\overline{c}}} x \;\middle|\; \overline{c} \in \pi(\overline{\mathcal{C}}) \right\} \in \mathbb{R}^{N \times T \times |\overline{\mathcal{C}}|}, \tag{5}$$

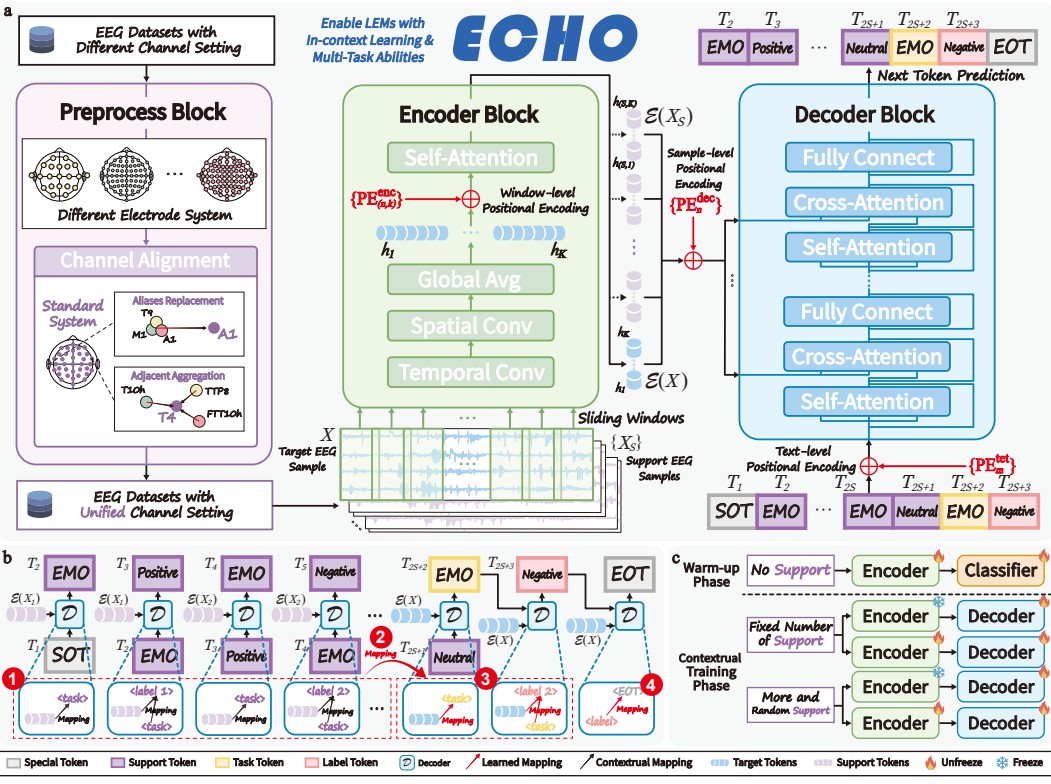

Figure 2: **a**, Overview of the ECHO framework. From left to right: preprocessing, encoder, and decoder blocks. The inputs and outputs are shown at the top-left, bottom-right, and top-right corners, respectively. **b**, Four sequential learning steps within the sequence format enable ECHO to capture diverse mapping relations. **c**, Multi-stage training strategy of ECHO.

where $\overline{\boldsymbol{X}}$ denotes the aligned EEG with standardized channel configuration. The normalization term $|\boldsymbol{X}_{\overline{c}}| + 1$ ensures stable averaging. If $|\boldsymbol{X}_{\overline{c}}| = 0$, the channel is padded with zero.

**Encoder Block:** The encoder block is designed to transform preprocessed EEG signals $\overline{\boldsymbol{X}} \in \mathbb{R}^{N \times T \times \overline{C}}$ into tokens. First, $\overline{\boldsymbol{X}}$ is segmented along the temporal dimension using sliding windows of length $L$ and stride $S$, yielding a collection of $K = (T - L)/S + 1$ segments denoted as $\{x_1, x_2, \ldots, x_K\}$. Each segment is processed by a convolutional head $\mathcal{F}_{\text{conv}}(\cdot)$ based on the deep ConvNet and tokenized into a sequence of vectors:

$$\mathcal{E}(\boldsymbol{X}) = \{h_1, h_2, \cdots, h_K\} = \mathcal{T}\big(\{h_k\}_{k=1}^K\big) = \mathcal{T}\big(\bigcup_{k=1}^K \mathcal{F}_{\text{conv}}(\{x_k\})\big), \tag{6}$$

where $\mathcal{E}(\boldsymbol{X})$ represents the sample-level EEG tokens produced by the encoder. $\mathcal{T}(\cdot)$ represents the tokenization process (including self-attention), which flattens and projects segment features into window-level tokens and arranges them into a sequence. Notably, since the tokenizer operates along the window dimension, we use $h_k$ to denote the window-level tokens and segment features.

**Decoder Block:** The decoder $\mathcal{D}(\cdot)$ adopts a standard transformer decoder architecture, where self-attention models the dependencies within the textual sequence and cross-attention enables interaction with $\mathcal{E}(\boldsymbol{X})$:

$$\mathbf{S}_{\text{out}} = \mathcal{D}\big(\mathbf{S}_{\text{in}}\big). \tag{7}$$

### 3.2 SEQ2SEQ FORMULATION

*Intuition:* The Seq2Seq formulation guides ECHO to perform progressive learning through a fixed serialization scheme.

**(a)** A fixed sequence corresponds to a consistent "problem-solving strategy." As illustrated in Figure 2 **b**, ① the support EEG samples and their tokens serve as worked examples, enabling ECHO to learn mappings between EEG, task and label tokens; ② the model then generalizes these mappings from examples to the target sample; ③ ECHO then conducts stepwise reasoning by first predicting the task token and subsequently deriving the label token conditioned on both the task and EEG tokens; ④ finally, by predicting the end-of-task (EOT) token, ECHO learns to recognize task termination. Through this unified sequence, ECHO acquires both in-context and multi-task learning capabilities, allowing it to autonomously select the most compatible label token from all known labels without requiring an explicit task specification.

**(b)** To prevent confusion among heterogeneous components, ECHO employs a three-part positional encoding strategy. The first models the temporal structure within each EEG sample, allowing the model to capture the sequential dynamics of neural activity. The second distinguishes support from target samples, clarifying their functional roles in the sequence. The third encodes the semantics of textual markers such as task tokens, label tokens, and the EOT token. Together, these positional cues enable the model to jointly handle the continuous dynamics of EEG and the discrete logic of tasks within a single serialized space.

**Sequence Format: During training,** the input sequence starts with `<|SOT|>`, followed by multiple `<|support|>` entries that encode the task and label tokens of the support EEG samples. The `<|task|>` token then specifies the paradigm of the target EEG (e.g., `<|MI|>`, `<|EMO|>`), and `<|y|>` denotes its ground-truth label. The output sequence mirrors this structure but ends with `<|EOT|>` to mark completion. **During inference,** the model receives `<|SOT|>` followed by the target EEG tokens (optionally with support samples). ECHO then generates only the task token and the predicted label token, and terminates with `<|EOT|>`.

**Hybrid Positional Encoding:** To solve **C2**, we propose a hybrid positional encoding. Given an EEG sample $X_n$, the window-level tokens $\{h_{(n,1)}, h_{(n,2)}, \ldots, h_{(n,K)}\}$ obtained along the temporal dimension are encoded with learnable window-level position encoding:

$$\bigcup_{k=1}^{K} \left\{ h_{(n,k)} + \text{PE}^{\text{enc}}_{(n,k)} \right\}, \tag{8}$$

where $\text{PE}^{\text{enc}}_{(n,k)}$ denotes the learnable positional encoding assigned to the $k$-th segment within the EEG sample. As for the decoder input, given a hybrid EEG sample set $\{\mathcal{E}(X_n)\}_{n=1}^{S+1}$ consisting of the target sample $\mathcal{E}(X)$ and support samples $\{\mathcal{E}(X_s)\}_{s=1}^{S}$, a distinct learnable sample-level positional encoding is assigned and uniformly added to all tokens within the same sample:

$$\bigcup_{n=1}^{S+1} \left\{ \mathcal{E}(X_n) + \text{PE}^{\text{dec}}_n \right\}, \tag{9}$$

where $\text{PE}^{\text{dec}}_n$ denotes the sample-level positional encoding shared across all tokens of the $n$-th EEG sample. For the sequence of textual tokens $\{T_m\}_{m=1}^{2S+3}$, standard learnable positional encodings $\{\text{PE}^{\text{txt}}_m\}_{m=1}^{2S+3}$ are applied:

$$\bigcup_{m=1}^{2S+3} \left\{ T_m + \text{PE}^{\text{txt}}_m \right\}, \tag{10}$$

## 3.3 IN-CONTEXT TRAINING

*Intuition:* Unlike large language models, where ICL often emerges implicitly, LEMs require explicit guidance to acquire this capability. Thus, ECHO is trained with autoregressive next-token prediction under a multi-stage strategy: first, the encoder is initialized to accelerate convergence and yield usable EEG representations; then, the decoder is trained with progressively larger and more diverse support sets to develop multi-task classification and contextual learning capabilities.

**Training Strategy:** To address C3, ECHO is trained in two phases. As shown in Figure 2c, the Warm-up Phase initializes the encoder by pairing it with a unified classifier across all datasets to obtain stable representations. The Contextual Training Phase then follows, consisting of two rounds:

the first uses a fixed number of support samples to stabilize decoder training, and the second randomizes the support size to expose the model to diverse contexts. In both rounds, training starts with the encoder frozen and later jointly optimized with the decoder.

**Next-token prediction:** During training, given a sequence of textual tokens $\{T_n\}_{n=1}^{2S+3}$ and EEG sample set $\{\mathcal{E}(X_\cup)\}$, the decoder $\mathcal{D}$ generates the output sequence step by step. The conditional probability for the $i$-th output token is modeled as

$$p(s_i \mid s_{<i}, \{T_n\}_{n=1}^{2S+3}, \{\mathcal{E}(X_\cup)\}) = \mathcal{D}(s_{<i}, \{T_n\}_{n=1}^{2S+3}, \{\mathcal{E}(X_\cup)\}), \quad (11)$$

where $s_i \in \mathbf{S}_{\text{out}}$ denotes the $i$-th token to be predicted, and $s_{<i}$ represents the prefix subsequence of previously generated tokens $\{s_1, s_2, \ldots, s_{i-1}\}$. The training objective minimizes the cross-entropy loss between predicted distributions and ground-truth tokens.

## 4 EXPERIMENT

### 4.1 EXPERIMENT SETUP

**Dataset Setting:** ECHO was trained on 12 publicly available EEG datasets spanning six task categories and 26 classes. **(1)** All datasets underwent a unified preprocessing pipeline—band-pass filtering, downsampling to 250 Hz, task-specific segmentation, and padding to 10s. **(2)** Heterogeneous electrode layouts were aligned to a standardized 75-channel system (see Section 3.1). **(3)** A consistent cross-subject split was applied across all experiments and baselines to ensure fair and generalizable evaluation. Table 1 summarizes the splits, task types, and input formats for the datasets reported in the main text (excluding BCIC 2020-T1 from training), with full details in Appendix A.1.

Table 1: Dataset Configurations

| Dataset | Experimental Paradigms | Train Indices | Validation Indices | Test Indices | Shape |
|---|---|---|---|---|---|
| BCIC-IV-2a (Brunner et al., 2008) | Multi-Limb Motor Imagery | 0–4 | 5–6 | 7–8 | 22 channels × 4s |
| High-Gamma (Schirrmeister et al., 2017) | Motor Imagery for Decoding | 0–7 | 8–10 | 11–13 | 128 channels × 4s |
| BCIC 2020-T1 (Jeong et al., 2022) | Hand Motor Imagery | 0–9 | 10–14 | 15–19 | 62 channels × 4s |
| SEED-IV (Zheng et al., 2018) | Film-induced Discrete Emotion Classification | 0–9 | 10–11 | 12–14 | 62 channels × 4s |
| SEED (Zheng & Lu, 2015) | Film-induced Emotional Valence Classification | 0–9 | 10–11 | 12–14 | 62 channels × 4s |
| Stieger2021-LR (Stieger et al., 2021) | Continuous 1D Cursor Control (Lateral) | 0–39 | 40–48 | 49–58 | 15 channels × 4s |
| Mumtaz2016 (Mumtaz, 2016) | Major Depressive Disorder Detection | 0–43 | 43–52 | 52–62 | 19 channels ×5s |
| Mental Arithmetic (Zyma et al., 2019) | Workload Assessment | 0–25 | 26–30 | 31–35 | 20 channels × 5s |
| Attention (Shin et al., 2018) | Discrimination/Selection Response | 0–15 | 16–20 | 21–25 | 30 channels × 4s |

**Baseline Selection:** To evaluate ECHO, we compared it with six representative baselines covering diverse representation learning paradigms. EEGNet (Lawhern et al., 2018) employs convolution to extract features from raw signals; BIOT (Yang et al., 2023) uses block-based continuous tokenization; and LaBraM (Jiang et al., 2024b) combines masked reconstruction with vector quantization. EEGPT (Wang et al., 2024a) and CBraMod (Wang et al., 2024b) emphasize masked reconstruction of raw signals, while CodeBrain (Ma et al., 2025) learns by predicting discrete time–frequency tokens. Full baseline details are in Appendix A.2.

**Model Setting:** After preprocessing, EEG signals were segmented with a sliding window (length 100, stride 90). The encoder comprises 4 convolutional layers and a tokenizer based on multi-head self-attention (8 heads, 4 layers, token dim 256). The decoder adopts a 4-layer Transformer with hidden size 384, 6 heads, and feed-forward dim 1536. Full model details are in Appendix A.3.

**Training & Environment Setting:** All training was conducted on 8×NVIDIA A100 (40GB) GPUs. The process had two stages. In the warm-up phase, the encoder was trained for 90 epochs (batch size 64) using Adam with an initial learning rate of $5 \times 10^{-5}$, cosine-decayed to $1 \times 10^{-6}$, and dropout 0.2 to stabilize EEG feature extraction. In the contextual training phase, the full model was trained for 40 epochs (batch size 48, dropout 0.1) with differential learning rates ($5 \times 10^{-5}$ for the decoder, $5 \times 10^{-6}$ for the encoder). Training followed a two-round schedule: 10 epochs with a fixed 8-shot configuration, followed by randomized support counts (0–12) to expose the model to varied contexts and improve ICL robustness. Additional details are provided in Appendix A.4.

**Evaluation Metrics:** We adopt four standard evaluation metrics: Balanced Accuracy, Cohen's Kappa, Weighted F1 score, and the Area Under the ROC and Precision-Recall Curves (AUROC and AUC-PR). Model selection is based on AUROC performance on the validation set. All experiments and baselines are conducted with five fixed random seeds 0, 1, 2, 3, 4, and we report the mean and standard deviation across runs.

Table 2: Comparison results of different methods on downstream tasks.

| Methods | SEED | | | Stieger2021-LR | | |
|---|---|---|---|---|---|---|
| | ACC-B | ROC AUC | PR AUC | ACC-B | ROC AUC | PR AUC |
| EEGNet | 0.7435 ± 0.0315 | 0.8631 ± 0.0235 | 0.8731 ± 0.0331 | 0.8051 ± 0.0124 | 0.8839 ± 0.0123 | 0.8565 ± 0.0078 |
| BIOT | 0.7234 ± 0.0215 | 0.8212 ± 0.0349 | 0.8043 ± 0.0156 | 0.7753 ± 0.0052 | 0.8747 ± 0.0054 | 0.8247 ± 0.0054 |
| EEGPT | 0.7085 ± 0.0350 | 0.8450 ± 0.0241 | 0.8244 ± 0.0194 | 0.7943 ± 0.0043 | 0.8968 ± 0.0057 | 0.8354 ± 0.0042 |
| LaBraM | 0.6851 ± 0.0431 | 0.7952 ± 0.0241 | 0.8021 ± 0.0136 | 0.8170 ± 0.0037 | 0.9024 ± 0.0015 | 0.8935 ± 0.0029 |
| CBraMod | 0.7262 ± 0.0235 | 0.8519 ± 0.0179 | 0.8400 ± 0.0232 | 0.8424 ± 0.0044 | 0.9339 ± 0.0026 | 0.9297 ± 0.0030 |
| CodeBrain | 0.7836 ± 0.0341 | 0.8755 ± 0.0248 | 0.8543 ± 0.0253 | 0.8126 ± 0.0037 | 0.9123 ± 0.0024 | 0.8932 ± 0.0031 |
| ECHO$^{\mathcal{E}}$ | 0.6548 ± 0.0272 | 0.7493 ± 0.0451 | 0.7592 ± 0.0378 | 0.6123 ± 0.0242 | 0.8918 ± 0.0206 | 0.9048 ± 0.0217 |
| ECHO (No Support) | 0.7407 ± 0.0047 | 0.8488 ± 0.0108 | 0.8522 ± 0.0103 | 0.8415 ± 0.0031 | 0.9245 ± 0.0021 | 0.9243 ± 0.0027 |
| ECHO | **0.8193** ± 0.0025 | **0.9020** ± 0.0004 | **0.8962** ± 0.0020 | **0.8534** ± 0.0014 | **0.9349** ± 0.0001 | **0.9363** ± 0.0016 |

| Methods | Mumtaz2016 | | | High-Gamma | | |
|---|---|---|---|---|---|---|
| | ACC-B | ROC AUC | PR AUC | ACC-B | ROC AUC | PR AUC |
| EEGNet | 0.9113 ± 0.0104 | 0.9512 ± 0.0096 | 0.9632 ± 0.0045 | 0.8320 ± 0.0289 | 0.8911 ± 0.0412 | 0.9002 ± 0.0291 |
| BIOT | 0.8789 ± 0.0190 | 0.9664 ± 0.0136 | 0.9744 ± 0.0083 | 0.7343 ± 0.0641 | 0.7931 ± 0.0372 | 0.8198 ± 0.0274 |
| EEGPT | 0.8475 ± 0.0233 | 0.9669 ± 0.0069 | 0.9695 ± 0.0076 | 0.7161 ± 0.0481 | 0.8276 ± 0.0385 | 0.8249 ± 0.0632 |
| LaBraM | 0.8986 ± 0.0028 | 0.9754 ± 0.0050 | 0.9791 ± 0.0041 | 0.7429 ± 0.0386 | 0.8516 ± 0.0255 | 0.8454 ± 0.0246 |
| CBraMod | 0.8946 ± 0.0047 | 0.9800 ± 0.0045 | 0.9765 ± 0.0061 | 0.7513 ± 0.0182 | 0.8277 ± 0.0176 | 0.8335 ± 0.0146 |
| CodeBrain | 0.9012 ± 0.0021 | 0.9729 ± 0.0037 | 0.9721 ± 0.0078 | 0.8138 ± 0.0217 | 0.8421 ± 0.0395 | 0.8601 ± 0.0102 |
| ECHO$^{\mathcal{E}}$ | **0.9698** ± 0.0012 | **0.9953** ± 0.0023 | **0.9952** ± 0.0015 | 0.8039 ± 0.0436 | 0.8900 ± 0.0279 | 0.8889 ± 0.0314 |
| ECHO (No Support) | 0.9056 ± 0.0211 | 0.9745 ± 0.0103 | 0.9748 ± 0.0058 | 0.8438 ± 0.0023 | 0.9125 ± 0.0016 | 0.9047 ± 0.0034 |
| ECHO | N/A | N/A | N/A | **0.8552** ± 0.0031 | **0.9208** ± 0.0011 | **0.9125** ± 0.0041 |

| Methods | Mental Arithmetic | | | Attention | | |
|---|---|---|---|---|---|---|
| | ACC-B | ROC AUC | PR AUC | ACC-B | ROC AUC | PR AUC |
| EEGNet | 0.5138 ± 0.0471 | 0.5395 ± 0.0109 | 0.5302 ± 0.0441 | 0.6004 ± 0.0123 | 0.6647 ± 0.0180 | 0.6294 ± 0.0288 |
| BIOT | 0.5281 ± 0.0384 | 0.5970 ± 0.0468 | 0.5567 ± 0.0249 | 0.6111 ± 0.0411 | 0.7367 ± 0.0162 | 0.7273 ± 0.0132 |
| EEGPT | 0.5117 ± 0.0317 | 0.5612 ± 0.0198 | 0.5041 ± 0.0384 | 0.6674 ± 0.0560 | 0.8015 ± 0.0372 | 0.8103 ± 0.0303 |
| LaBraM | 0.5793 ± 0.0631 | 0.5700 ± 0.0232 | 0.6341 ± 0.0422 | 0.6785 ± 0.0223 | 0.7838 ± 0.0307 | 0.7994 ± 0.0198 |
| CBraMod | 0.5906 ± 0.0531 | 0.5045 ± 0.0519 | 0.7047 ± 0.0428 | 0.6478 ± 0.0258 | 0.7417 ± 0.0175 | 0.7468 ± 0.0198 |
| CodeBrain | 0.6318 ± 0.0845 | 0.6472 ± 0.0361 | 0.7412 ± 0.0451 | 0.6215 ± 0.0358 | 0.6321 ± 0.0281 | 0.7029 ± 0.0349 |
| ECHO$^{\mathcal{E}}$ | 0.6008 ± 0.0343 | 0.6555 ± 0.0281 | 0.6416 ± 0.0337 | 0.7422 ± 0.0346 | 0.7949 ± 0.0623 | 0.8021 ± 0.0278 |
| ECHO (No Support) | 0.5442 ± 0.0023 | 0.6896 ± 0.0036 | 0.6897 ± 0.0042 | 0.8056 ± 0.0021 | 0.8895 ± 0.0029 | **0.8955** ± 0.0034 |
| ECHO | **0.6851** ± 0.0032 | **0.7500** ± 0.0062 | **0.7530** ± 0.0015 | **0.8194** ± 0.0009 | **0.8973** ± 0.0019 | 0.8952 ± 0.0027 |

Note: **Bold** indicates the best performance. Cyan highlight marks ECHO.

**Task Setting:** All baselines are evaluated under a **single-task setting**, where each dataset is fine-tuned separately and tested on its own test set. In contrast, ECHO is evaluated under a strict **multi-task setting**: trained once across all datasets without task-specific fine-tuning and directly tested on all test sets in a single pass. For ICL, 20 instances per subject are randomly sampled from each standardized test set as fixed context and excluded from evaluation. ECHO is given 8 support samples, but no task tokens, and must autonomously infer both the task paradigm and subcategories. The only exception is the Mumtaz2016 dataset, where each subject has a single label; evaluation for it is performed without supports.

## 4.2 EXPERIMENT RESULT

As shown in Table 2, six representative downstream datasets (different tasks) under a unified experimental setup (differing only in single-task vs. multi-task training) are reported. Complete results are reported in Appendix B.1. ECHO$^{\mathcal{E}}$ denotes the ECHO encoder trained without the Seq2Seq paradigm, while ECHO (No Support) refers to the ECHO model evaluated without any support-sample prompts. The advantages and corresponding limitations of ECHO are as follows:

**(a) Comparison between ECHO and Baselines:**

**ECHO exhibits strong generalization capability.** On cognitive tasks (SEED, Stieger2021-LR, Mental Arithmetic, and Attention), ECHO achieves an average improvement of +0.0602 in Balanced Accuracy, +0.0566 in ROC AUC, and +0.0316 in PR AUC over the strongest baseline. On clinical diagnostic tasks (Mumtaz2016 and High-Gamma), the best-performing ECHO variant shows an average gain of +0.0409 in Balanced Accuracy, +0.0225 in ROC AUC, and +0.0142 in PR AUC over the strongest baseline. ECHO also demonstrates a unique capability: even without any external prompts, it can autonomously identify the corresponding task and its specific paradigm solely from the EEG sample itself.

**ECHO does not surpass SOTA on several tasks.** On tasks such as Stieger2021-UD, BCIC-IV-2a, and SEED-IV, ECHO remains noticeably below encoder-centric models that are explicitly optimized for those domains. This highlights a limitation in ECHO's generalization–specialization trade-off: while the framework aims for unified cross-task reasoning, its lightweight encoder design is insufficient to model all domain-specific structures required by certain specialized EEG tasks. Detailed results and analysis are provided in Appendix B.1.

**(b) Comparison between ECHO and ECHO$^{\mathcal{E}}$:**

**Seq2Seq paradigm improves performance.** ECHO outperforms ECHO$^{\mathcal{E}}$ on most datasets, indicating that the full Seq2Seq paradigm is a primary contributor to ECHO's gains. An exception is Mumtaz2016, where ECHO$^{\mathcal{E}}$ achieves the best performance (Table 2). Removing the sequential structure often leads to substantial drops (e.g., SEED ACC-B: 0.8193 → 0.6548).

**ECHO's performance ceiling is constrained by encoder quality.** While Seq2Seq offers consistent gains, their impact is limited when the encoder cannot model the dataset's structure, as seen in TUEV and PhysioNet. In these cases, ICL can provide improvements but cannot overcome the encoder's representational shortcomings or match encoder-centric SOTA models. Full results are in Appendix B.1.

**(c) Comparison between ECHO and No Support:**

**ICL provides additional gains.** ECHO surpasses its No-Support setting across all benchmarks, confirming the effectiveness of in-context learning. Without support samples, performance decreases notably (e.g., SEED ACC-B: 0.8193 → 0.7407).

**The effectiveness of ICL depends strongly on the distributional stability of support samples.** This works well in datasets with clear structure, such as High-Gamma and SEED. In most EEG datasets, substantial cross-subject variability and noise make support unstable; few support samples provide insufficient signal, whereas many support samples introduce accumulated noise and distribution shifts that reduce performance. Full results are in Appendix D.2.2.

## 4.3 ABLATION STUDY

We conducted ablation experiments to evaluate the two additional positional encodings introduced in ECHO. As shown in Figure 3, removing either encoding makes the model ineffective. **(1)** Removing the sample-level positional encodings caused performance to drop to chance level, as the model could no longer distinguish boundaries between EEG samples and instead treated them as a continuous sequence. **(2)** Removing the decoder textual positional encodings led to complete structural collapse, with the model producing disordered symbol sequences and no valid predictions. These results highlight that both encodings are indispensable: the former ensures functional separation of EEG samples, while the latter preserves syntactic and semantic coherence in decoding.

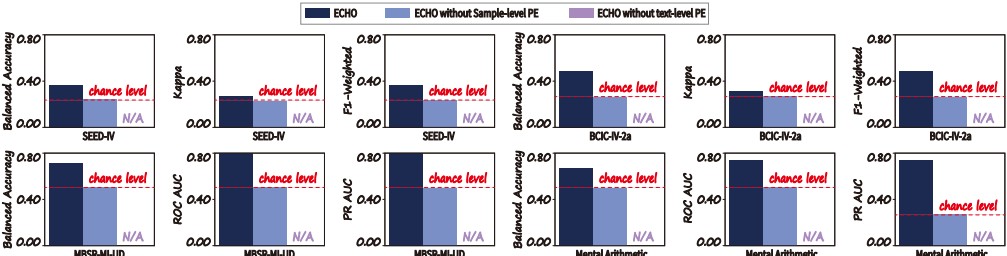

Figure 3: The result of the ablation study for positional encoding

## 4.4 ZERO-SHOT EVALUATION

As shown in Table 3, we evaluate ECHO's zero-shot generalization on two unseen datasets: Cho2017 and BCIC 2020-T1. On BCIC 2020-T1, ECHO exhibits strong cross-dataset transfer, applying the mapping strategies learned during multi-task training even without task prompts, while support samples further enhance performance. On Cho2017, ECHO again surpasses the encoder-only baseline, indicating that the Seq2Seq formulation provides stable adaptation to new acquisition settings and task paradigms. Additional generalization experiments are provided in Appendix B.3

Table 3: Result of Zero-Shot Experiment.

| Methods | Cho2017 (Unseen for ECHO) | | | BCIC 2020-T1 (Unseen for ECHO) | | |
|---|---|---|---|---|---|---|
| | ACC-B | ROC AUC | PR AUC | ACC-B | ROC AUC | PR AUC |
| CBraMod | $0.7033 \pm 0.0327$ | $0.8081 \pm 0.0429$ | $0.7959 \pm 0.0230$ | $0.5700 \pm 0.0185$ | $0.6058 \pm 0.0236$ | $0.5371 \pm 0.0186$ |
| ECHO$^{\mathcal{E}}$ | $0.5640 \pm 0.0315$ | $0.6021 \pm 0.0498$ | $0.6126 \pm 0.0207$ | $0.5250 \pm 0.0751$ | $0.6796 \pm 0.0146$ | $0.6697 \pm 0.0602$ |
| ECHO(No Support) | $0.5881 \pm 0.0023$ | $0.6399 \pm 0.0035$ | $0.6267 \pm 0.0021$ | $0.7500 \pm 0.0024$ | $0.8232 \pm 0.0053$ | $0.8153 \pm 0.0118$ |
| ECHO | $\mathbf{0.6044} \pm 0.0032$ | $\mathbf{0.6470} \pm 0.0053$ | $\mathbf{0.6460} \pm 0.0019$ | $\mathbf{0.7782} \pm 0.0042$ | $\mathbf{0.8566} \pm 0.0028$ | $\mathbf{0.8552} \pm 0.0032$ |

Note: **Bold** indicates the best performance. Cyan highlight marks ECHO.

## 4.5 EXTRA EXPERIMENT

To evaluate ECHO under more diverse and clinically relevant conditions, we introduce three specialized variants in the appendix. The long-sequence variant in Appendix D.1.1 (ECHO$^{\mathcal{L}}$) shows the Seq2Seq + ICL framework remains effective under unified 30-second windows. The motor-imagery variant in Appendix D.1.2 (ECHO$^{\mathcal{MI}}$) indicates strong cross-dataset transfer across MI datasets. The epilepsy-augmented variant in Appendix D.1.3 (ECHO$^{EP}$) further demonstrates that support examples substantially improve seizure detection, highlighting ICL's ability to compensate for limited seizure-specific capacity.

We include additional diagnostic studies in the appendix to validate ECHO's design choices and framework-level contributions. Appendix D.2.1 analyzes channel-fusion and shows simple averaging is sufficient across heterogeneous layouts. Appendix D.2.2 studies support-size sensitivity and finds performance typically peaks at moderate support sizes. Appendix D.2.3 confirms the extensibility of the Seq2Seq paradigm, while Appendix D.2.4 shows more stable generalization than standard multi-task learning with fewer parameters. Overall, these results support that ECHO's gains stem from the Seq2Seq + ICL framework rather than encoder capacity or task-specific shortcuts.

## 5 RELATED WORK

In recent years, research on large models for neural signals has increasingly centered on representation learning, with the primary goal of extracting robust and generalized representations. Existing approaches can be broadly categorized into two directions: reconstruction-based and contrastive-based representation learning. **In reconstruction-based methods**, representations are learned by recovering missing or future signals through masking or autoregression. For example, frequency-domain masking has been applied to enforce temporal-frequency consistency in representations (Wang et al., 2023), while spatiotemporal joint masking has been introduced to model dependencies across both temporal and spatial domains (Dong et al., 2024). Autoregressive frameworks further extend this idea by predicting future signal segments, enabling representations that capture long-range dynamics (Caro et al., 2023). **In contrastive-based methods**, representations are improved by constructing positive and negative pairs to enhance robustness and discriminability. Temporal perturbations and frequency shifts have been used to ensure consistent representations across augmented views (Cai et al., 2023), while cross-modal contrastive learning has been explored to expand the representational space, such as aligning EEG with text to ground neural signals in richer semantic domains (Jiang et al., 2024a).

## 6 CONCLUSION

In this work, we introduced ECHO, a decoder-centric paradigm for LEMs, designed to highlight the untapped potential of decoders in EEG representation learning and task modeling. ECHO adopts a Seq2Seq formulation that jointly models the hierarchical relationships among signals, labels, and tasks within a unified sequence space, while leveraging discrete support samples to enable ICL. This design allows the model to dynamically adapt to diverse tasks without parameter updates. Extensive experiments on multiple public EEG datasets demonstrate that, even with basic architectural components, ECHO consistently outperforms state-of-the-art single-task LEMs in multi-task settings, and further exhibits generalization in zero-shot and cross-dataset evaluations. Overall, these results show that ECHO provides a viable pathway to overcoming the decoder bottleneck in existing LEMs.

## ACKNOWLEDGEMENT

This work is supported by the Youth Science Fund Project of National Natural Science Foundation of China (Grant No. 62306317), and the Open Research Fund of the Key Laboratory of Social Computing and Cognitive Intelligence (Dalian University of Technology), Ministry of Education (Grant No. SCCI2025YB01), and Sponsored by Beijing Nova Program (Grant No. 20250484804), and Young Elite Scientists Sponsorship Program of the Beijing High Innovation Plan (Grant No. 20250912).

## ETHICS STATEMENT

All datasets utilized in this study are publicly available and distributed under appropriate usage licenses. The data are processed and presented exclusively in aggregated, privacy-preserving formats, ensuring that no personally identifiable information is disclosed. We have thoroughly reviewed the possible ethical implications of this research and foresee no risks or adverse outcomes associated with its application.

## REPRODUCIBILITY

We are committed to ensuring the reproducibility of all experimental results. The code has been made publicly available via a GitHub repository:
*https://github.com/wythedee/ECHO*.

## USE OF LLMS

The authors affirm that the core research, methodology, and scientific findings presented in this work were independently conceived and developed by the authors. All figures and tables were created and formatted by the authors themselves. The scientific content, including research ideas, experimental design, and result analysis, was not produced by LLMs. LLMs were employed solely to assist in improving the clarity and readability of the manuscript.

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

APPENDIX

# A  EXPERIMENT SETUP DETAIL

## A.1  DATASET SETTING

For the pre-training phase of ECHO, a comprehensive corpus of 12 public datasets was aggregated, spanning six distinct Brain-Computer Interface (BCI) tasks. These include Emotion Recognition, Motor Imagery, Major Depressive Disorder (MDD) detection, workload assessment, event type classification, and attention monitoring. This multi-task, multi-dataset approach is designed to expose the model to a wide variety of EEG signal characteristics, thereby fostering the development of robust and generalizable representations. All signals were uniformly resampled to 250 Hz to ensure consistency across the corpus. A summary of the detailed information for each dataset is shown in Table 4. The detailed introductions for each task are listed below (includes datasets used by all ECHO variants):

Table 4: Detailed Information of Datasets Used for Pre-training.

| Task | Dataset | #Subjects | #Channels | Duration | Sampling Rate | #Classes |
|---|---|---|---|---|---|---|
| | SEED-IV | 15 | 62 | 4s | 250 Hz | 4 |
| Emotion Recognition | SEED-V | 16 | 62 | 1s | 250 Hz | 5 |
| | SEED | 15 | 62 | 4s | 250 Hz | 2 |
| | BCI IV 2a | 9 | 22 | 4s | 250 Hz | 4 |
| | High-Gamma | 14 | 128 | 4s | 250 Hz | 2 |
| Motor Imagery | Stieger2021-LR | 64 | 15 | 4s | 250 Hz | 2 |
| | Stieger2021-UD | 64 | 15 | 4s | 250 Hz | 2 |
| | PhysioNet | 109 | 64 | 10s | 250 Hz | 4 |
| | KoreaU | 54 | 62 | 4s | 250 Hz | 2 |
| MDD Detection | Mumtaz | 119 | 19 | 5s | 250 Hz | 2 |
| Workload Assessment | Mental Arithmetic | 36 | 20 | 5s | 250 Hz | 2 |
| Event Type Classification | TUEV (Events) | 370 | 16 | 5s | 250 Hz | 6 |
| Attention Monitoring | Attention | 26 | 30 | 4s | 250 Hz | 2 |
| Sleep Staging | ISRUC S1 | 100 | 6 | 30s | 250 Hz | 5 |
| | ISRUC S3 | 10 | 6 | 30s | 250 Hz | 5 |
| Seizure Detection | CHB-MIT | 22 | 23 | 10s | 256 Hz | 2 |

### A.1.1  EMOTION RECOGNITION

This task aims to identify human emotional states from EEG signals. The datasets used involve recordings of subjects exposed to stimuli designed to elicit specific emotions.

- **SEED-V**: The SEED-V dataset focuses on the recognition of five distinct emotional categories: **happy**, **sad**, **neutral**, **disgust**, and **fear**. It features **62-channel** electroencephalogram (EEG) recordings acquired from 16 participants. For analytical purposes, these signals are segmented into 1-second windows. It should be noted that data corruption necessitated the exclusion of subject 7, leaving 15 subjects in the SEED-V dataset, which is then partitioned into a 5:5:5 split for training, validation, and testing.

- **SEED-IV**: SEED-IV focuses on the recognition of **four emotional states**: happiness, sadness, fear, and neutrality. Data were collected from **15 subjects** who participated in **3 sessions**, watching a total of 72 film clips chosen to induce these emotions. The data were segmented into **4-second nonoverlapping segments** for analysis. Key features extracted include Power Spectral Density (PSD) and Differential Entropy (DE) across five distinct frequency bands: delta, theta, alpha, beta, and gamma

- **SEED**: SEED originally owns 3 labels, Positive, Neutral and Negative. To align with the settings in baseline, we remove Neutral EEG data and transform it into a binary classification task. Data is segmented into **4-second nonoverlapping segments**.

### A.1.2  MOTOR IMAGERY (MI)

Motor Imagery (MI) is the mental rehearsal of a motor action without any overt physical movement. In the context of EEG-based Brain-Computer Interfaces (BCIs), this task involves classifying dif-

ferent imagined movements—such as those of the left hand, right hand, or feet—from the user's brain signals. These imagined actions elicit distinct patterns of neural activity, particularly within the sensorimotor cortex, which can be decoded to control external devices.

- **BCI IV 2a**: This dataset contains recordings from **9 subjects** performing a cue-based motor imagery task. The task involves four distinct classes: imagined movements of the **left hand**, **right hand**, **feet**, and **tongue**. EEG data was recorded from **22 channels** at a sampling rate of **250 Hz**, with each MI trial lasting for **4 seconds**.

- **High Gamma**: This dataset contains EEG recordings from **14 subjects** performing executed, rather than imagined, motor tasks, with a focus on capturing high-frequency components of brain activity. The paradigm includes four classes: sequential finger-tapping of the **left hand**, finger-tapping of the **right hand**, repetitive toe clenching representing **both feet**, and a **rest** condition. Signals were recorded from **128 channels** and subsequently downsampled to **250 Hz**, with each trial lasting for **4 seconds**.

- **Stieger2021 (LR)**: Part of a study investigating the effects of Mindfulness-Based Stress Reduction (MBSR) on BCI skill acquisition, this dataset involves **64 subjects** performing a horizontal cursor control task. The paradigm consists of two classes: motor imagery of the **left hand** (to move left) and the **right hand** (to move right). Data was recorded from **15 channels** at **250 Hz** and segmented into **4-second** trials.

- **Stieger2021 (UD)**: Sourced from the same subject pool, this dataset focuses on a vertical cursor control task. It includes two classes: motor imagery of **both hands** (to move up) and **voluntary rest** (to move down). The recording setup and trial segmentation are identical to the LR dataset.

- **PhysioNet**: This dataset provides motor imagery EEG recordings from **109 subjects**. The paradigm consists of four classes: imagined movements of the **left fist**, **right fist**, **both fists**, and **both feet**. Data was recorded from **64 channels** with a sampling rate of **250 Hz**, and trials are segmented into **10-second** windows.

- **KoreaU**: This dataset features EEG recordings from 54 subjects performing a binary-class motor imagery task. The paradigm involves two classes: imagined movements of the left hand and the right hand. Signals were recorded from 62 channels at a sampling rate of 1000 Hz, with each MI trial lasting for 4 seconds.

### A.1.3 MAJOR DEPRESSIVE DISORDER (MDD) DETECTION

This task focuses on identifying biomarkers for Major Depressive Disorder from EEG signals, typically differentiating between patients with MDD and healthy controls.

- **Mumtaz**: This dataset is designed for MDD detection, containing EEG recordings from **34 patients with MDD** and **30 healthy controls** during eyes-open and eyes-closed resting states. Signals were recorded from **19 channels** following the 10-20 system and were subsequently downsampled to **250 Hz**. For analysis, the data is segmented into **5-second windows**.

### A.1.4 WORKLOAD ASSESSMENT

This task, often framed as mental stress detection, aims to quantify a subject's cognitive load or stress level based on their EEG signals.

- **Mental Arithmetic**: This dataset supports mental stress detection by recording EEG from 36 subjects under two conditions: a resting state ("no stress") and an active mental arithmetic task ("stress"). The signals were acquired using 20 electrodes and segmented into 5-second windows.

### A.1.5 EVENT TYPE CLASSIFICATION

This task involves the classification of various event types from clinically annotated EEG recordings, which is crucial for automated analysis and diagnosis.

- **TUEV (Events)**: This clinically annotated corpus is used for multi-class event type classification, including six categories such as spike and sharp wave (SPSW), eye movements (EYEM), and artifacts (ARTF). Signals were recorded using 16 bipolar montage channels and segmented into 5-second windows.

Table 5: Hyperparameters for Model Architecture.

| Component | Hyperparameter | Setting |
|---|---|---|
| EEG Sample | Channels | 75 |
| | Time points | 2500 |
| | Patch dimension | 256 |
| | Sequence length | 10 |
| CNN(Encoder) | Window size | 100 |
| | Step | 90 |
| | Input dimensions | {1, 64, 64, 128} |
| | Output dimensions | {64, 64, 128, 256} |
| | Kernel sizes | {(1, 5), (75, 1), (1, 5), (1, 5)} |
| | Strides | {(1, 1), (1, 1), (1, 1), (1, 1)} |
| | Paddings | {(0, 2), (0, 0), (0, 2), (0, 2)} |
| Transformer(Encoder) | FFN Hidden Size | 512 |
| | Head Number | 8 |
| | Token Dimension Size | 256 |
| Decoder | Layers | 4 |
| | Hidden dimension | 384 |
| | Attention Heads | 6 |
| | Feed-forward dimension | 1536 |
| Connector | Input dimension | 256 |
| | Output dimension | 384 |
| | Activation Function | GELU |

### A.1.6 ATTENTION MONITORING

This task aims to distinguish between states of attention and inattention using EEG signals.

- **Attention**: This dataset was collected from **26 subjects** performing a Discrimination/Selection Response (DSR) task to assess cognitive attention. Each subject participated in **three sessions**, with each session consisting of alternating **40-second attention periods** and **20-second rest periods**. To create a balanced binary classification problem (attention vs. inattention), the first **20 seconds** of each attention period were used. The data was then segmented into **4-second windows** with no overlap.

### A.1.7 SLEEP STAGING

This task aims to automatically classify a subject's sleep stage by analyzing their EEG signals. The objective is to assign labels such as Wake, REM, and non-REM (N1, N2, N3) to sequential epochs of EEG data, which is essential for analyzing sleep patterns and quality.

- **ISRUC S1**: This dataset is a subset of the ISRUC-Sleep collection, designed for sleep stage classification. It contains polysomnographic (PSG) recordings from **100 subjects**, including both healthy individuals and patients with sleep disorders. Each recording was visually scored by two human experts, providing labels for different sleep stages. The data includes various electrophysiological signals crucial for sleep analysis.

- **ISRUC S3**: Derived from the same ISRUC-Sleep collection, this dataset shares the same data acquisition and scoring protocols as ISRUC S1. It comprises recordings from **10 subjects**, maintaining consistent signal types and label standards.

### A.1.8 SEIZURE DETECTION

This task focuses on identifying seizure events from long-term scalp EEG recordings. The goal is to distinguish ictal segments from interictal background activity, which is essential for epilepsy diagnosis and continuous monitoring in clinical and home settings.

- **CHB-MIT**: This dataset is part of the PhysioNet EEG collections and contains pediatric scalp EEG recordings from 22 subjects monitored at Boston Children's Hospital. Each subject includes multiple continuous sessions sampled at 256 Hz using the international 10–20 system, with approximately 23 to 26 EEG channels. A total of 198 seizure events are annotated by clinical experts, providing detailed temporal boundaries for ictal activity. The dataset is widely used for benchmarking seizure detection and event prediction models.

Table 6: Hyperparameters for Training Process.

| Phase | Hyperparameter | Setting |
|---|---|---|
| Warm-up | Epochs | 90 |
| | Batch size | 64 |
| | Dropout | 0.2 |
| | Optimizer | Adam |
| | Learning rate | 5e-5 |
| | Adam $\beta$ | (0.9, 0.999) |
| | Adam $\epsilon$ | 1e-8 |
| | Scheduler | Custom Cosine Schedule |
| | Minimal learning rate | 1e-6 |
| In-context Training | Epochs | 40 |
| | Batch size | 48 |
| | Dropout | 0.1 |
| | Optimizer | Adam |
| | Learning rate | 5e-5 (Decoder/Connector), 5e-6 (Encoder) |
| | Adam $\beta$ | (0.9, 0.999) |
| | Adam $\epsilon$ | 1e-8 |
| | Scheduler | Custom Cosine Decay (via LambdaLR) |
| | Cosine cycle epochs | 100 |
| | Minimal learning rate factor | 0.5 |
| | ICL Support Samples (Stage 1) | 8 (fixed) |
| | ICL Support Samples (Stage 2) | Random (0-12) |
| | First Stage epochs | 10 |
| | EEG Sample Length | 30 seconds |

### A.2 BASELINE SELECTION

We introduce the baseline for comparative experiment in this section. Our baseline includes traditional CNN network, Transformer architecture models as well as recent self-supervised Large EEG Models.

**EEGNet** (Lawhern et al., 2018): A compact Convolutional Neural Network that introduced depthwise and separable convolutions to create an efficient architecture for EEG classification. It is designed to generalize effectively across diverse BCI paradigms, demonstrating robust performance even with limited training data.

**BIOT** (Yang et al., 2023): A Transformer architecture engineered for robust cross-dataset EEG classification. It improves generalization across different subjects and recording settings by employing contrastive learning and a domain-invariant attention mechanism to mitigate domain shift effects.

**LaBraM** (Jiang et al., 2024b): A scalable Transformer framework for learning general-purpose EEG representations from extensive datasets. It is pretrained on a diverse collection of recordings to capture features that are broadly applicable to downstream BCI tasks, utilizing efficient self-attention and task-specific adapters to facilitate fine-tuning.

**EEGPT** (Wang et al., 2024a): Utilizes a dual self-supervised pretraining approach that combines masked autoencoding with spatio-temporal representation alignment. Its hierarchical design decouples spatial and temporal feature extraction for greater computational efficiency and adaptability across different BCI applications.

**CBraMod** (Wang et al., 2024b): An EEG foundation model designed to handle the complex dependencies in brain signals. It features a criss-cross Transformer architecture with parallel attention mechanisms that independently model spatial and temporal relationships within the data.

**CodeBrain** (Ma et al., 2025): An efficient two-stage EEG foundation model. It first employs a novel TFDual-Tokenizer to generate discrete representations by independently processing temporal and frequency components. Subsequently, its EEGSSM architecture, which integrates structured global convolutions with a sliding window attention mechanism, is trained via masked prediction to efficiently capture the multi-scale dependencies inherent in brain signals.

### A.3 MODEL SETTING

To ensure the reproducibility of our work, this section provides a complete and detailed specification of our model's architecture. We initialize ECHO's decoder block with (Radford et al., 2022). The following Table 5 enumerates the specific hyperparameter settings for every component of the model pipeline, beginning with the initial EEG sample processing, through the feature extraction and encoding stages (Jiang et al., 2025), and concluding with the Connector and Decoder modules.

### A.4 TRAINING & ENVIRONMENT SETTING

Our model is trained in two distinct phases, each with a unique set of hyperparameters as specified in Table 6. The process begins with an **Encoder Warm-up** phase to stabilize the feature extractor. This is followed by the main **Contextual Training Phase phase**, which itself includes staged settings for in context learning. The table details the optimizer configurations, learning rate schedules, and other crucial settings for both phases. The entire training pipeline was executed in a PyTorch Lightning[1] environment on NVIDIA A100 40G GPUs. For better illustration, here is the pseudo code 1 for the whole training and inference process:

---

[1]https://lightning.ai/pytorch-lightning

---

**Algorithm 1** Contextual Training Phase of ECHO

---

**Input:** query_eeg, support_pairs, query_text (for training only)

1: **function** PREPROCESS(raw_eeg)                              ▷ Unify channels, filter, etc.
2:     **return** processed_eeg

3: **function** ENCODER(processed_eeg)                    ▷ Convert EEG to a sequence of tokens
4:     **return** eeg_tokens

   **Step 1: Encode all EEG samples into a unified context**
5: all_eegs ← [s.eeg for s in support_pairs] + [query_eeg]
6: eeg_tokens ← [encoder(preprocess(eeg)) for eeg in all_eegs]
7: eeg_context ← concat(eeg_tokens)

   **Step 2: Prepare the initial text sequence**
8: support_texts ← [s.text for s in support_pairs]
9: text_sequence ← tokenize(concat(start_token, support_texts, query_token))

   **Step 3: Decoder performs autoregressive prediction**
10: logits ← decoder(input_tokens=text_sequence, cross_attention_context=eeg_context)

   **Step 4: Execute task based on the mode**
11: **if** training **then**                          ▷ Update model parameters by calculating loss
12:     target_tokens ← tokenize(concat(support_texts, query_text, end_token))
13:     loss ← loss_function(logits, target_tokens)
14:     backpropagate(loss)

15: **else** (inference)                     ▷ Obtain the final result via autoregressive generation
16:     result ← autoregressive_generate(logits)
17:     **return** result

---

## B  EXPERIMENT RESULT DETAIL

### B.1  RESULT OF ALL DATASETS

As shown in Table 7, ECHO performs slightly below the strongest baseline, CBraMod, on the SEED-IV dataset. A key factor behind this result lies in the label overlap between SEED-IV and SEED-V, which share the same four categories. Within the multi-task unified training framework, ECHO must first infer which paradigm a given EEG sample belongs to before performing classification. In scenarios with highly overlapping label spaces, the model is prone to misinterpreting SEED-IV samples as belonging to the SEED-V label set, thereby introducing classification errors. Nevertheless, it is worth noting that ECHO still achieves performance comparable to, or in some cases better than, other baselines. For instance, ECHO reaches near-best results in the Weighted F1 score, demonstrating a degree of robustness. This suggests that even under conditions of significant label overlap across tasks, ECHO maintains strong generalization and resilience.

Table 7: Results on the SEED-IV.

| Methods | ACC-B | Kappa | F1-Weighted |
|---|---|---|---|
| EEGNet | 0.3684 ± 0.0312 | 0.1945 ± 0.0258 | 0.3251 ± 0.0390 |
| BIOT | 0.3165 ± 0.0388 | 0.1623 ± 0.0183 | 0.3255 ± 0.0371 |
| EEGPT | 0.3520 ± 0.0437 | 0.1322 ± 0.0254 | 0.3154 ± 0.0220 |
| LaBraM | 0.2647 ± 0.0219 | 0.1652 ± 0.0308 | 0.3572 ± 0.0243 |
| CBraMod | **0.4146** ± 0.0228 | **0.2088** ± 0.0344 | **0.3744** ± 0.0454 |
| CodeBrain | 0.3641 ± 0.0328 | 0.1685 ± 0.0300 | 0.3341 ± 0.0249 |
| ECHO | 0.3747 ± 0.0121 | 0.1595 ± 0.0029 | 0.3601 ± 0.0037 |

Note: **Bold** indicates the best performance. Cyan highlight marks ECHO.

As shown in Table 8, ECHO achieves the overall best performance on the SEED-V dataset, outperforming all baselines across ACC-B, Kappa, and F1-Weighted. Compared to SEED-IV, SEED-V has less overlap in label space with other tasks, which reduces ambiguity in paradigm identification and allows the model to better exploit its unified modeling capacity. In this setting, ECHO avoids the classification errors caused by label interference and demonstrates strong ability to capture task-specific representations under the multi-task framework. These results indicate that when task paradigms are more clearly separated, ECHO can fully realize its potential and consistently surpass state-of-the-art baselines.

Table 8: Results on the SEED-V.

| Methods | ACC-B | Kappa | F1-Weighted |
|---|---|---|---|
| EEGNet | 0.2413 ± 0.0021 | 0.0592 ± 0.0054 | 0.2317 ± 0.0017 |
| BIOT | 0.2245 ± 0.0061 | 0.0432 ± 0.0010 | 0.2153 ± 0.0038 |
| EEGPT | 0.2202 ± 0.0044 | 0.0496 ± 0.0036 | 0.2301 ± 0.0096 |
| LaBraM | 0.2372 ± 0.0053 | 0.0562 ± 0.0028 | 0.2237 ± 0.0078 |
| CBraMod | 0.2432 ± 0.0046 | 0.0586 ± 0.0059 | 0.2452 ± 0.0043 |
| CodeBrain | 0.2447 ± 0.0044 | 0.0610 ± 0.0032 | 0.2411 ± 0.0037 |
| ECHO | **0.2484** ± 0.0021 | **0.0640** ± 0.0008 | **0.2456** ± 0.0010 |

Note: **Bold** indicates the best performance. Cyan highlight marks ECHO.

As shown in Table 9, ECHO achieves performance comparable to the strongest baselines, while obtaining the best result on the F1-Weighted metric. This indicates that, in motor imagery tasks, ECHO demonstrates an advantage in capturing discriminative features under class imbalance. However, in terms of Balanced Accuracy and Cohen's Kappa, ECHO falls slightly behind CBraMod, suggesting that distinguishing between complex categories such as left–right hand and upper–lower limb imagery remains challenging under the cross-subject multi-task setting. Overall, ECHO maintains competitive performance and shows robustness on metrics emphasizing intra-class consistency, underscoring its adaptability to motor imagery EEG within the Seq2Seq framework.

Table 9: Results on the BCIC-IV-2a.

| Methods | ACC-B | Kappa | F1-Weighted |
|---------|-------|-------|-------------|
| EEGNet | 0.4583 ± 0.0281 | 0.2937 ± 0.0612 | 0.4265 ± 0.0498 |
| BIOT | 0.4421 ± 0.0415 | 0.2768 ± 0.0387 | 0.4180 ± 0.0634 |
| EEGPT | 0.4676 ± 0.0304 | 0.2889 ± 0.0529 | 0.4312 ± 0.0391 |
| LaBraM | 0.4538 ± 0.0468 | 0.3011 ± 0.0432 | 0.4147 ± 0.0587 |
| CBraMod | **0.4816** ± 0.0355 | **0.3088** ± 0.0473 | 0.4571 ± 0.0543 |
| CodeBrain | 0.4721 ± 0.0341 | 0.2984 ± 0.0471 | 0.4478 ± 0.0480 |
| ECHO | 0.4763 ± 0.0011 | 0.3015 ± 0.0012 | **0.4632** ± 0.0002 |

Note: **Bold** indicates the best performance. Cyan highlight marks ECHO.

On the Stieger2021-UD dataset (Table 10), ECHO delivers performance largely comparable to other strong baselines. Specifically, it achieves a Balanced Accuracy of 0.7311, which is close to LaBraM and CodeBrain but still falls short of the best-performing CBraMod. For ROC AUC and PR AUC, ECHO does not surpass CBraMod; however, it maintains stable results, with a PR AUC of 0.8258 that is competitive with the top baseline. These findings indicate that while ECHO preserves cross-subject generalization in upper- and lower-limb motor imagery tasks, its discriminative capacity is somewhat constrained under the more challenging multi-task setting.

Table 10: Results on the Stieger2021-UD.

| Methods | ACC-B | ROC AUC | PR AUC |
|---------|-------|---------|--------|
| EEGNet | 0.6952 ± 0.0125 | 0.8113 ± 0.0068 | 0.7741 ± 0.0097 |
| BIOT | 0.7035 ± 0.0098 | 0.8237 ± 0.0042 | 0.7895 ± 0.0112 |
| EEGPT | 0.7189 ± 0.0153 | 0.8378 ± 0.0056 | 0.8032 ± 0.0075 |
| LaBraM | 0.7274 ± 0.0087 | 0.8421 ± 0.0091 | 0.8126 ± 0.0051 |
| CBraMod | **0.7598** ± 0.0079 | **0.8622** ± 0.0037 | **0.8524** ± 0.0039 |
| CodeBrain | 0.7304 ± 0.0201 | 0.8143 ± 0.0078 | 0.8121 ± 0.0061 |
| ECHO | 0.7311 ± 0.0001 | 0.8242 ± 0.0013 | 0.8258 ± 0.0001 |

Note: **Bold** indicates the best performance. Cyan highlight marks ECHO.

On the TUEV dataset (Table 11), ECHO achieves a Balanced Accuracy of 0.5322, notably outperforming its encoder-only counterpart (ECHO$^{\mathcal{E}}$, 0.4816). While the absolute performance does not match state-of-the-art models like CodeBrain, this comparison reveals a crucial insight: the lightweight encoder inherently limits the model's capacity to resolve complex clinical events. However, the proposed ECHO paradigm successfully uplifts this baseline, demonstrating that the generative pre-training strategy effectively enhances representation quality even when the underlying architecture is constrained.

Similar observations apply to the PhysioNet dataset (Table 12). ECHO achieves a Balanced Accuracy of 0.5667, showing a clear improvement over the encoder-only baseline (ECHO$^{E}$, 0.5253). Although it trails behind specialized methods like CBraMod, the consistent gain over ECHO$^{E}$ validates the efficacy of the methodology. This suggests that while the base encoder struggles with the specific frequency-spatial patterns of this task, the ECHO framework extracts significantly richer features than supervised training alone, proving the value of the paradigm despite architectural limitations.

The extended experimental results in the appendix reveal that ECHO's performance varies across tasks and datasets. For SEED-IV and SEED-V emotion recognition tasks, the substantial label overlap within the SEED family makes it difficult for ECHO to disentangle paradigms and sub-class categories in a multi-task setting, which in turn leads to relatively weaker performance compared to some single-task baselines. Nevertheless, ECHO is still able to achieve results close to or even surpassing the strongest baselines on certain metrics (e.g., F1-Weighted), highlighting its robustness. On the BCIC-IV-2a dataset, ECHO performs comparably to the strongest baseline and achieves

Table 11: Results on the TUEV dataset.

| Methods | ACC-B | Kappa | F1-Weighted |
|---------|-------|-------|-------------|
| EEGNet | 0.3876 ± 0.0143 | 0.3577 ± 0.0155 | 0.6539 ± 0.0120 |
| BIOT | 0.5281 ± 0.0225 | 0.5273 ± 0.0249 | 0.7492 ± 0.0082 |
| EEGPT | 0.6232 ± 0.0114 | 0.6351 ± 0.0134 | 0.8187 ± 0.0063 |
| LaBraM | 0.6409 ± 0.0065 | 0.6637 ± 0.0093 | 0.8312 ± 0.0052 |
| CBraMod | **0.6671** ± 0.0107 | 0.6772 ± 0.0096 | 0.8342 ± 0.0064 |
| CodeBrain | 0.6428 ± 0.0062 | **0.6912** ± 0.0101 | **0.8362** ± 0.0048 |
| ECHO$^{\mathcal{E}}$ | 0.4816 ± 0.0152 | 0.4921 ± 0.0118 | 0.7406 ± 0.0094 |
| ECHO | 0.5322 ± 0.0045 | 0.4973 ± 0.0032 | 0.7442 ± 0.0058 |

Note: **Bold** indicates the best performance. Cyan highlight marks ECHO methods. ECHO$^{\mathcal{E}}$ denotes the encoder-only baseline.

the best score on F1-Weighted, demonstrating its ability to maintain cross-subject generalization in classical motor imagery tasks. For Stieger2021-UD, while ECHO does not surpass the best baseline in terms of Balanced Accuracy and AUC, it delivers stable performance overall, indicating its robustness in more complex upper- and lower-limb motor imagery scenarios. In summary, these extended results suggest that while ECHO may encounter performance bottlenecks in multi-task and cross-dataset contexts with overlapping task labels, it nevertheless exhibits strong robustness and generalization. This further validates the applicability and potential advantages of its decoder-centric, Seq2Seq design for neural representation learning.

## B.2 VALIDATION OF SEQ2SEQ PARADIGM

As shown in Table 13, we systematically compares the performance of two configurations across multiple downstream tasks to validate the performance of the Seq2Seq paradigm:

*No Support*: ECHO performs inference without any samples (ICL = 0, and no task sample), relying solely on sample modeling and the decoder's intrinsic capacity for prediction. *Encoder Only*: A conventional paradigm using only the encoder with a lightweight classification head.

Both configurations share identical data splits and evaluation protocols, differing only in whether decoder-centric Seq2Seq prediction and contextual support are employed. Across 12 datasets and 3 evaluation metrics (36 comparisons in total), *No Support* outperforms *Encoder Only* in 29 out of 36 cases and achieves overall superiority on 10 out of 12 datasets. The only exceptions are SEED-IV and Mumtaz, where *Encoder Only* consistently leads, and Mental Arithmetic, where *Encoder Only* is stronger in Balanced Accuracy but *No Support* surpasses it in ROC AUC and PR AUC. This trend suggests that even without access to support samples, decoder-centric sequential modeling provides significant and stable gains, rather than relying solely on ICL-based retrieval for improvement.

**Representative Comparisons and Quantitative Differences:**

Table 12: Results on the PhysioNet dataset.

| Methods | ACC-B | Kappa | F1-Weighted |
|---------|-------|-------|-------------|
| EEGNet | 0.5814 ± 0.0125 | 0.4468 ± 0.0199 | 0.5796 ± 0.0115 |
| EEGConformer | 0.6049 ± 0.0104 | 0.4736 ± 0.0171 | 0.6062 ± 0.0095 |
| ST-Transformer | 0.6035 ± 0.0081 | 0.4712 ± 0.0199 | 0.6053 ± 0.0075 |
| BIOT | 0.6153 ± 0.0154 | 0.4875 ± 0.0272 | 0.6158 ± 0.0197 |
| LaBraM | 0.6173 ± 0.0122 | 0.4912 ± 0.0192 | 0.6177 ± 0.0141 |
| CBraMod | **0.6417** ± 0.0091 | **0.5222** ± 0.0169 | **0.6427** ± 0.0100 |
| ECHO$^{\mathcal{E}}$ | 0.5253 ± 0.0110 | 0.3619 ± 0.0145 | 0.5177 ± 0.0089 |
| ECHO | 0.5667 ± 0.0121 | 0.4214 ± 0.0185 | 0.5604 ± 0.0109 |

Note: **Bold** indicates the best performance. Cyan highlight marks ECHO methods.

Table 13: Comparison results of No Support and Encoder Only methods on downstream tasks.

| Methods | SEED-IV | | | SEED-V | | | SEED | | |
|---|---|---|---|---|---|---|---|---|---|
| | ACC-B | Kappa | F1-W | ACC-B | Kappa | F1-W | ACC-B | ROC AUC | PR AUC |
| No Support | 0.3398 | 0.1174 | 0.3400 | **0.2353** | **0.0466** | **0.2353** | **0.7407** | **0.8488** | **0.8522** |
| Encoder Only | **0.3740** | **0.1609** | **0.3684** | 0.2223 | 0.0284 | 0.2196 | 0.6548 | 0.7493 | 0.7592 |

| Methods | BCI IV 2a | | | High-Gamma | | | Stieger2021-LR | | |
|---|---|---|---|---|---|---|---|---|---|
| | ACC-B | Kappa | F1-W | ACC-B | ROC AUC | PR AUC | ACC-B | ROC AUC | PR AUC |
| No Support | **0.4627** | **0.2836** | **0.4432** | **0.8438** | **0.9125** | **0.9047** | **0.8534** | **0.9349** | **0.9363** |
| Encoder Only | 0.3406 | 0.1242 | 0.2339 | 0.8039 | 0.8900 | 0.8889 | 0.6123 | 0.8918 | 0.9048 |

| Methods | Stieger2021-UD | | | PhysioNet | | | Mental Arithmetic | | |
|---|---|---|---|---|---|---|---|---|---|
| | ACC-B | ROC AUC | PR AUC | ACC-B | Kappa | F1-W | ACC-B | ROC AUC | PR AUC |
| No Support | **0.6924** | **0.8112** | **0.8117** | **0.5437** | **0.3918** | **0.5318** | 0.5442 | **0.6896** | **0.6897** |
| Encoder Only | 0.6058 | 0.7759 | 0.7858 | 0.5253 | 0.3619 | 0.5177 | **0.6008** | 0.6555 | 0.6416 |

| Methods | Mumtaz | | | TUEV (Events) | | | Attention | | |
|---|---|---|---|---|---|---|---|---|---|
| | ACC-B | ROC AUC | PR AUC | ACC-B | Kappa | F1-W | ACC-B | ROC AUC | PR AUC |
| No Support | 0.9056 | 0.9745 | 0.9748 | **0.5214** | **0.5085** | **0.7489** | **0.8056** | **0.8895** | **0.8955** |
| Encoder Only | **0.9698** | **0.9953** | **0.9952** | 0.4816 | 0.4921 | 0.7406 | 0.6472 | 0.7329 | 0.7367 |

Note: **Bold** indicates the best performance between the two methods for each metric.

**Motor Imagery:** On datasets such as BCI IV 2a, Stieger2021-LR/UD, and SEED, *No Support* achieves consistent superiority across all metrics. For example, in BCI IV 2a, Balanced Accuracy rises from 0.3406 (*Encoder Only*) to 0.4627 (*No Support*), a substantial improvement; in Stieger2021-LR, the gains in ROC AUC (0.9349 vs. 0.8918) and PR AUC (0.9363 vs. 0.9048) are particularly notable, underscoring stronger ranking ability and robustness to class imbalance.

**Event Detection:** On TUEV (Events), *No Support* consistently outperforms *Encoder Only* across all metrics, indicating that sequential decoding is particularly effective for modeling context dependencies in event-based labeling. Notably, even though the lightweight encoder performs poorly on this dataset—near random—the seq2seq paradigm itself remains effective, demonstrating that the decoding framework can compensate for weak encoder representations.

**Clinical Depression:** On the Mumtaz dataset, *Encoder Only* clearly dominates across all metrics (e.g., balanced accuracy 0.9698 vs. 0.9056), suggesting that in highly homogeneous, binary clinical datasets with relatively sharp decision boundaries, an encoder-classifier paradigm tailored to the task can more easily reach performance ceilings.

**Emotion Recognition:** On the SEED family dataset, *No Support* demonstrates clear advantages on SEED and remains competitive on SEED-V, yet it is surpassed by *Encoder Only* on SEED-IV. This discrepancy is likely due to overlapping label spaces and paradigms within the SEED family, which introduce ambiguity in paradigm determination. Without support samples, the decoder must simultaneously infer the task paradigm and predict labels; the heavy overlap between SEED-IV and SEED-V can induce "paradigm boundary confusion," weakening the advantage of *No Support*.

**Workload Assessment:** On Mental Arithmetic, *No Support* performs better on AUC metrics (0.6896/0.6897 vs. 0.6555/0.6416), but lags slightly in Balanced Accuracy (0.5442 vs. 0.6008). This reflects a divergence between ranking quality and thresholded accuracy: while *No Support* offers better probability calibration and ranking, its fixed-threshold accuracy does not dominate. With post-processing or threshold tuning, the gap in Balanced Accuracy may be further reduced.

In most datasets and evaluation metrics, *No Support* significantly outperforms *Encoder Only*. The underlying reason is that ECHO's sequence-to-sequence decoding paradigm enables it to jointly model the hierarchical relationships among signals, tasks, and labels within a unified symbolic space. Even without support samples, ECHO can rely on the pattern-matching mechanisms acquired during training to perform cross-task and cross-paradigm reasoning. This not only enhances the model's

robustness but also allows it to maintain strong generalization performance in scenarios where task boundaries are ambiguous or data distributions differ.

By contrast, *Encoder Only* relies on a conventional discriminative classification head, which is more suitable for single-paradigm or structurally simpler tasks but struggles with generalization and flexibility in complex, heterogeneous multi-task settings. Thus, ECHO's superior performance demonstrates that it can autonomously infer task paradigms and predict labels without task-specific prompts while effectively integrating knowledge from diverse datasets through unified modeling. This capability is a key piece of evidence for ECHO's success, showing that it overcomes the limitations of traditional LEMs and achieves stronger adaptability and generalization in multi-task and cross-dataset scenarios.

### B.3 ZERO-SHOT PERFORMANCE

As shown in Table 14, we evaluated the model performance on the SEED-IV dataset. It is noted that ECHO$^{\mathcal{E}}$ and the No Support variant included this dataset during the training phase. Conversely, ECHO w/o SEED-IV operated under a zero-shot setting where SEED-IV was excluded from the training corpus. Experimental results indicate that while ECHO w/o SEED-IV yields slightly lower metrics compared to CBraMod, the performance gap remains marginal. This suggests that the model retains competitive capability despite the absence of domain-specific training data, reflecting positive generalization potential.

Table 14: Results on SEED-IV Dataset.(Unseen for ECHO)

| Methods | ACC-B | Kappa | F1-Weighted |
|---|---|---|---|
| CBraMod | **0.4146** ± 0.0228 | **0.2088** ± 0.0344 | **0.3744** ± 0.0454 |
| ECHO$^{\mathcal{E}}$ | 0.3740 ± 0.0152 | 0.1609 ± 0.0085 | 0.3684 ± 0.0110 |
| ECHO (No Support) | 0.3398 ± 0.0134 | 0.1174 ± 0.0062 | 0.3400 ± 0.0125 |
| ECHO w/o SEED-IV | 0.3491 ± 0.0141 | 0.1283 ± 0.0078 | 0.3357 ± 0.0119 |

Note: **Bold** indicates the best performance. Cyan highlight marks ECHO variants.

### B.4 TRAINING LOSS

Figure 4 illustrates the pretraining loss curve for the Contextual Training Phase of our model, ECHO. The loss exhibits a rapid initial convergence during the first few epochs, followed by a gradual and steady decline. A minor spike is observed at the transition between the two stages of this phase. We attribute this transient increase to the shift from a fixed to a variable number of support samples and the introduction of random EEG data. Notably, the magnitude of this spike is minimal, suggesting that the ECHO decoder had already acquired robust sequence prediction capabilities during Stage 1. Therefore, Stage 2 serves to refine this ability, prompting the model to focus more on the nuanced sequential relationships among the EEG samples.

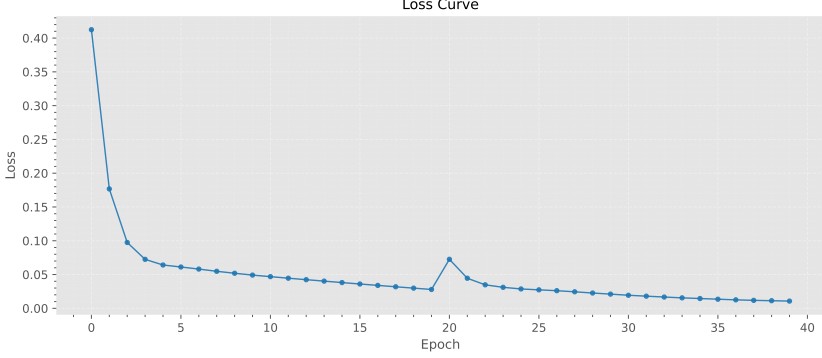

Figure 4: The loss curve of ECHO Contextual Training Phase

## C RELATED WORK DETAIL

Traditional EEG decoding pipelines were primarily based on domain-specific feature extraction methods, such as common spatial patterns (CSP), in combination with shallow classifiers, including linear discriminant analysis (LDA) and support vector machines (SVMs) (Lotte et al., 2007; Guler & Ubeyli, 2007). Although computationally efficient, these approaches were constrained by their reliance on prior assumptions about neural dynamics and exhibited limited generalizability across heterogeneous settings. The introduction of deep learning marked a critical turning point. CNN-based methods enabled the direct learning of spatiotemporal features from raw EEG signals (Zhou et al., 2024; Ding et al., 2024), while RNNs, including LSTMs architectures, provided tools for modeling sequential neural dependencies. Despite their effectiveness in task-specific scenarios, these architectures often exhibited poor transferability, primarily due to variations in electrode configurations, sampling rates, and task paradigms across datasets. This limitation motivated the transition toward LEMs, also referred to as EEG Foundation Models (Zhou et al., 2025a), which aim to learn universal representations from extensive collections of unlabeled EEG data.

Current research on LEMs emphasizes self-supervised pretraining strategies that can be broadly categorized into two classes: reconstruction-based and contrastive-based methods. Reconstruction-based approaches encourage models to learn temporal and spatial dependencies by predicting masked or future signal segments. For instance, EEGPT (Wang et al., 2024a), which employs a dual-masking strategy to reconstruct both raw signals and their spatiotemporal representations, and LaBraM (Jiang et al., 2024b) and CodeBrain (Ma et al., 2025), which tokenize EEG into discrete neural codes and learns by reconstructing masked tokens. Further innovations are seen in models like CSBrain (Zhou et al., 2025b), which uses cross-scale tokenization to handle the multi-resolution characteristics of EEG, and CBraMod (Wang et al., 2024b), which introduces a criss-cross transformer to model complex spatial and temporal dependencies. In contrast, contrastive-based methods enhance robustness and discriminability by constructing positive and negative pairs. Models such as BIOT (Yang et al., 2023) and NeuroGPT (Cui et al., 2024) have explored contrastive objectives alongside masked modeling to improve representation quality and improve generalization between subjects and recording conditions while mitigating domain shift.

While existing LEMs have demonstrated the capacity to produce powerful encoders, their decoding paradigm remains a critical bottleneck. Typically, high-capacity encoders are paired with lightweight classifiers, leading to a mismatch restricting the full exploitation of pretrained representations. This limitation has led to only using LLMs as encoders, where EEG embeddings are aligned with text embeddings and decoded via instruction-based prompting. Models such as NeuroLM (Jiang et al., 2024a) and UniMind (Lu et al., 2025) exemplify this paradigm by unifying EEG and language representations. However, recent work has highlighted that such LLM encoder-centric approaches remain fundamentally limited, as they shift the EEG-to-label mapping into text space without resolving the inductive bias mismatch between static semantic structures in language and the dynamic temporal patterns inherent to EEG signals.

To overcome these limitations, the ECHO framework introduces a decoder-centric Seq2Seq paradigm. Unlike encoder- or LLM-centric methods, ECHO structures both support and target EEG samples as serialized sequences, thereby enabling ICL directly in the EEG modality. This design equips LEMs with the ability to dynamically adapt to new tasks without parameter updates, while preserving task-discriminative capacity and cross-task generalization. By reframing EEG decoding as contextual sequence modeling, ECHO advances the field beyond prior encoder-focused paradigms and provides a principled pathway to unlock the full potential of EEG foundation models.

# D EXTRA EXPERIMENT

## D.1 TASK EXPANSION

### D.1.1 SLEEP STAGING

To evaluate ECHO in long-sequence scenarios, we designed a long-sequence variant and incorporated the ISRUC-S1 sleep staging dataset, which is called ECHO$^{\mathcal{L}}$. Unlike conventional tasks, sleep staging requires classification of complete 30-second segments, placing higher demands on temporal modeling and cross-segment consistency. For the following two reasons, we did not place it as the main result in the main text, but placed it in the appendix for reference as a law and result exploration.

**ECHO$^{\mathcal{L}}$ includes fewer datasets.** In the multi-task training setup, all tasks must be aligned to the 30s window length of ISRUC-S1. For datasets originally segmented into shorter clips (e.g., 1s for emotion recognition), this required padding to 30s, inflating their size by up to 30×. Combined with the additional support samples required for ICL, this drastically increased sequence length and computational overhead. To keep training feasible, the long-sequence version of ECHO was trained only on a reduced set of six datasets (Mumtaz2016, SEED-V, ISRUC-S1, High-Gamma, BCIC-IV-2a, and Mental Arithmetic). As a result, this version is reported in the appendix rather than as the main ECHO model in the paper.

**ECHO$^{\mathcal{L}}$ applies a 30s sleep staging paradigm.** Baseline methods in sleep staging commonly exploit 10-minute temporal context (20×30s consecutive segments), which provides a significant advantage by modeling long-range dependencies. Extending this setup to ECHO, however, would result in nearly 1-hour equivalent sequences per forward pass once the serialized Seq2Seq structure and ICL support tokens are included. With the simplified DeepConvNet encoder and limited computational resources, this was not feasible. Therefore, ECHO was evaluated under a stricter setting, relying solely on the current 30s segment without additional temporal context, making its task substantially harder than that of the baselines (baselines: 10min continuous context; ECHO: 30s + discrete support tokens).

As shown in Table 15, ECHO achieved ACC-B 0.7311, Kappa 0.6878, and F1-Weighted 0.7580. Compared to the strongest baselines CBraMod (ACC-B 0.7865, Kappa 0.7442, F1-Weighted 0.8011) and CodeBrain (Kappa 0.7476, F1-Weighted 0.8020), ECHO lags by only 0.055, 0.056, and 0.043, respectively. Given that ECHO does not benefit from 10min temporal context and simultaneously handles the extra sequence load from ICL, these gaps are both expected and relatively small. Furthermore, ECHO exhibits extremely low variance (e.g., ACC-B ±0.0012, Kappa ±0.0032, F1-Weighted ±0.0022), demonstrating strong training stability and robust inference consistency. Overall, despite operating under much stricter conditions, ECHO delivers performance that is still comparable to state-of-the-art baselines, validating the feasibility of its decoder-centric Seq2Seq + ICL framework for long-sequence sleep staging.

Table 15: Results on the ISRUC-S1.

| Methods | ACC-B | Kappa | F1-Weighted |
|---|---|---|---|
| EEGNet | 0.6238 ± 0.0142 | 0.5921 ± 0.0142 | 0.7032 ± 0.0309 |
| BIOT | 0.7527 ± 0.0121 | 0.7192 ± 0.0231 | 0.7790 ± 0.0146 |
| LaBraM | 0.7633 ± 0.0102 | 0.7231 ± 0.0182 | 0.7810 ± 0.0133 |
| EEGPT | 0.4012 ± 0.0177 | 0.2223 ± 0.0227 | 0.3111 ± 0.0110 |
| CBraMod | **0.7865** ± 0.0110 | 0.7442 ± 0.0152 | 0.8011 ± 0.0099 |
| CodeBrain | 0.7835 ± 0.0033 | **0.7476** ± 0.0040 | **0.8020** ± 0.0018 |
| ECHO$^{\mathcal{L}}$ | 0.7838 ± 0.0012 | 0.7303 ± 0.0032 | 0.7893 ± 0.0022 |

Note: **Bold** indicates the best performance. Cyan highlight marks ECHO.

### D.1.2 MOTOR IMAGERY

In this study, we further designed a task-specialized version of ECHO, denoted as ECHO$^{\mathcal{MI}}$, which was trained exclusively on motor imagery (MI) datasets. The primary motivation was to investigate whether datasets of the same task type are complementary and to assess the contribution of individual datasets within this domain. To this end, we trained a full ECHO$^{\mathcal{MI}}$ model with all MI datasets and a reduced version, ECHO$^{\mathcal{MI}}$ (no KoreaU), in which the KoreaU(Lee et al., 2019) dataset was excluded. By evaluating performance on KoreaU, which was unseen during training in the reduced model, we can examine whether removing one dataset significantly undermines generalization. The results show that although excluding KoreaU leads to a slight drop in performance (e.g., ACC-B decreases from around 64% to 62%), ECHO$^{\mathcal{MI}}$ (no KoreaU) still generalizes effectively to this unseen dataset. This demonstrates that the model captures transferable representations across MI datasets rather than overfitting to any single dataset, thereby highlighting the robustness and cross-dataset generalization capacity of ECHO in task-specialized scenarios.

Table 16: Results on the KoreaU.

| Methods | ACC-B | Kappa | F1-Weighted |
|---|---|---|---|
| ECHO$^{\mathcal{MI}}$ (no KoreaU) | $0.6235 \pm 0.0118$ | $0.4012 \pm 0.0125$ | $0.6154 \pm 0.0142$ |
| ECHO$^{\mathcal{MI}}$ | $\mathbf{0.6412} \pm 0.0123$ | $\mathbf{0.4289} \pm 0.0107$ | $\mathbf{0.6335} \pm 0.0131$ |

Note: Cyan highlight marks MI-specialized versions of ECHO.

### D.1.3 SEIZURE DETECTION

We extend the original ECHO pre-training corpus by incorporating the CHB-MIT epilepsy dataset, yielding the ECHO$^{EP}$ variant. This addition enables the model to acquire characteristic high-frequency and transient seizure patterns prior to downstream evaluation, addressing the limited clinical coverage of the original pre-training setup.

As shown in Table 17, encoder-centric baselines such as CBraMod and CodeBrain achieve the strongest performance (ACC-B and ROC AUC), reflecting the advantage of their large-capacity encoders in modeling epileptic dynamics. In contrast, ECHO$^{EP}$ without support examples underperforms these extensively pretrained models. However, once support samples are provided, ECHO$^{EP}$ exhibits substantial improvements across all metrics (ACC-B +0.053, ROC AUC +0.047, PR AUC +0.111), ultimately achieving the highest PR AUC among all methods. This sharp gain surpasses all encoder-centric SOTA baselines and highlights the effectiveness of the ICL mechanism in leveraging a few support examples to compensate for gaps in seizure-related time–frequency modeling. Overall, although ECHO$^{EP}$ (No Support) trails existing SOTA models in the multi-task pre-training setting, the addition of ICL successfully compensates for the limitations of the lightweight encoder, enabling ECHO$^{EP}$ to deliver competitive performance.

Table 17: Results on the CHB-MIT.

| Methods | ACC-B | ROC AUC | PR AUC |
|---|---|---|---|
| BIOT | $0.7068 \pm 0.0457$ | $0.8761 \pm 0.0284$ | $0.3277 \pm 0.0460$ |
| LaBraM | $0.7075 \pm 0.0358$ | $0.8679 \pm 0.0199$ | $0.3287 \pm 0.0402$ |
| EEGPT | $0.5481 \pm 0.0151$ | $0.8892 \pm 0.0066$ | $0.3073 \pm 0.0641$ |
| CBraMod | $\mathbf{0.7398} \pm 0.0284$ | $0.8892 \pm 0.0154$ | $0.3689 \pm 0.0382$ |
| CodeBrain | $0.7273 \pm 0.0240$ | $\mathbf{0.8961} \pm 0.0174$ | $0.4377 \pm 0.0288$ |
| ECHO$^{EP}$ (No Support) | $0.5671 \pm 0.0078$ | $0.8290 \pm 0.0052$ | $0.3872 \pm 0.0049$ |
| ECHO$^{EP}$ | $0.6199 \pm 0.0040$ | $0.8762 \pm 0.0077$ | $\mathbf{0.4985} \pm 0.0032$ |

Note: **Bold** indicates the best performance. Cyan highlight marks ECHO.

### D.2 COMPONENT VERIFICATION

### D.2.1 CHANNEL FUSION STRATEGIES COMPARISON

The primary goal of the this comparison is to validate the channel-fusion method of ECHO, which employs a simple averaging strategy to handle channel alignment to stress the contribution of the paradigm itself. We compared this approach with two alternative strategies to assess its effectiveness. The first alternative is the Full Channel strategy, which utilizes all available electrodes without reduction. The second is the Channel Deletion strategy, which selects channels (the same with Average method) by priority and deletes directly to eliminate phase misalignment caused by averaging.

Table 18 presents the performance comparison on the KoreaU and High-Gamma datasets. The results indicate that the averaging strategy maintains stable performance. On the High-Gamma dataset, the averaging method outperforms the Full Channel baseline. On the KoreaU dataset, the performance gap between the averaging strategy and the Full Channel approach remains small. These findings verify that the averaging operation is a sufficient and effective design choice for handling channel heterogeneity.

Table 18: Performance comparison of channel fusion strategies on KoreaU and High-Gamma datasets.

| Methods | KoreaU (62 Channels) | | | High-Gamma (128 Channels) | | |
|---|---|---|---|---|---|---|
| | ACC | ROC AUC | PR AUC | ACC | ROC AUC | PR AUC |
| Full Channel | **0.7217** | **0.8138** | **0.8079** | 0.6780 | 0.7956 | **0.8209** |
| Average (Ours) | 0.7031 | 0.7806 | 0.7790 | **0.6911** | **0.8418** | 0.8049 |
| Channel Deletion | 0.6983 | 0.7808 | 0.7759 | 0.6476 | 0.7710 | 0.8134 |

### D.2.2 SUPPORT SIZE SCALING ANALYSIS

To systematically evaluate how ECHO's in-context learning behaves under different support sample sizes, we conduct a support-scaling sensitivity analysis across downstream tasks. Because datasets differ widely in raw performance ranges, we first apply min-max normalization to each dataset's curve, enabling fair comparison in a unified coordinate space. This allows us to focus on the trend itself rather than absolute accuracy.

As shown in Figure 5, from the overall trend (black averaged curve), increasing the number of support samples does not lead to a strictly monotonic improvement. Instead, performance typically peaks around k = 8, after which further increases produce diminishing or even negative returns. This phenomenon is partly attributable to pretraining effects that the decoder may inherently favor shorter sequences and partly reflects a core property of ICL: the model does not always benefit from more examples Zhang et al. (2025). When k becomes too large, cross-subject variability, trial-level fluctuations, and distributional noise can accumulate, making it harder for the model to construct a stable label-space mapping and ultimately causing performance drop-off.

At the same time, we observe a second class of datasets where performance improves steadily as more support samples are added, as shown in Figure 6. These datasets generally have cleaner distributions, higher class separability, and more stable mappings. In such cases, ECHO can exploit additional support samples more effectively, resulting in nearly monotonic gains. This indicates that the usefulness of support samples is highly dependent on the intrinsic characteristics of the task: for well-structured datasets with clear decision boundaries, support examples strongly reinforce label semantics; for noisier or more heterogeneous datasets, too many support samples may instead introduce disruptive variance.

In summary, the optimal number of support samples is not universal but closely tied to dataset complexity. For many EEG tasks, a moderate support size (around k = 8) provides the strongest ICL effect, while several cleaner and more separable datasets benefit from larger support sets.

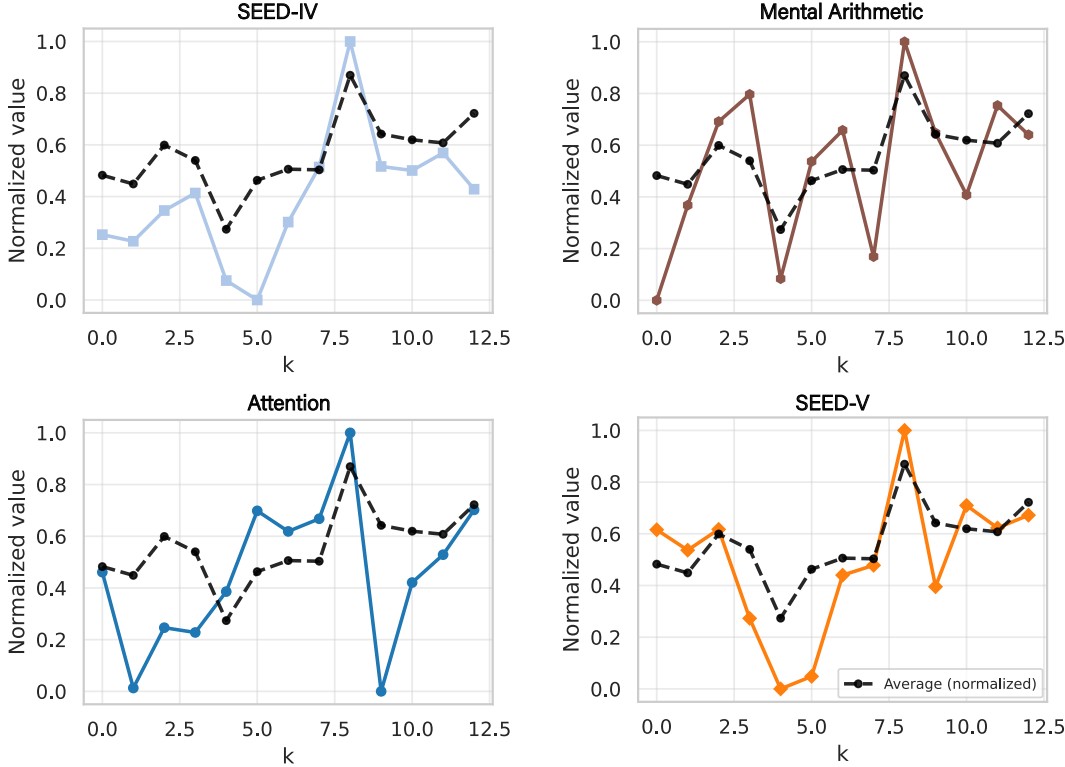

Figure 5: Variation of balanced accuracy with respect to the number of support samples. Each subplot displays the curve for an individual dataset, and the black line indicates the average performance across all datasets. The balanced accuracy values are normalized using min-max scaling to visualize the trends.

Table 19: Results comparison between Seq2Seq paradigm with encoder-centric baselines.

| Method | PhysioNet | | | Mumtaz2016 | | | Stieger2021_LR | | |
|---|---|---|---|---|---|---|---|---|---|
| | ACC-B | Kappa | F1-W | ACC-B | ROC AUC | PR AUC | ACC-B | ROC AUC | PR AUC |
| CBraMod | **0.6257** | **0.5009** | **0.6264** | 0.8946 | **0.9800** | 0.9765 | **0.8424** | **0.9339** | **0.9297** |
| ECHO-CBraMod | 0.5236 | 0.3650 | 0.5234 | **0.9035** | 0.9773 | **0.9809** | 0.8370 | 0.9234 | 0.9241 |
| ECHO$^\mathcal{E}$ | 0.5253 | 0.3619 | 0.5177 | **0.9698** | **0.9953** | **0.9952** | 0.6123 | 0.8918 | 0.9048 |
| ECHO | **0.5667** | **0.4214** | **0.5604** | 0.9056 | 0.9745 | 0.9748 | **0.8534** | **0.9349** | **0.9363** |

| Method | Attention | | | High-Gamma | | | ISURC S3 | | |
|---|---|---|---|---|---|---|---|---|---|
| | ACC-B | ROC AUC | PR AUC | ACC-B | ROC AUC | PR AUC | ACC-B | Kappa | F1-W |
| CBraMod | 0.6478 | 0.7417 | 0.7468 | 0.7478 | 0.8292 | 0.8314 | **0.6784** | **0.5730** | **0.6716** |
| ECHO-CBraMod | **0.7222** | **0.8100** | **0.8056** | **0.7499** | **0.8481** | **0.8363** | 0.6490 | 0.4974 | 0.5914 |
| ECHO$^\mathcal{E}$ | 0.6472 | 0.7329 | 0.7367 | 0.8039 | 0.8900 | 0.8889 | 0.6868 | 0.5950 | 0.6823 |
| ECHO | **0.8194** | **0.8973** | **0.8952** | **0.8552** | **0.9208** | **0.9125** | **0.7283** | **0.6560** | **0.7031** |

Note: **Bold** indicates the best performance between the two methods for each metric. Cyan highlight marks ECHO.

### D.2.3 SEQ2SEQ PARADIGM EXTENSIBILITY VALIDATION

We include an additional set of experiments to further validate the extensibility and robustness of the proposed Seq2Seq paradigm. The goal is to determine whether the performance of ECHO originates from the Seq2Seq and ICL modeling framework itself, rather than from factors such as encoder capacity or task-specific pretraining bias. To this end, we replaced the lightweight encoder in ECHO with a stronger pretrained encoder, CBraMod, and named this hybrid variant ECHO-CBraMod.

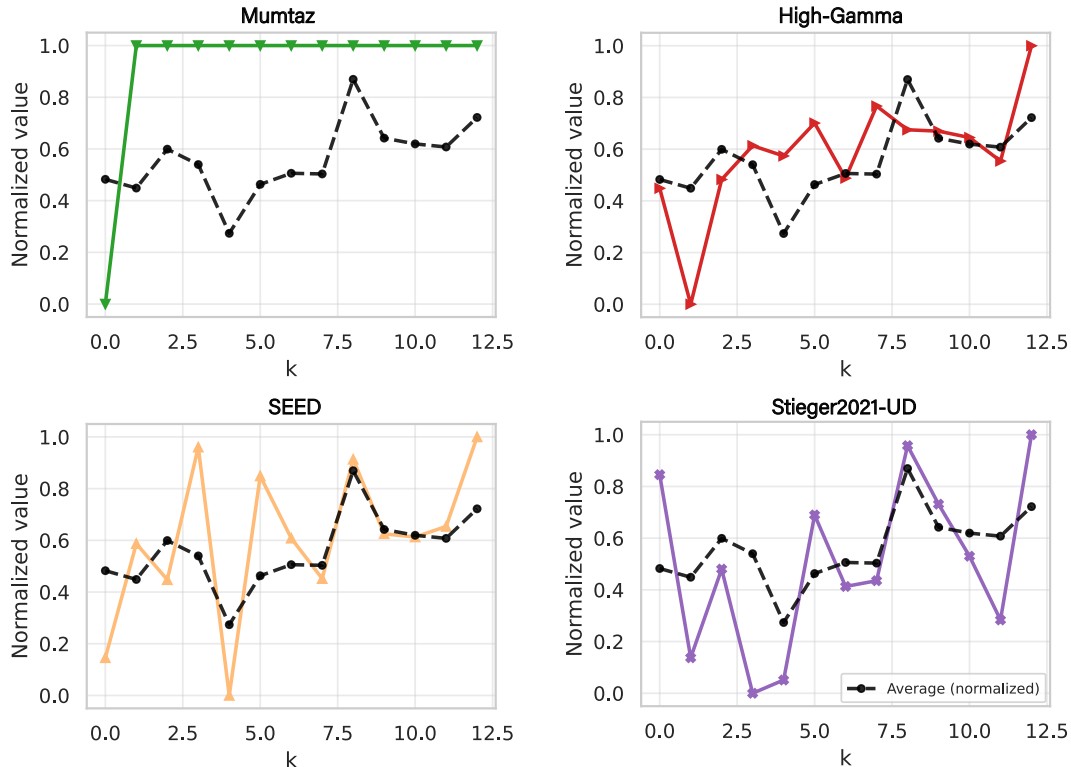

Figure 6: Variation of balanced accuracy with respect to the number of support samples for datasets that benefit monotonically from larger support sets

This configuration allows us to assess whether the Seq2Seq decoding mechanism remains effective when the representational power of the encoder is substantially enhanced. We compare this variant against the original CBraMod baseline to evaluate the contribution of the Seq2Seq paradigm at the framework level.

Results shown in Table 19 indicate the effectiveness of the Seq2Seq paradigm. In the comparison between CBraMod and ECHO-CBraMod, the introduction of the generative decoder leads to performance gains on datasets such as Attention and High-Gamma. Specifically, ECHO-CBraMod achieves higher accuracy and ROC AUC scores on these tasks compared to the encoder-only CBraMod. Furthermore, when observing the native implementation, the full ECHO model consistently outperforms its encoder-only counterpart ECHO$^{\mathcal{E}}$ across the majority of datasets, including PhysioNet, Stieger2021, and ISURC S3. These observations suggest that the proposed generative framework effectively utilizes learned representations and provides positive improvements independent of the specific encoder architecture.

### D.2.4 MULTI-TASK LEANING COMPARISON

To compare the performance of the Seq2Seq paradigm with the standard multi-task paradigm, we equipped the CBraMod with multiple task-specific classification heads, utilizing it as a shared feature extractor for joint training. Under this configuration, the total parameter count of CBraMod-MH is approximately twice that of ECHO with only six datasets. As shown in Table 20, ECHO achieves leading performance on the majority of tasks. Specifically, on the ISRUC S3 and Attention datasets, ECHO maintains stable performance, whereas CBraMod-MH exhibits a substantial decline and predicts randomly. These results suggest that the sequence generation approach adopted by ECHO effectively handles the heterogeneity across tasks, achieving superior generalization with fewer parameters.

Table 20: Results comparison between the Multi-Task baseline and ECHO.

| Method | PhysioNet | | | Mumtaz2016 | | | Stieger2021_LR | | |
|---|---|---|---|---|---|---|---|---|---|
| | ACC-B | Kappa | F1-W | ACC-B | ROC AUC | PR AUC | ACC-B | ROC AUC | PR AUC |
| CBraMod-MH | **0.6340** | **0.5119** | **0.6345** | 0.8542 | 0.9507 | 0.9622 | 0.8374 | 0.9286 | 0.9331 |
| ECHO | 0.5667 | 0.4214 | 0.5604 | **0.9056** | **0.9745** | **0.9748** | **0.8534** | **0.9349** | **0.9363** |

| Method | Attention | | | High-Gamma | | | ISURC S3 | | |
|---|---|---|---|---|---|---|---|---|---|
| | ACC-B | ROC AUC | PR AUC | ACC-B | ROC AUC | PR AUC | ACC-B | Kappa | F1-W |
| CBraMod-MH | 0.5000 | 0.4593 | 0.4654 | 0.7519 | 0.8426 | 0.8375 | 0.2039 | 0.0005 | 0.2004 |
| ECHO | **0.8194** | **0.8973** | **0.8952** | **0.8552** | **0.9208** | **0.9125** | **0.7283** | **0.6560** | **0.7031** |

Note: **Bold** indicates the best performance between the two methods for each metric.

## D.3 OTHER ANALYSIS

### D.3.1 MODEL COMPUTATIONAL COST ANALYSIS

To evaluate model complexity and efficiency, we compare ECHO with baseline methods in terms of parameter size and computational cost, as shown in Table 21. For fairness, the parameter counts of all baseline models include their classification heads. Since classifier dimensions vary across datasets, we report the average classifier size to ensure consistent comparison.

Structurally, ECHO consists of a CNN–Transformer encoder, a fully connected projection layer, and a standard Transformer decoder, resulting in a total of 41.98M parameters. Despite this capacity, ECHO exhibits competitive computational efficiency, requiring only 2.91G MACs and 5.81G FLOPs, which is lower than CodeBrain at 4.37G MACs and EEGPT at 4.89G MACs. Overall, these results show that ECHO reduces inference-time computational overhead while retaining sufficient modeling capacity through its decoder architecture.

Table 21: Model performance comparison on MACs, Parameters, and FLOPs.

| Model | MACs | Params | FLOPs |
|---|---|---|---|
| BENDR | 12.51G | 959.84M | 25.02G |
| BIOT | 0.255G | 3.20M | 0.510G |
| LaBraM | 0.67G | 6.02M | 1.34G |
| CBraMod | 4.21G | 4.03M | 6.29G |
| EEGPT | 4.89G | 25.24M | 9.79G |
| CodeBrain | 4.37G | 15.17M | 8.74G |
| **ECHO** | **2.91G** | **41.98M** | **5.81G** |

Note: The MACs and FLOPs are calculated based on a 1-second input and a single-token output.

### D.3.2 DECODER SIZE ANALYSIS

To systematically evaluate how model architecture and scale influence ECHO's downstream performance, we conducted a comparative study between two decoder configurations: the standard ECHO with 41M parameters and the larger ECHO-Large with 66M parameters. This experiment aims to address three key questions: (1) Does the Seq2Seq paradigm provide clear advantages over a pure encoder design? (2) Do support samples reliably improve cross-subject generalization? (3) Can enlarging the decoder further enhance model performance? As shown in Table 22, we have the following conclusions:

**The Seq2Seq paradigm substantially enhances encoder performance.** Across all datasets, the encoder trained without the Seq2Seq objective (ECHO$^{\mathcal{E}}$) lags notably behind both ECHO and ECHO-Large. This confirms that Seq2Seq training encourages the model to learn richer and more structured internal mappings, resulting in consistently stronger representations.

Table 22: Comparison of ICL capabilities of ECHO molecules of different sizes.

| Methods | SEED-IV | | | SEED-V | | | SEED | | |
|---|---|---|---|---|---|---|---|---|---|
| | ACC-B | Kappa | F1-W | ACC-B | Kappa | F1-W | ACC-B | ROC AUC | PR AUC |
| ECHO$^{\mathcal{E}}$ | 0.3740 | **0.1609** | **0.3684** | 0.2223 | 0.0284 | 0.2196 | 0.6548 | 0.7493 | 0.7592 |
| ECHO (No Support) | 0.3398 | 0.1174 | 0.3400 | 0.2353 | 0.0466 | 0.2353 | 0.7407 | 0.8488 | 0.8522 |
| ECHO | **0.3747** | 0.1595 | 0.3601 | **0.2484** | **0.0640** | 0.2456 | **0.8193** | **0.9020** | **0.8962** |
| ECHO-Large (No Support) | 0.3447 | 0.1234 | 0.3455 | 0.2422 | 0.0545 | 0.2420 | 0.6778 | 0.7874 | 0.7986 |
| ECHO-Large | 0.3647 | 0.1477 | 0.3599 | 0.2474 | 0.0603 | **0.2474** | 0.7583 | 0.8467 | 0.8479 |

| Methods | BCI IV 2a | | | High-Gamma | | | Stieger2021-LR | | |
|---|---|---|---|---|---|---|---|---|---|
| | ACC-B | Kappa | F1-W | ACC-B | ROC AUC | PR AUC | ACC-B | ROC AUC | PR AUC |
| ECHO$^{\mathcal{E}}$ | 0.3406 | 0.1242 | 0.2339 | 0.8039 | 0.8900 | 0.8889 | 0.6123 | 0.8918 | 0.9048 |
| ECHO (No Support) | 0.4627 | 0.2836 | 0.4432 | 0.8438 | 0.9125 | 0.9047 | **0.8534** | **0.9349** | **0.9363** |
| ECHO | **0.4763** | **0.3015** | **0.4632** | 0.8552 | 0.9208 | 0.9125 | **0.8534** | **0.9349** | **0.9363** |
| ECHO-Large (No Support) | 0.3950 | 0.1933 | 0.3838 | 0.8368 | 0.9042 | 0.8956 | 0.8494 | 0.9283 | 0.9293 |
| ECHO-Large | 0.4376 | 0.2499 | 0.4353 | **0.8667** | **0.9292** | **0.9220** | 0.8499 | 0.9322 | 0.9336 |

| Methods | Stieger2021-UD | | | PhysioNet | | | Mental Arithmetic | | |
|---|---|---|---|---|---|---|---|---|---|
| | ACC-B | ROC AUC | PR AUC | ACC-B | Kappa | F1-W | ACC-B | ROC AUC | PR AUC |
| ECHO$^{\mathcal{E}}$ | 0.6058 | 0.7759 | 0.7858 | 0.5253 | 0.3619 | 0.5177 | 0.6008 | 0.6555 | 0.6416 |
| ECHO (No Support) | 0.6924 | 0.8112 | 0.8117 | 0.5437 | 0.3918 | 0.5318 | 0.5442 | 0.6896 | 0.6897 |
| ECHO | **0.7311** | 0.8242 | **0.8258** | 0.5667 | **0.4214** | **0.5604** | **0.6851** | 0.7500 | **0.7530** |
| ECHO-Large (No Support) | 0.6794 | 0.8213 | 0.8187 | 0.5398 | 0.3866 | 0.5319 | 0.5617 | 0.7124 | 0.6886 |
| ECHO-Large | 0.7185 | **0.8259** | 0.8254 | **0.5982** | 0.4039 | 0.5461 | 0.5986 | **0.7796** | 0.7377 |

| Methods | Mumtaz | | | TUEV (Events) | | | Attention | | |
|---|---|---|---|---|---|---|---|---|---|
| | ACC-B | ROC AUC | PR AUC | ACC-B | Kappa | F1-W | ACC-B | ROC AUC | PR AUC |
| ECHO$^{\mathcal{E}}$ | **0.9698** | **0.9953** | **0.9952** | 0.4816 | 0.4921 | 0.7406 | 0.6472 | 0.7329 | 0.7367 |
| ECHO (No Support) | 0.9056 | 0.9745 | 0.9748 | 0.5214 | 0.5085 | **0.7489** | 0.8056 | 0.8895 | **0.8955** |
| ECHO | N/A | N/A | N/A | **0.5322** | **0.5973** | 0.7442 | **0.8194** | **0.8973** | 0.8952 |
| ECHO-Large (No Support) | 0.8813 | 0.9844 | 0.9846 | 0.4626 | 0.4391 | 0.7142 | 0.7962 | 0.8795 | 0.8883 |
| ECHO-Large | N/A | N/A | N/A | 0.5008 | 0.4906 | 0.7398 | 0.8042 | 0.8864 | 0.8879 |

Note: **Bold** indicates the best performance for each
metric per dataset.

**ICL training provides stable and meaningful performance gains.** Comparing each method with its (No Support) counterpart shows that adding support examples yields consistent improvements across almost all datasets. This demonstrates that ICL enables the model to exploit the relationship between support and query samples, thereby improving prediction quality.

**Increasing decoder size does not yield significant performance benefits.** ECHO-Large does not outperform the standard ECHO on most tasks, indicating that ECHO's performance ceiling is not constrained by decoder capacity. Instead, the limiting factor lies in the encoder's EEG representations. Since the encoder remains relatively lightweight, with limited ability to capture inter-subject variability, multi-band structure, and long-range dependencies, simply enlarging the decoder cannot compensate for the encoder's representational bottleneck.

# E  STANDARDIZED CHANNEL SYSTEM

To unify channels settings in different datasets, we use a channel map to map all sorts of channels to standard channels. Details are shown in Table 23.

Table 23: Detail channels mapping of ECHO. **Center** refers to the target channel, and **Included Channels** refers to channels originally from the source datasets.

| Center | Included Channels | Center | Included Channels |
|---|---|---|---|
| A1 | T9, M1, A1 | Fp1 | Fp1, AFp7h, AFp5h, AFp3h, Fp1h, AFp5, AFp3 |
| A2 | T10, M2, A2 | Fp2 | Fp2, AFp6h, AFp8h, Fp2h, AFp4, AFp6 |
| AF3 | AF3, AFF5h, AFF3h, AF5h, AF3h | Fpz | Fpz, AFp1h, AFp2h, AFp1, AFp2 |
| AF4 | AF2, AF4, AFp4h, AFF4h, AF4h, AF6h | FT10 | FT10, FFT10, FTT10 |
| AF7 | AF7, AF5, AFp9h, AFF7h, AF9h, AF7h, AFp9, AFp7, AFF7 | FT7 | FT7, FTT9h, FTT7h, FFT7 |
| AF8 | AF6, AF8, AFp10h, AFF8h, AF8h, AF10h, AFp8, AFp10, AFF8 | FT8 | FT8, FFT10h, FTT8h, FT8h, FT10h, FFT8, FTT8 |
| AFz | AF1, AFz, AFF1h, AF1h, AF2h, AFpz, AFFz | FT9 | FT9, FFT9h, FT9h, FFT9, FTT9 |
| C1 | C1, FCC1h, C3h, CCP1 | Fz | Fz, FFC1h, F1h |
| C2 | C2, CCP2h, C2h, C4h, CCP2 | O1 | O1, POO7h, POO5h, OI1h, O1h, I1h, POO7, POO5, OI1 |
| C3 | C3, FCC3h, C5h, FCC3, CCP3 | O2 | O2, POO6h, POO8h, OI2h, I2h, POO6, POO8, OI2 |
| C4 | C4, C6h, CCP4 | Oz | Oz, Iz, POO1h, POO2h, O2h, POO1, POOz, POO2, OIz |
| C5 | C5, FCC5h, T7h, FCC5, CCP5 | P1 | P1, CPP3h, CPP1, PPO1 |
| C6 | C6, FCC6h, CCP6 | P10 | P10, TPP10h, PPO10 |
| CP1 | CP1, CCP1h | P2 | P2, CPP2h, P2h, CPP2, PPO2 |
| CP2 | CP2, CP2h, CP4h | P3 | P3, P5h, P3h, CPP3, PPO3 |
| CP3 | CP3, CCP3h, CP5h, CP3h | P4 | P4, CPP4h, PPO4h, P4h, P6h, CPP4, PPO4 |
| CP4 | CP4, CCP4h | P5 | P5, CPP5h, P7h, CPP5 |
| CP5 | CP5, CCP5h | P6 | P6, CPP6h, PPO6 |
| CP6 | CP6, CCP6h, CP6h, TP8h, CPP6 | P9 | P9, TPP9h |
| CPz | CPz, CP1h | PO10 | PO10, I2, PPO10h, POO10, CB2 |
| Cz | Cz, C1h, CCPz | PO3 | PO3, PPO3h, POO3h, PO3h, POO3 |
| F1 | F1, FFC3h, AFF1, FFC1 | PO4 | PO2, PO4, POO4h, PO4h, POO4 |
| F10 | AF10, F10, AFF10 | PO5 | PO5, PPO5h, PO7h, PO5h, PPO5 |
| F2 | F2, AFF2h, FFC4h, F2h, F4h, AFF2 | PO6 | PO6, PPO6h, PO6h |
| F3 | F3, F5h, F3h, AFF3, FFC3 | PO7 | PO7, PPO7h, POO9h, PO9h, PPO7 |
| F4 | F4, AFF6h, F6h, AFF4, FFC4 | PO8 | PO8, PPO8h, POO10h, PO8h, PO10h, PPO8 |
| F5 | F5, FFC5h, F7h, AFF5, FFC5 | PO9 | PO9, I1, PPO9h, PPO9, POO9, CB1 |
| F6 | F6, FFC6h, F8h, AFF6, FFC6 | POz | PO1, POz, PPO1h, PPO2h, PO1h, PO2h, PPOz |
| F7 | F7, AFF9h, FFT7h | Pz | Pz, CPP1h, P1h, CPPz |
| F8 | F8, AFF10h, FFT8h, F10h | T3 | T7, T9h, FTT7, T3 |
| F9 | AF9, F9, F9h, AFF9 | T4 | T8, FTT10h, T8h, T10h, TTP8, T4 |
| FC1 | FC1, FCC1 | T5 | P7, TPP7h, P9h, TPP7, T5 |
| FC2 | FC2, FFC2h, FCC4h, FC2h, FC4h, FFC2, FCC2 | T6 | P8, TPP8h, P8h, P10h, T6 |
| FC3 | FC3, FC3h | TP10 | TP10, TTP10h, TTP10, TPP10 |
| FC4 | FC4, FC6h, FCC4 | TP7 | TP7, TTP7h, TP9h, TP7h, TTP7 |
| FC5 | FC5, FT7h, FC5h | TP8 | TP8, TTP8h, TP10h, TPP8 |
| FC6 | FC6, FCC6 | TP9 | TP9, TTP9h, TTP9, TPP9 |
| FCz | FCz, FCC2h, FC1h, FFCz, FCCz | | |

