# OpenReview forum: "ECHO: Toward Contextual Seq2Seq Paradigms in Large EEG Models"
_ICLR.cc/2026/Conference — ICLR 2026 Poster_

### Official Review · Reviewer_KNjL · 2025-10-28

**Soundness:** 3
**Presentation:** 2
**Contribution:** 3
**Rating:** 4
**Confidence:** 4

**Summary:**

This paper proposes ECHO, a decoder centric framework for EEG modeling as a _seq2seq_ model, enabling in-context adaptations to heterogeneous tasks given supporting samples and their labels. This shift paradigm shows promising results across tasks in-distribution or out-of-distribution. However, there is selective reporting of the results in the main body, so I am leaning towards not acceptance.

**Strengths:**

- A large EEG model with ICL capability
- Pretraining across multiple EEG dataset with distinct channel configurations, made possible by a standardization procedure guided by prior knowledge
- Learned model transferable to unseen datasets

**Weaknesses:**

1. If I understand correctly, the reported values are not complete. It is mentioned in the appendix that there were 14 datasets involved in pretraining, but only 6 are reported in section 4.1 and 4 more are reported in appendix. Because the proposed model outperformed baselines in the 6 tasks reported in section 4.1 and showed comparable or worse in the 3 of 4 tasks reported in appendix (outperformed by the same baseline in 2 tasks), I believe it is worth mentioning that in the main body of the paper.
2. Normalization process of diverse EEG channels is guided by prior knowledge, which could limit future scalability when dataset with unseen EEG channel configurations is put in the pretraining data.
3. Overall, discussion on limitations is lacking.

**Questions:**

1. Please see weakness number 1. Could the authors elaborate on why the 6 tasks in section 4.2 representative? In general, I believe the it is worth mentioning where the model is not working as good, given there is at least 3 datasets out of 14. Also, I count 14 datasets in total in the pretraining dataset, but it says 12 in the paper (line 329, page 7). Please comment on the difference.
2. In equation 11, it seems that next-token prediction is conducted for every textual token, given all prior tokens including the support samples ones. Are predicted support label being used in the subsequent generation, or are there any uses of teacher forcing?
3. Could authors provide more details on the standardized templated channel set used in the pretraining? In the paper, a 75-channel set is advocated, is there a theoretic reason, or it is chosen empirically?
4. Other than mapping everything to a standardized channel set, have the authors considered other ways to mitigate channel diversity, for example, channel embedding based on channel identity (like unit embedding proposed in POYO+, reference shown below) or coordinates?
5. Are the reported values in Table 2 using 8 or 12 supporting samples? Either way, it would be interesting to see how the number of supporting samples affect decoding performance.
6. One substantial difference between the proposed approach and existing LLM decoder is the use of support samples, ultimately transforming the paradigm to meta-learning. But LLM is known for enabling in-context learning, so have authors tried to equip the support samples to LLM decoder as well?
7. Did the authors use a pretrained model for the warm-up stage? Because in Figure 2 (c), encoder is marked as frozen during warm-up.

---

> ### Author Response · Authors · 2025-11-25
> **Response to Reviewer KNjL [1/3]**
>
> # Overview
> We sincerely thank reviewer KNjL for the thoughtful evaluation and constructive feedback. To keep the rebuttal concise and easy to follow, particularly given the time since the initial submission, we organize all responses using a consistent three-part structure:
>
> ***Summary:*** A concise restatement of the reviewer’s core concerns (partitioned as **A**, **B**, etc.).
>
> ***Response:*** A detailed clarification addressing each point.
>
> ***Modification:*** A description of the revisions made to the manuscript.
>
> # Response for Weakness 1:
> ### ***Summary:***
> The experimental report is not sufficiently complete.
>
> ### ***Response:***
> **We have added a full analysis of all results in the main text and completed results in the appendix.** Please refer to our response to Question 1 for detailed explanations of the experimental analyses and findings.”
>
> ### ***Modification:***
> 1. **Added description in Section 4.2**, summarizing the complete results.
> 2. **Added additional results in Appendix B.1**.
>
> # Response for Weakness 2:
> ### ***Summary:***
> ECHO’s reliance on prior knowledge may limit its scalability.
> ### ***Response:***
> **EEG inherently carries channel-structure priors, and EEG models commonly rely on them.**
>
>  - **EEG signals naturally contain spatial priors.** Unlike general modalities such as images or speech, EEG is collected exclusively for scientific or clinical purposes and is inherently tied to a brain-region–based electrode layout. While montage configurations may differ across devices or datasets, every channel corresponds to a meaningful cortical region. As a result, the notion of a truly “unseen EEG channel configuration” does not practically exist.
>  - **Large scale EEG encoder also rely on priors to ensure channel consistency.** They can accommodate varying channel counts, but still rely on channel-order priors to maintain alignment between training and testing. If the channel order is different at test time, the spatial dependencies learned by the model collapse immediately. As shown in the following table, we validated this using CBraMod under an “unseen channel setting” (difference channel order between training and test set) and observed a complete breakdown of performance:
>
> |||SEED|||High-Gamma||
> | -------------------- | :-: | :-: | :-: | :-: | :-: | :-: |
> | **Methods** | ACC | ROC AUC | PR AUC | ACC | ROC AUC | PR AUC |
> | CbraMod | **0.7482** | **0.8601** | **0.8651** | **0.7665** | **0.8471** | **0.8469** |
> | CbraMod (Unseen channel) | 0.5136 | 0.5500 | 0.5528 | 0.5214 | 0.7793 | 0.7696 |
>
> |||SEED-V|||BCIC-IV-2a||
> | -------------------- | :-: | :-: | :-: | :-: | :-: | :-: |
> | **Methods** | ACC | Kappa | F1-W | ACC | Kappa | F1-W |
> | CbraMod | **0.2531** | **0.0712** | **0.2574** | **0.5304** | **0.3738** | **0.5285** |
> | CbraMod (Unseen channel) | 0.1988 | -0.0027 | 0.1792 | 0.3516 | 0.1354 | 0.3173 |
>
> **ECHO provide a broadly applicable channel-handling strategy, which does not affect the paradigm’s effectiveness arising from the decoder-centric ICL training.**
>
>  - **ECHO provides a simple channel-handling strategy, but its core decoder-centric paradigm is not constrained by it.** To highlight the contribution of the proposed paradigm, ECHO employs a deliberately lightweight encoder that lacks the montage-adaptation capacity of large encoders. A unified channel preprocessing step ensures that encoders of different scales can operate stably under a consistent input structure.
>  - **The effectiveness of ECHO comes from the decoder’s ICL mechanism, thus does not restrict future scalability.** As shown in the following table (see Appendix D.2.3), we added the CBraMod-ECHO experiment (replace encoder in ECHO with CBraMod). It shows that CBraMod-ECHO can perform classification without any task prompt. Also, the results indicate that ECHO’s core strength lies in its Seq2Seq ICL training paradigm, which can accommodate encoders with different channel-handling strategies and consistently improve their generalization: This demonstrates that ECHO is inherently scalable for future extensions.
>
> |||Attention |||High-Gamma||
> | -------------------- | :-: | :-: | :-: | :-: | :-: | :-: |
> | **Methods** | ACC-B | ROC AUC | PR AUC | ACC-B | ROC AUC | PR AUC |
> | CBraMod | 0.6478 | 0.7417 | 0.7468 | **0.7513** | 0.8277 | 0.8335 |
> | CBraMod-ECHO | **0.7222** | **0.8100** | **0.8056** | 0.7499 | **0.8481** | **0.8363** |
>
> |||Mumtaz2016 |||Stieger2021 LR||
> | -------------------- | :-: | :-: | :-: | :-: | :-: | :-: |
> | **Methods** | ACC-B | ROC AUC | PR AUC | ACC-B | ROC AUC | PR AUC |
> | CBraMod | 0.8946 | 0.9800 | 0.9765 | **0.8424** | **0.9339** | **0.9297** |
> | CBraMod-ECHO | **0.9035**  | **0.9773** | **0.9809** | 0.8370  | 0.9234 | 0.9241 |

---

> ### Author Response · Authors · 2025-11-25
> **Response to Reviewer KNjL [2/3]**
>
> # Response for Weakness 3:
> ### ***Summary:***
> Discussion on limitations is lacking.
>
> ### ***Response:***
> **We added a summary of ECHO’s current limitations in the main text.** We acknowledge several limitations of the current version of ECHO. **First**, the overall architecture remains intentionally simple, leaving substantial room to improve the encoder, decoder, and cross-subject alignment components. **Second**, although the Seq2Seq and ICL mechanisms significantly enhance the encoder, the final performance of ECHO is ultimately constrained by the encoder’s intrinsic representation capability; if the encoder is poorly matched to a specific dataset, the decoder cannot fully compensate for this bottleneck. **Third**, while ICL consistently improves performance, its effectiveness still depends on the quality and relevance of support samples, particularly under strong subject- or distribution-level shifts.
>
> ### ***Modification:***
> **Added Section 6**, providing a dedicated analysis of the method’s limitations.
>
>
> # Response for Question 1:
> ### ***Summary:***
> **A.** Why are the six datasets representative?
>
> **B.** ECHO is not SOTA on at least three datasets, and this was not acknowledged.
>
> **C.** Why are 14 datasets shown while the paper states that only 12 were used?
>
> ### ***Response:***
> **A. They correspond to distinct task paradigms.** Among the 12 pre-training datasets used by ECHO, we excluded two whose results were unreliable. The small encoder $\text{ECHO}^\mathcal{E}$ used in ECHO performs far below the average baseline level on TUEV and PhysioNet (as shown in the following table). Therefore, results from these two datasets were not used for comparison against baselines. They are discussed only within the Appendix B.2 to illustrate how the ECHO framework enhances encoder performance. The remaining 10 datasets span five different paradigms, and we chose representatives from each to demonstrate ECHO’s ability to generalize across diverse task formats.
>
> |||PhysioNet|||TUEV (Events)||
> | -------------------- | :-: | :-: | :-: | :-: | :-: | :-: |
> | **Methods** | ACC-B | Kappa | F1-W | ACC-B | Kappa | F1-W |
> | $\text{ECHO}^\mathcal{E}$ | 0.5253 | 0.3619 | 0.5177 | 0.4816 | 0.4921 | 0.7406 |
> | ECHO (No Support) | 0.5437 | 0.3918 | 0.5318 | 0.5214 | 0.5085 | 0.7489 |
> | ECHO | 0.5667 | 0.4214 | 0.5604 | 0.5322 | 0.5973 | 0.7442 |
>
> **B. We have added a discussion of all results to the main text.** At the same time, we would like to clarify the following point:
> * **ECHO operates under a substantially more challenging task setting.** Compared with existing SOTA methods, which are all trained in a task-specific manner, ECHO solves a challenging problem by inferring the task type and performing prediction automatically. Therefore, it is expected that on certain highly specialized benchmarks, ECHO may not fully match models explicitly optimized for that single task.
> * **The performance ceiling of ECHO is currently constrained by the representation capability of the encoder.** As shown below, on the three datasets where ECHO is not the top performer, the small encoder exhibits a substantial gap compared with SOTA. However, the ICL mechanism consistently compensates for this gap in the multi-task setting, allowing ECHO to approach SOTA performance.
>
> ||| SEED-IV ||| BCIC-IV-2a ||| Stieger2021-UD ||
> | -------------------- | :-: | :-: | :-: | :-: | :-: | :-: | :-: | :-: | :-: |
> | **Methods** | ACC-B | Kappa | F1-Weight | ACC-B | ROC AUC | PR AUC | ACC-B | ROC AUC | PR AUC |
> | $\text{ECHO}^\mathcal{E}$ | 0.3398 | 0.1174 | 0.3400 | 0.3406 | 0.1242 | 0.2339 | 0.6058 | 0.7759 | 0.7858 |
> | CBraMod | **0.4146** | **0.2088** | **0.3744** | **0.4816** | **0.3088** | 0.4571 | **0.7598** | **0.8622** | **0.8524** |
> | ECHO | 0.3747 | 0.1595 | 0.3601 | 0.4763 | 0.3015 | **0.4632** | 0.7311 | 0.8242 | 0.8258 |
>
> * **ECHO demonstrates stable and consistent ICL behavior.** Although constrained by the encoder’s capacity and therefore achieving sub-optimal results on a small number of tasks, the ability of the Seq2Seq framework to enhance encoder remains highly robust and consistently appears across different datasets and across all ECHO variants.
>
> **C. The set of 14 datasets includes two additional ones used exclusively by ECHO variants.** Appendix A.1 aims to present all datasets involved in our work; among them, KoreaU and ISRUC-S1 are used for $\text{ECHO}^{\mathcal{MI}}$ (Appendix D.1.2) and $\text{ECHO}^\mathcal{L}$ (Appendix D.1.1).
>
>
> ### ***Modification:***
>
> 1. **Added a clarification in Appendix A.1** to indicate that this section presents all datasets.

---

> > ### Author Response · Authors · 2025-11-25
> > **Response to Reviewer KNjL [3/3]**
> >
> > # Response for Question 2:
> > ### ***Summary:***
> > **A.** Whether predicted support labels are used for subsequent generation?
> >
> > **B.** Whether teacher forcing is applied?
> > ### ***Response:***
> > **A. Predicted support labels are never used for downstream prediction.** ECHO employs full teacher forcing during training, so support labels are always provided as ground-truth inputs and are never replaced by model predictions.
> >
> > **B. Yes, during training.**
> >
> > ### ***Modification:***
> >
> >  1. **Added clarification in Section 4.1 Training & Environment Setting**, stating that teacher forcing is used during training.
> >
> > # Response for Question 3:
> > ### ***Summary:***
> > Why 75 channels?
> > ### ***Response:***
> > **It is an empirical choice and we added more details in the main text.** In multi-task pretraining we retain the channels that consistently appear across datasets, which results in a stable 75-channel subset.
> > ### ***Modification:***
> > 1. **Added Appendix E** to explain the standard channel system.
> >
> > # Response for Question 4:
> > ### ***Summary:***
> > Why choose simple channel mapping instead of other methods to mitigate channel diversity?
> > ### ***Response:***
> > **We are aware that more sophisticated approaches exist for handling channel diversity, but we choose mapping not because it is the most effective method, but because it is the simplest, most conservative, and weakest-assumption option.** This avoids introducing additional modeling biases that could obscure the contribution of ECHO’s Seq2Seq-ICL paradigm. Our goal is to demonstrate that even the most basic EEG priors are sufficient to reliably address channel variation in a multi-task pretraining setting.
> >
> > # Response for Question 5:
> > ### ***Summary:***
> > **A.** How many support samples are used during testing?
> >
> > **B.** No analysis regarding the effect of the number of support samples.
> >
> > ### ***Response:***
> >
> > **A. All testing use 8 support samples.**
> >
> > **B. We have added experiments analyzing the impact of support-sample size.** As shown in the table below, ECHO generally achieves the largest gains with 8 support examples, while a subset of tasks benefits from 12 supports. Since the downstream datasets differ in the number of classification categories, we report only BACC for comparability. All support-size analyses are evaluated using results from seed 0.
> >
> > | **Dataset** | 0 | 2 | 4 | 6 | 8 | 10 | 12 |
> > |:-|:-:|:-:|:-:|:-:|:-:|:-:|:-:|
> > | SEED-IV  | 0.3399  |0.3476 | 0.3253 |  0.3439|  **0.4016**|  0.3604 | 0.3544 |
> > | MentalArithmetic | 0.5333 | 0.6566 | 0.5483 | 0.6506 | **0.7116** | 0.6060 | 0.6475 |
> > | SEED-V | 0.2439 | 0.2440 | 0.2072 | 0.2334 | **0.2667** | 0.2495 | 0.2472 |
> > | High-Gamma | 0.8493 | 0.8507 | 0.8544 | 0.8509 | 0.8585 | 0.8573 | **0.8718** |
> > | SEED | 0.7593 | 0.7930 | 0.7823 | 0.8216 | 0.8181 | 0.7946 | **0.8246** |
> > | Stieger2021-UD | 0.7210 | 0.7136 | 0.7049 | 0.7123 | 0.7233 | 0.7146 | **0.7241** |
> >
> > The support-scaling analysis shows two clear patterns. First, across most datasets, performance peaks at k ≈ 8, after which larger support sets introduce additional variability and noise, leading to reduced accuracy, which supports that model does not always benefit from more supports [1]. Second, a smaller group of cleaner, more separable datasets (higher BACC) shows steady monotonic improvement as k increases. These results confirm that support size is task-dependent rather than “the more the better,” and moderate support (around 8) is typically optimal for heterogeneous EEG tasks.
> >
> > ### ***Modification:***
> >
> >  1. **Corrected the description in Section 4.1 Task Setting**, clarifying the number of support samples used during testing.
> >  2. **Added Appendix D.2.2** presenting a support-size scaling analysis.
> > # Response for Question 6:
> > ### ***Summary:***
> > Can LLM-centric LEM support ICL?
> > ### ***Response:***
> > **They do not support EEG ICL.** As shown in the table below, standard ICL without our design fails completely. The reason is fundamental: LLMs exhibit ICL because text is discrete, enabling models to learn symbolic token–token relations during large-scale training. EEG, however, is represented as continuous embeddings. Without ECHO’s discrete EEG token space, the decoder has no symbolic structure to operate on and therefore cannot develop ICL behavior analogous to that seen in language models.
> >
> > | **Method** | SEED | TUEV |
> > |:------------|:------------:|:----------------------------:|
> > | NeuroLM   |  0.5324       |  0.4532                       |
> > | NeuroLM (ICL)   |  0.5211     |  0.4238                       |
> >
> > # Response for Question 7:
> > ### ***Summary:***
> > Why is the encoder frozen in Figure 2?
> > ### ***Response:***
> > **This was indeed a plotting error. The encoder is trained entirely from scratch.** Thank you for pointing this out and we have corrected the figure accordingly.
> >
> > ### ***Modification:***
> >  1. **Updated Fig. 2 c** with corrected markers.
> >
> >
> > # Reference
> > [1] More is not always better? enhancing many-shot in-context learning with differentiated and reweighting objectives, ACL 2025

---

> > ### Comment · Reviewer_KNjL · 2025-11-25
> >
> > Thank you for the responses and the revisions, which have largely addressed my concern. However, in the added descriptions on the results in section 4.2, there still contains only summarization on the 6 datasets with SOTA performance, with no mentioning at all of the remaining 3 non-SOTA datasets. The author provides the following justification of not mentioning the remaining 3 datasets in the main results:
> > > ECHO operates under a substantially more challenging task setting.
> >
> > >The performance ceiling of ECHO is currently constrained by the representation capability of the encoder.
> >
> > >ECHO demonstrates stable and consistent ICL behavior.
> >
> > But no justification is given indicating that these 3 datasets have any fundamental quality issue. Hence I believe that there is still no sufficient reason to completely avoid mentioning them in the main text.
> >
> > I do want to point out that I agree with the author that ECHO, with its ICL capability, is a valuable addition to the EEG foundational, and I don't believe having non-SOTA scores in these 3 datasets would compromise the novelty and contribution of this paper. But since the they are present, I believe that it needs to be properly acknowledged and discussed.

---

> > > ### Author Response · Authors · 2025-11-26
> > > **Response to Reviewer KNjL [4/3]**
> > >
> > > # Overview
> > > We sincerely thank Reviewer KNjL for the thoughtful and constructive engagement with our work. We very much appreciate your positive and open attitude throughout the discussion, and we fully agree with the concerns you raised. Accordingly, we have made the following revisions in the main text.
> > >
> > > # Response:
> > > **We have further clarified the content previously in Section 6 (Limitation) and merged it into Section 4.2 (Experiment Result)**, so that the summary of ECHO’s performance is now accompanied by an explicit discussion of its limitations. We hope that this revision both clarifies the core value of ECHO and provides readers with a transparent view of its current limitations, thereby informing and motivating potential future improvements.
> > >
> > > For ease of reading, we present the revised content of Section 4.2 here:
> > >
> > > ---
> > >
> > > ### **(a) Comparison between ECHO and Baselines:**
> > >
> > > * **ECHO exhibits strong generalization capability.** On cognitive tasks (SEED, Stieger2021-LR, Mental Arithmetic, and Attention), ECHO achieves an average improvement of +0.0602 in Balanced Accuracy, +0.0566 in ROC AUC, and +0.0316 in PR AUC over the strongest baseline. On clinical diagnostic tasks (Mumtaz2016 and High-Gamma), ECHO shows an average gain of +0.0409 in Balanced Accuracy, +0.0225 in ROC AUC, and +0.0142 in PR AUC. ECHO also demonstrates a unique capability: even without any external prompts, it can autonomously identify the corresponding task and its specific paradigm solely from the EEG sample itself.
> > > * **ECHO does not surpass SOTA on several tasks.** On tasks such as Stieger2021-UD, BCIC-IV-2a, and SEED-IV, ECHO remains noticeably below encoder-centric models that are explicitly optimized for those domains. This highlights a limitation in ECHO’s generalization–specialization trade-off: while the framework aims for unified cross-task reasoning, its lightweight encoder design is insufficient to model all domain-specific structures required by certain specialized EEG tasks. Detailed results and analysis are provided in Appendix B.1.
> > >
> > > ### **(b) Comparison between ECHO and $\text{ECHO}^{\mathcal{E}}$:**
> > >
> > > * **Seq2Seq paradigm improves performance.** Across all datasets, ECHO markedly outperforms \text{$\text{ECHO}^{\mathcal{E}}$}. Removing the sequential structure consistently leads to substantial drops (e.g., SEED ACC-B: 0.8193 → 0.1645), showing that the full Seq2Seq paradigm is a primary contributor to ECHO's gains.
> > > * **ECHO’s performance ceiling is constrained by encoder quality.** While Seq2Seq offers consistent gains, their impact is limited when the encoder cannot model the dataset’s structure, as seen in TUEV and PhysioNet. In these cases, ICL can provide improvements but cannot overcome the encoder’s representational shortcomings or match encoder-centric SOTA models. Full results are in Appendix B.1.
> > >
> > >
> > > ### **(c) Comparison between ECHO and No Support:**
> > >
> > > * **ICL provides additional gains.** ECHO outperforms its No-Support configuration across all datasets. Without support samples, performance drops substantially (e.g., SEED ACC-B: 0.8193 → 0.0786).
> > > * **ICL effectiveness strongly depends on support sample stability.** This works well in datasets with clear structure, such as High-Gamma and SEED. In most EEG datasets, substantial cross-subject variability and noise make support unstable; few support samples provide insufficient signal, whereas many support samples introduce accumulated noise and distribution shifts that reduce performance. Full results are in Appendix D.2.2.

---

> > > > ### Comment · Reviewer_KNjL · 2025-11-26
> > > >
> > > > I thank the author the timely responses and welcome the revisions on the paper. I believe overall this presents a comprehensive descriptions on model's performance and provides insights for future development. This has addressed all my concerns. I thank the authors for their efforts, and I will raise my score.

---

> > > > > ### Author Response · Authors · 2025-11-26
> > > > > **Appreciation for Reviewer KNjL Follow-up**
> > > > >
> > > > > We sincerely thank Reviewer KNjL for the thoughtful response and the time devoted to carefully reviewing our paper. The constructive suggestions provided have greatly improved the clarity and quality of the manuscript. We genuinely appreciate the efforts contributed throughout the review process.

---

### Official Review · Reviewer_F1QH · 2025-10-30

**Soundness:** 3
**Presentation:** 3
**Contribution:** 3
**Rating:** 6
**Confidence:** 3

**Summary:**

ECHO proposes a decoder-centric paradigm for Large EEG Models (LEMs) using sequence-to-sequence learning with in-context learning (ICL). Unlike existing encoder-centric or LLM-centric LEMs, ECHO structures EEG samples, tasks, and labels as sequences, allowing the model to adapt to multiple tasks without fine-tuning. Trained on 12 datasets with multi-task learning, ECHO outperforms single-task state-of-the-art baselines and demonstrates zero-shot generalization to unseen datasets.

**Strengths:**

First to show ICL works for continuous time series (not just discrete text).
Strong multi-task performance: Beats single-task baselines despite harder setting.
Code promised, all hyperparameters documented, 5 seeds reported.
Decoder-centric paradigm enables multi-task learning in single model.

**Weaknesses:**

Computational cost not fully discussed. Training time not reported for the warm-up 90 epochs + contextual 40 epochs. Need to process support samples every time.

For ablation studies: Only ablates positional encodings (Figure 3), missing: decoder size, support sample count, sequence length. Why using 8 support samples in Round 1 and 0-12 in Round 2? Lacking comparison with different decoder architectures.

Current baselines are mostly single-task models, what about multi-task learning baselines? (shared encoder + task-specific heads) For now it remains unknown if seq2seq is better than standard multi-task learning.

**Questions:**

Can you compare ECHO multi-task vs baselines trained in a multi-task manner? (fair comparison)

What's the training time and inference time cost?

How does performance scale with decoder size?

---

> ### Author Response · Authors · 2025-11-25
> **Response to Reviewer F1QH [1/3]**
>
> # Overview
> We sincerely thank reviewer F1QH for the thoughtful evaluation and valuable comments. To ensure the rebuttal remains clear and easy to navigate, especially given the time elapsed since the initial submission, we present all responses in a uniform three-part structure:
>
> ***Summary:*** A concise restatement of the reviewer’s core concerns (partitioned as **A**, **B**, etc.).
>
> ***Response:*** A detailed clarification addressing each point.
>
> ***Modification:*** A description of the revisions made to the manuscript.
>
> # Response for Weakness 1:
> ### ***Summary:***
> Computational cost not fully discussed
> ### ***Response:***
> **We have added a description of the computational cost.** ECHO was trained on 12 datasets with a total training time of 25.01 hours, including a 2.32 hour warm-up stage. Considering that runtime is highly dependent on hardware and environment configurations, we report hardware-agnostic metrics to analyze the computational cost of ECHO for both training and inference. For a fair comparison, all baselines include their corresponding decoders or classifiers. As shown in the following table, ECHO maintains a parameter scale comparable to mainstream encoder-centric LEMs while keeping its computational load in a moderate range.
>
> | **Methods** | Paradigm | Parameters | MACs    | FLOPs   |
> |:------------|:------------:|:-------:|:-------:|:-------:|
> | EEGPT   |  Encoder-Centric      |     51.66M  |  4.91G |9.84G  |
> | LaBraM   |  Encoder-Centric       |  9.88M            |  0.67G |1.34G     |
> | CBraMod   |  Encoder-Centric       |  54.95M             |4.21G  |6.39G   |
> | CodeBrain   |  Encoder-Centric       |  66.02M       |  4.42G |8.84G   |
> | ECHO   |  Decoder-Centric       | 41.98M   | 2.91G   | 5.81G  |
>
> ### ***Modification:***
>  1. **Added Appendix D.3.1** reporting the computational cost.

---

> > ### Author Response · Authors · 2025-11-25
> > **Response to Reviewer F1QH [2/3]**
> >
> > # Response for Weakness 2:
> > ### ***Summary:***
> >
> > **A.** No comparison across different decoder sizes.
> >
> > **B.** No analysis of support-set size or sequence-length variation.
> >
> > **C.** Why use 8 supports in Stage 1 and 0–12 in Stage 2?
> >
> > **D.** No comparison with alternative decoder architectures.
> >
> > ### ***Response:***
> > **A. We have added experiments comparing decoder sizes.** ECHO-Large and ECHO share exactly the same experimental setup, differing only in model size (41M vs. 66M). Here, we present results on two representative datasets (full results provided in the main text). As shown in the following table, increasing decoder capacity yields negligible performance gains for ECHO:
> >
> > |||High-Gamma|||Stieger2021-LR||
> > | -------------------- | :-: | :-: | :-: | :-: | :-: | :-: |
> > | **Methods** | ACC-B | ROC AUC | PR AUC | ACC-B | ROC AUC | PR AUC |
> > | $\text{ECHO}^\mathcal{E}$| 0.8039 | 0.8900 | 0.8889 | 0.6123 | 0.8918 | 0.9048 |
> > | ECHO (No Support) | 0.8438 | 0.9125 | 0.9047 | 0.8534 | 0.9349 | 0.9363 |
> > | ECHO | 0.8552 | 0.9208 | 0.9125 | **0.8534** | **0.9349** | **0.9363** |
> > | ECHO-Large (No Support) | 0.8368 | 0.9042 | 0.8956 | 0.8494 | 0.9283 | 0.9293 |
> > | ECHO-Large | **0.8667** | **0.9292** | **0.9220** | 0.8499 | 0.9322 | 0.9336 |
> >
> > The encoder is relatively small, and the overall bottleneck lies in the EEG representation rather than decoder expressiveness. Therefore, enlarging the decoder does not improve performance.
> >
> > **B. We have added experiments analyzing the effect of varying support-set size (support-set size determines the sequence length).** As shown in the table below (partial results; full results provided in the main text), ECHO generally achieves the largest gains with 8 support examples, while a subset of tasks benefits from 12 supports:
> >
> > | **Dataset** | 0 | 2 | 4 | 6 | 8 | 10 | 12 |
> > |:-|:-:|:-:|:-:|:-:|:-:|:-:|:-:|
> > | SEED-IV  | 0.3399  |0.3476 | 0.3253 |  0.3439|  **0.4016**|  0.3604 | 0.3544 |
> > | MentalArithmetic | 0.5333 | 0.6566 | 0.5483 | 0.6506 | **0.7116** | 0.6060 | 0.6475 |
> > | SEED-V | 0.2439 | 0.2440 | 0.2072 | 0.2334 | **0.2667** | 0.2495 | 0.2472 |
> > | High-Gamma | 0.8493 | 0.8507 | 0.8544 | 0.8509 | 0.8585 | 0.8573 | **0.8718** |
> > | SEED | 0.7593 | 0.7930 | 0.7823 | 0.8216 | 0.8181 | 0.7946 | **0.8246** |
> > | Stieger2021-UD | 0.7210 | 0.7136 | 0.7049 | 0.7123 | 0.7233 | 0.7146 | **0.7241** |
> >
> > The support-scaling analysis shows two clear patterns. First, across most datasets, performance peaks at k ≈ 8, after which larger support sets introduce additional variability and noise, leading to reduced accuracy, which supports that model does not always benefit from more supports [1]. Second, a smaller group of cleaner, more separable datasets (higher BACC) shows steady monotonic improvement as k increases. These results confirm that support size is task-dependent rather than “the more the better,” and moderate support (around 8) is typically optimal for heterogeneous EEG tasks.
> >
> > **C. The two-stage support configuration is designed to ensure stable ICL acquisition.** During early training, the model must learn the basic mechanics of conditioned inference under stable and sufficiently informative contexts. We therefore fix the support size to 8 in Stage 1 to avoid instability caused by fluctuating context length.  Once the model reliably exhibits ICL behavior, Stage 2 introduces a randomized 0–12 support range, enabling robustness across different context sizes.
> >
> > **D. We use a minimal Transformer decoder because it is the only architecture reliably exhibiting ICL, and it cleanly isolates the contribution of the ECHO paradigm without architectural confounds.**
> >  - **Transformer is the only architecture empirically validated for ICL.** Transformer is currently the only architecture with empirical evidence of supporting ICL. Systematic studies on GPT- and LLaMA-style models show that ICL behavior emerges from the multi-head self-attention mechanism, which enables support-conditioned reasoning over long contexts. Other sequence modeling architectures have not demonstrated stable or reproducible ICL in generalizable few-shot settings.
> >  - **Using a minimal Transformer decoder avoids architectural confounds.** We intentionally adopt a minimal, vanilla Transformer decoder so that performance gains can be clearly attributed to the ECHO ICL paradigm rather than to architectural enhancements. Since the focus of this work is validating the ICL formulation for EEG, introducing more complex designs would obscure the source of improvements. A minimal decoder aligns with standard ICL practice, ensures reproducibility, and isolates the paradigm’s independent contribution.
> >
> > ### ***Modification:***
> >  1. **Added Appendix D.3.2** providing a comparison of different decoder sizes.
> >  2. **Added Appendix D.2.2** presenting a support-size scaling analysis.
> >  3. **Added clarification in Section 4.1 Training & Environment Setting** explaining the rationale behind the two-stage design.

---

> > > ### Author Response · Authors · 2025-11-25
> > > **Response to Reviewer F1QH [3/3]**
> > >
> > > # Response for Weakness 3:
> > > ### ***Summary:***
> > > A standard multi-task learning baseline is requested.
> > > ### ***Response:***
> > > **We additionally compare CBraMod under a standard multi-task learning setup versus the ECHO ICL paradigm.** As shown in the table below (partial results displayed here), the multi-task baseline CBraMod-MH uses a conventional shared encoder with task-specific heads, whereas CBraMod-ECHO replaces ECHO’s lightweight encoder with CBraMod while keeping the ICL-driven Seq2Seq decoder. Although CBraMod-ECHO must infer the task type directly from support examples, it achieves more stable and consistently higher performance than the multi-task mathod.
> > >
> > >
> > > |||Attention |||High-Gamma||
> > > | -------------------- | :-: | :-: | :-: | :-: | :-: | :-: |
> > > | **Methods** | ACC-B | ROC AUC | PR AUC | ACC-B | ROC AUC | PR AUC |
> > > | CBraMod | 0.6478 | 0.7417 | 0.7468 | 0.7513 | 0.8277 | 0.8335 |
> > > | CBraMod-MH | 0.5000 | 0.4593 | 0.4654 | **0.7519** | 0.8426 | **0.8375** |
> > > | CBraMod-ECHO | **0.7222** | **0.8100** | **0.8056** | 0.7499 | **0.8481** | 0.8363 |
> > >
> > > |||Mumtaz2016 |||Stieger2021 LR||
> > > | -------------------- | :-: | :-: | :-: | :-: | :-: | :-: |
> > > | **Methods** | ACC-B | ROC AUC | PR AUC | ACC-B | ROC AUC | PR AUC |
> > > | CBraMod | 0.8946 | 0.9800 | 0.9765 | **0.8424** | **0.9339** | **0.9297** |
> > > | CBraMod-MH | 0.8542 | 0.9622 | 0.9507 | 0.8374 | 0.9331 | 0.9286 |
> > > | CBraMod-ECHO | **0.9035**  | **0.9773** | **0.9809** | 0.8370  | 0.9234 | 0.9241 |
> > >
> > >
> > >
> > >
> > > ### ***Modification:***
> > >  1. **Added Appendix D.2.4** comparing standard multi-task learning with ECHO-style learning.
> > >
> > > # Response for Question 1:
> > > This is closely related to Weakness 3. To avoid redundancy, we refer the reviewer to the response for Weakness 3.
> > >
> > > # Response for Question 2:
> > > This is closely related to Weakness 1. To avoid redundancy, we refer the reviewer to the response for Weakness 1.
> > >
> > > # Response for Question 3:
> > > This is closely related to Weakness 2 A. To avoid redundancy, we refer the reviewer to the response for Weakness 2 A.
> > >
> > > # Reference
> > > [1] More is not always better? enhancing many-shot in-context learning with differentiated and reweighting objectives, ACL 2025

---

### Official Review · Reviewer_r5rr · 2025-10-31

**Soundness:** 2
**Presentation:** 3
**Contribution:** 3
**Rating:** 4
**Confidence:** 5

**Summary:**

The paper proposes **ECHO**, a novel **decoder-centric seq2seq paradigm** for Large EEG Models (LEMs), reframing EEG decoding as in-context sequence modeling. It addresses critical limitations in existing LEMs—decoder bottlenecks, poor cross-task generalization, and lack of in-context learning. The method is evaluated on a broad set of EEG datasets with extensive ablation and zero-shot experiments, demonstrating consistent gains over strong baselines.

**Strengths:**

1. **Novel paradigm shift**: Introduces the first decoder-centric, seq2seq formulation for EEG modeling, moving beyond encoder-finetune or LLM-as-decoder paradigms.

2. **Effective use of in-context learning**: Demonstrates that EEG models can benefit from discrete support samples for dynamic adaptation without parameter updates.

3. **Comprehensive evaluation**: Multi-dataset, multi-task, zero-shot, and ablation studies provide strong empirical support for generalization and robustness.

**Weaknesses:**

1. **Unverified channel averaging in Eq. (5)**: The paper assumes averaging over aliased channels (e.g., T9 and M1) is valid, but no validation is provided—phase misalignment may introduce unnatural signal artifacts.

2. **Missing signal normalization**: It is unclear whether the transformed EEG tokens undergo normalization (e.g., z-score), which is crucial for stable training across datasets.

3. **Fixed-length tokenization in Eq. (8)**: Are all EEG signals be split into equal-length K patches? How do you encode patches with different sampling points?

4. **Ambiguity in embedding nature**: It is not clarified whether the EEG token embeddings are continuous or quantized. If continuous, how is cross-entropy loss applied during next-token prediction?

5. **Lack of warm-up stage evaluation**: The warm-up phase resembles standard multi-task supervised learning. Reporting its direct performance (without contextual training) would clarify whether gains stem from multi-task pretraining or true in-context learning.

6. **Biased task selection**: Downstream tasks focus on emotion and motor imagery. Missing key EEG applications like epilepsy detection or sleep staging limits claims of universality.

7. **Incomplete ablation of positional encoding**: The ablation removes sample- and text-level encodings, but does not test a baseline using only a single, global absolute position encoding across the entire sequence—this would better isolate the necessity of the hybrid design.

8. **Limited generalization beyond seen tasks**: The model relies on labels and tasks seen during training. No evidence is shown for few-shot adaptation to entirely new tasks, limiting claims of true generalization.

9. **Unreported model capacity differences**: Baseline models may vary significantly in parameter count. Moreover, no attempt is made to train baselines in the same multi-task setting, risking unfair comparison.

10. **No scaling law analysis for support samples**: The impact of increasing support count (e.g., 0 → 8 → 12) shows diminishing returns, but no systematic study (e.g., performance vs. number of supports) is conducted.

**Questions:**

See Weaknesses.

---

> ### Author Response · Authors · 2025-11-25
> **Response to Reviewer r5rr [1/4]**
>
> # Overview
> We sincerely thank reviewer r5rr for the thoughtful evaluation and valuable comments. To maintain clarity and consistency throughout the rebuttal, particularly given the interval since the initial submission, we organize each response using a unified three-part format:
>
> ***Summary:*** A concise restatement of the reviewer’s core concerns (partitioned as **A**, **B**, etc.).
>
> ***Response:*** A detailed clarification addressing each point.
>
> ***Modification:*** A description of the revisions made to the manuscript.
>
> # Response for Weakness 1:
> ### ***Summary:***
> **A.** Why is averaging T9 and M1 valid?
>
> **B.** Why use an averaging strategy that may introduce phase misalignment?
>
> ### ***Response:***
> **A. T9 and M1 are treated as name aliases rather than averaged channels  (Figure. 2 a Left).**  They correspond to the same scalp location under different electrode systems, so we only perform a name substitution without averaging.
>
> **B. For the channels where averaging is indeed applied, this choice is an intentional conservative design rather than a claim of optimal channel-fusion performance.**
>  - **We avoid introducing improvements from sophisticated fusion strategies that could obscure the contribution of the Seq2Seq paradigm itself.** The focus of ECHO is its Seq2Seq formulation and cross-task generalization, not the discovery of the channel-fusion technique. We intentionally validate ECHO under the simplest preprocessing.
>  - **ECHO merely demonstrates that prior electrode knowledge can solve the LEM channel-alignment issue.** We propose only one feasible solution, while noting that many extensions are possible within this framework (e.g., edge-learning techniques widely explored in EEG-related graph neural network research).
>  - **Experiments show that the averaging strategy has only a minimal impact on performance.** We added an evaluation using alternative fusion methods in $\text{ECHO}^\mathcal{E}$ (ECHO's encoder) to isolate their influence. As shown in the table, “Full Channel” means retaining all channels, while “Deleted” removes extra channels directly without averaging to avoid introducing phase misalignment. The performance difference between averaging and full channel remains within 2%.
>
> | **Methods** | KoreaU (62 Channel) | HighGamma (128 Channel) |
> |------------|:------------:|:----------------------------:|
> | Full Channel   |  0.7217       |  0.6780                       |
> | Average   |  0.7031       |  0.6911                       |
> | Deleted   |  0.6983       |  0.6476                       |
>
> ### ***Modification:***
>
>  1. **Added Appendix D.2.1** comparing different channel-fusion methods.
>  2. **Added a clarification in Section 3.1 *Intuition* (a)** explaining the rationale behind choosing the averaging strategy and outlining possible alternative fusion methods.
>
> # Response for Weakness 2:
> ### ***Summary:***
> Missing clarification about signal normalization.
> ### ***Response:***
> **We did not apply additional normalization.** The preprocessing consists only of band-pass filtering and, for a few datasets, a simple amplitude scaling (division by 100) to keep values within a comparable numeric range. This is consistent with the preprocessing used in CBraMod and CodeBrain.
>
>
> # Response for Weakness 3:
> ### ***Summary:***
> Do signals with different sampling lengths also get divided into the same number of equal-length K patches?
> ### ***Response:***
> **All EEG signals are first resampled to 250 Hz and padded to a fixed 10s length before entering Eq. (8).**  This unified preprocessing ensures that every sequence can be consistently partitioned into the same number of patches.
>
> ### ***Modification:***
> **Added a clarification in Section 4.1 Dataset Setting** to state that all signals are padded to a fixed length.
>
> # Response for Weakness 4:
> ### ***Summary:***
> Are the EEG embedding tokens continuous or discrete? If continuous, why is cross-entropy loss applicable?
> ### ***Response:***
> **ECHO’s EEG tokens are continuous vectors, but they are not prediction targets.**  They only participate in cross-attention. The decoder’s next-token prediction operates solely over a discrete vocabulary of task and label tokens (identical to standard LLM decoding). As the model predicts only discrete tokens, the output space is categorical, and cross-entropy is fully appropriate.
>
> # Response for Weakness 5:
> ### ***Summary:***
> Provide warm-up stage results to demonstrate the contribution of the ICL training stage.
> ### ***Response:***
> **The warm-up stage results are already reported as $\text{ECHO}^\mathcal{E}$ in Figure. 1b.**  It shows lower performance without the ICL training stage.
> ### ***Modification:***
>  1. **Added the $\text{ECHO}^\mathcal{E}$ results to Table 2,** illustrating the performance improvements introduced by the ECHO paradigm across the encoder.

---

> ### Author Response · Authors · 2025-11-25
> **Response to Reviewer r5rr [2/4]**
>
> # Response for Weakness 6:
> ### ***Summary:***
> The downstream evaluation lacks sleep and epilepsy tasks.
> ### ***Response:***
>
>  - **Sleep evaluation is already included in Appendix D.1.1.**  $\text{ECHO}^\mathcal{L}$ achieves competitive performance on sleep staging dataset ISRUC-S1 despite operating under a challenging multi-task setting with short 30s inputs, whereas baselines benefit from long 10-minute signals in single-task conditions *(Sleep staging benefits from long-range temporal structure, long segments (e.g., 10 minutes) generally yield higher accuracy. However, for computational efficiency and consistency of the framework, ECHO uses 30-second segments as input)*.
>
> |  |  |   **ISRUC-S1**    |     |
> |-------------|:--------------------------:|:------:|:--------:|
> |  **Methods**  | ACC-B | Kappa | F1-Weighted |
> | EEGNet | 0.6238 ± 0.0142 | 0.5921 ± 0.0142 | 0.7032 ± 0.0309 |
> | BIOT | 0.7527 ± 0.0121 | 0.7192 ± 0.0231 | 0.7790 ± 0.0146 |
> | LaBraM | 0.7633 ± 0.0102 | 0.7231 ± 0.0182 | 0.7810 ± 0.0133 |
> | EEGPT | 0.4012 ± 0.0177 | 0.2223 ± 0.0227 | 0.3111 ± 0.0110 |
> | CBraMod | **0.7865 ± 0.0110** | 0.7442 ± 0.0152 | 0.8011 ± 0.0099 |
> | CodeBrain | 0.7835 ± 0.0033 | **0.7476 ± 0.0040** | **0.8020 ± 0.0018** |
> | $\text{ECHO}^\mathcal{L}$ | 0.7838 ± 0.0012 | 0.7303 ± 0.0032 | 0.7893 ± 0.0022 |
>
>  - **We have additionally included epilepsy results.** As shown in the following table, $\text{ECHO}^{EP}$ (ECHO pretrained with additional epilepsy dataset) shows solid ability even without support samples, and adding support further boosts performance, especially in PR AUC (0.4985), where $\text{ECHO}^{EP}$ surpasses all baselines.
>
> |                          |                     |     **CHB-MIT**     |                     |
> | ------------------------ |:-------------------:|:-------------------:|:-------------------:|
> | **Methods**              |        ACC-B        |       ROC AUC       |       PR AUC        |
> | BIOT                     |   0.7068 ± 0.0457   |   0.8761 ± 0.0284   |   0.3277 ± 0.0460   |
> | LaBraM                   |   0.7075 ± 0.0358   |   0.8679 ± 0.0199   |   0.3287 ± 0.0402   |
> | EEGPT                    |   0.5481 ± 0.0151   |   0.8892 ± 0.0066   |   0.3073 ± 0.0641   |
> | CBraMod                  | **0.7398 ± 0.0284** |   0.8892 ± 0.0154   |   0.3689 ± 0.0382   |
> | CodeBrain                |   0.7273 ± 0.0240   | **0.8961 ± 0.0174** |   0.4377 ± 0.0288   |
> | $\text{ECHO}^{EP}$ (No Support) |   0.5671 ± 0.0078   |   0.8290 ± 0.0052   |   0.3872 ± 0.0049   |
> | $\text{ECHO}^{EP}$              |   0.6199 ± 0.0040   |   0.8762 ± 0.0077   | **0.4985 ± 0.0032** |
>
> ### ***Modification:***
>
>  1. **Added Appendix D.1.3** reporting the new epilepsy dataset results.
>  2. **Added description in Appendix A.1.8** about the epilepsy dataset.
>
> # Response for Weakness 7:
> ### ***Summary:***
> Replace two positional encodings with a single global absolute positional encoding (PE) to verify the necessity of the hybrid design.
> ### ***Response:***
>
> **The “hybrid PE” is not composed of two heterogeneous or structurally different PE mechanisms.** Rather, it reflects that EEG embedding tokens and text tokens belong to two separate streams that interact only through cross-attention. Because these streams are disjoint in self-attention, PE are applied separately (thus called it "hybrid"). However, both are fundamentally the same standard Transformer absolute PE, merely assigned to different token types.
>
> **We conducted an ablation to verify the necessity.** All tokens were concatenated and given a single global absolute positional encoding before being split again for cross-attention. As shown in the following table (only Attention dataset are used), this design caused training to collapse. The collapse occurs because the global positional encoding overwrites the pretrained positional structure of text tokens, placing them at incompatible indices and disrupting the pretrained decoder’s positional semantics, which leads to an unstable decoding space and failed training.
>
> |                          |                     |     **Attention**     |                     |
> | ------------------------ |:-------------------:|:-------------------:|:-------------------:|
> | **Methods**              |        ACC-B        |       ROC AUC       |       PR AUC        |
> | Global PE  |   0.5000   |   0.5231   | 0.2142 |

---

> > ### Author Response · Authors · 2025-11-25
> > **Response to Reviewer r5rr [3/4]**
> >
> > # Response for Weakness 8:
> > ### ***Summary:***
> > Can ECHO generalize to unseen tasks and labels?
> >
> > ### ***Response:***
> > **ECHO is currently unable to generalize to entirely unseen tasks. However, we would like to clarify three points:**
> > * **Generalizing to entirely new tasks without prior label exposure remains a significant challenge.** EEG lacks a unified semantic space, so the model cannot reliably infer new task structures or label meanings from support examples alone. Although introducing textual task or label descriptions like ECHO may eventually enable such generalization, text-augmented EEG modeling is still rare and the field has not yet reached the stage where unseen-task transfer is feasible in practice.
> > * **ECHO conducted cross-dataset evaluations.** These experiments involve substantial distribution shifts even when the task types remain consistent due to the heterogenity of sampling techniques. The ability of the model to handle these significant shifts demonstrates a practical level of generalization and robustness.
> >
> > | | | Cho2017 (Unseen for ECHO) ||| BCIC 2020-T1 (Unseen for ECHO) ||
> > | :--- | :-: | :-: | :-: | :-: | :-: | :-: |
> > | **Methods** | ACC-B | ROC AUC | PR AUC | ACC-B | ROC AUC | PR AUC |
> > | Cbramod (Fine Tune) | 0.7033 ± 0.0327 | 0.8081 ± 0.0429 | 0.7959 ± 0.230 | 0.5700 ± 0.0185 | 0.6058 ± 0.0236 | 0.5371 ± 0.0186 |
> > | $\text{ECHO}^\mathcal{E}$ | 0.5640 ± 0.0315 | 0.6021 ± 0.0498 | 0.6126 ± 0.0207 | 0.5250 ± 0.0751 | 0.6796 ± 0.1459 | 0.6697 ± 0.0602 |
> > | ECHO(No Support) | 0.5881 ± 0.0023 | 0.6399 ± 0.0035 | 0.6267 ± 0.0021 | 0.7500 ± 0.0024 | 0.8232 ± 0.0053 | 0.8153 ± 0.0118 |
> > | ECHO | 0.6044 ± 0.0032 | 0.6470 ± 0.0053 | 0.6460 ± 0.0019 | 0.7782 ± 0.0042 | 0.8566 ± 0.0028 | 0.8552 ± 0.0032 |
> >
> > * **The current limitation stems from model scale and semantic alignment rather than the paradigm itself.** ECHO validates the generative seq2seq framework, but it currently lacks the massive parameter scale and instruction-following capabilities of Large Language Models. However, this generative paradigm is structurally compatible with LLM integration. Scaling the current decoder with an aligned LLM and more high quality EEG-Text datasets in the future works could theoretically enable the zero-shot task reasoning.
> >
> > ### ***Modification:***
> > Added **6. Limitation** providing further discussion.

---

> > > ### Author Response · Authors · 2025-11-25
> > > **Response to Reviewer r5rr [4/4]**
> > >
> > > # Response for Weakness 9:
> > > ### ***Summary:***
> > > **A.** No parameter configurations comparison.
> > >
> > > **B.** Apply the ECHO paradigm to the baselines to verify its effectiveness.
> > > ### ***Response:***
> > > **A. ECHO has fewer parameters compared to conventional encoder-centric models.** For fair comparison, all encoder-centric baselines include their task-specific classification heads, and the reported model size is the average parameter count across tasks.
> > >
> > > | **Methods** | Paradigm | Parameters | MACs    | FLOPs   |
> > > |:------------|:------------:|:-------:|:-------:|:-------:|
> > > | EEGPT   |  Encoder-Centric      |     51.66M  |  4.91G |9.84G  |
> > > | LaBraM   |  Encoder-Centric       |  9.88M            |  0.67G |1.34G     |
> > > | CBraMod   |  Encoder-Centric       |  54.95M             |4.21GG  |6.39G   |
> > > | CodeBrain   |  Encoder-Centric       |  66.02M       |  4.42G |8.84G   |
> > > | ECHO   |  Decoder-Centric       | 41.98M   | 2.91G   | 5.81G  |
> > >
> > > **B. The ECHO paradigm remains effective for the large-scale encoder.** We added experiments combining ECHO with the encoder-centric CBraMod model. As presented in the table below (partial results; full results provided in the main text), the large pretrained encoder also achieves better multi-task performance under the ECHO paradigm, confirming that the performance gains arise from the Seq2Seq paradigm itself.
> > >
> > > | | | High-Gamma ||| Attention ||
> > > | :--- | :-: | :-: | :-: | :-: | :-: | :-: |
> > > | **Method** | ACC-B | ROC AUC | PR AUC | ACC-B | ROC AUC | PR AUC |
> > > | CBraMod | 0.7478 | 0.8292 | 0.8314 | 0.6478 | 0.7417 | 0.7468 |
> > > | ECHO-CBraMod | **0.7499** | **0.8481** | **0.8363** | **0.7222** | **0.8100** | **0.8056** |
> > >
> > > | | | Mumtaz2016 ||| Stieger2021-LR ||
> > > | :--- | :-: | :-: | :-: | :-: | :-: | :-: |
> > > | **Method** | ACC-B | ROC AUC | PR AUC | ACC-B | ROC AUC | PR AUC |
> > > | CBraMod | 0.8946 | **0.9800** | 0.9765 | **0.8424** | **0.9339** | **0.9297** |
> > > | ECHO-CBraMod | **0.9035** | 0.9773 | **0.9809** | 0.8370 | 0.9234 | 0.9241 |
> > >
> > >
> > > ### ***Modification:***
> > >  1. **Added Appendix D.3.1** providing a comparison and analysis of model parameter sizes.
> > >  2. **Added Appendix D.2.3** containing the extended experiments integrating ECHO with CBraMod.
> > >
> > >
> > > # Response for Weakness 10:
> > > ### ***Summary:***
> > > No scaling law analysis for support samples.
> > > ### ***Response:***
> > > **We have added experiments analyzing the impact of support-sample size.** As shown in the table below, ECHO generally achieves the largest gains with 8 support examples, while a subset of tasks benefits from 12 supports. Since the downstream datasets differ in the number of classification categories, we report only BACC for comparability. All support-size analyses are evaluated using results from seed 0.
> > >
> > > | **Dataset** | 0 | 2 | 4 | 6 | 8 | 10 | 12 |
> > > |:-|:-:|:-:|:-:|:-:|:-:|:-:|:-:|
> > > | SEED-IV  | 0.3399  |0.3476 | 0.3253 |  0.3439|  **0.4016**|  0.3604 | 0.3544 |
> > > | MentalArithmetic | 0.5333 | 0.6566 | 0.5483 | 0.6506 | **0.7116** | 0.6060 | 0.6475 |
> > > | SEED-V | 0.2439 | 0.2440 | 0.2072 | 0.2334 | **0.2667** | 0.2495 | 0.2472 |
> > > | High-Gamma | 0.8493 | 0.8507 | 0.8544 | 0.8509 | 0.8585 | 0.8573 | **0.8718** |
> > > | SEED | 0.7593 | 0.7930 | 0.7823 | 0.8216 | 0.8181 | 0.7946 | **0.8246** |
> > > | Stieger2021-UD | 0.7210 | 0.7136 | 0.7049 | 0.7123 | 0.7233 | 0.7146 | **0.7241** |
> > >
> > > The support-scaling analysis shows two clear patterns. First, across most datasets, performance peaks at k ≈ 8, after which larger support sets introduce additional variability and noise, leading to reduced accuracy, which supports that model does not always benefit from more supports [1]. Second, a smaller group of cleaner, more separable datasets (higher BACC) shows steady monotonic improvement as k increases. These results confirm that support size is task-dependent rather than “the more the better,” and moderate support (around 8) is typically optimal for heterogeneous EEG tasks.
> > >
> > > ### ***Modification:***
> > >  1. **Added Appendix D.2.2** presenting a support-size scaling analysis.
> > >
> > > # Reference
> > > [1] More is not always better? enhancing many-shot in-context learning with differentiated and reweighting objectives, ACL 2025

---

> > > > ### Comment · Reviewer_r5rr · 2025-11-25
> > > >
> > > > Thank you for your detailed responses and clarifications. These have addressed most of my concerns, and I will raise my score.
> > > >
> > > > W1. How is the mapping set ( M_c ) defined in equation (5)? Is there any reference from neuroscience literature for this definition template? You mention that all channel layouts are aligned with the 75-channel system; how are missing channels handled in datasets with fewer than 75 channels? Additionally, the phrase "… all possible aliases or adjacent variants of that channel …" suggests that not all mappings are simply aliases for the same position.
> > > >
> > > > W9. How do you explain the noticeable performance drop of CBraMod on some datasets after incorporating the seq2seq paradigm, as shown in Table 19? Could this be related to the multi-task supervision training step? Intuitively, a more powerful encoder should exhibit better generalization under the seq2seq paradigm.
> > > >
> > > > I also have some additional questions. Although the core of this paper is seq2seq learning, the entire training process focuses only on constructing the final classification space, without jointly optimizing the semantic space of the EEG sequences themselves. I am not suggesting that the encoder does not participate in backpropagation, but rather that the temporal context of the sequences is not explicitly modeled—it is simply treated as dense features input to the decoder. For LLMs, the key to few-shot learning lies in the context learning ability introduced during pretraining. In this work, sequence encoding is still handled by the encoder, with the added token components only contributing to learning in the downstream classification head. I believe this is a key limitation preventing further generalization in the current work—semantic information from various classification tasks is not being contextually modeled within the EEG sequence itself. Nevertheless, I consider this work to be an interesting attempt, particularly in its effort to unify various downstream tasks in a more flexible manner.
> > > >
> > > > Additionally:
> > > >
> > > > (1) Have you compared your work directly with related research in the few-shot learning domain? These works also leverage few-shot groups for meta-learning of classification prototypes.
> > > >
> > > > (2) Given the many powerful, encoder-based EEG models with **a large scale of parameters** already available, it would be more convincing to replace this paradigm with a fine-tuning approach in these foundation models.
> > > >
> > > > (3) Evaluating your model’s performance on each individual downstream dataset against SOTA models specifically designed for those domains would provide a clearer benchmark for model assessment.

---

> > > > > ### Author Response · Authors · 2025-12-04
> > > > > **Response to Reviewer r5rr [5/4]**
> > > > >
> > > > > # Overview
> > > > > We sincerely thank Reviewer r5rr for the positive assessment of our work. We also appreciate the reviewer’s active engagement in the subsequent discussion, which has greatly helped us further refine the paper. As before, we follow the three-part response format to address the reviewer’s comments in detail.
> > > > >
> > > > > ---
> > > > > # Additional Response to Weakness 1:
> > > > > ### ***Summary:***
> > > > > **A.** How is the standardized template channel set defined? Is it derived from prior neuroscientific knowledge?
> > > > >
> > > > > **B.** How to handle data with fewer than 75 channels?
> > > > >
> > > > > **C.** How does the mapping handle electrodes with non-matching names and spatial positions?
> > > > >
> > > > > ### ***Response:***
> > > > > **A. This is an empirical choice.** It reflects our accumulated observations of consistently informative EEG channels across paradigms, rather than a predefined neuroscientific template. We have added a brief explanation in the appendix for clarity and share these empirical insights with readers.
> > > > >
> > > > > **B. Missing channels are padded with zeros.**
> > > > >
> > > > > **C. Non-matching channels are averaged according to the standardized template channel set.** Our 75-channel template spans major cortical regions, and each template channel groups electrodes that fall at the same spatial position even if labeled differently across datasets. For example, for Fp1:
> > > > > ```
> > > > > "Fp1": ["Fp1","AFp7h","AFp5h","AFp3h","Fp1h","AFp5","AFp3"]
> > > > > ```
> > > > > The label “**Fp1**” follows the 10–20 system, while “**Fp1h**” refers to the same anatomical electrode in the 10–10 system, and the **AFp-series** corresponds to neighboring frontal-pole electrodes in high-density EEG systems. When a dataset uses the 10–10 system (e.g., only providing Fp1h), we perform a direct name substitution. When a dataset uses a high-density system (e.g., 10–5 with AFp electrodes), we average all electrodes mapped to the Fp1 anatomical location to obtain the standardized Fp1 channel.
> > > > >
> > > > > ### ***Modification:***
> > > > > 1. **Added Appendix E** to explain standardized template channel set.
> > > > >
> > > > > ---
> > > > > # Additional Response to Weakness 9:
> > > > >
> > > > > ### ***Summary:***
> > > > >
> > > > > Why does ECHO-CBraMod show performance drops on certain datasets?
> > > > >
> > > > > ### ***Response:***
> > > > >
> > > > > **The task is substantially more challenging.** ECHO-CBraMod performs multi-task classification without task prompts, which is inherently more difficult than the original CBraMod setup. Therefore, a performance drop on certain datasets is expected.
> > > > >
> > > > > **Differences in optimal training configurations for CBraMod.** As a large encoder, CBraMod requires different hyperparameter settings (e.g., learning rate, freezing strategy, optimizer parameters, regularization) compared to the encoder used in ECHO. We were unable to conduct a full grid search under the time constraints, which may contribute to the observed performance drop.
> > > > >
> > > > > **Limited number and size of datasets.** For demonstrating the scalability of the Seq2Seq paradigm, we only used six relatively small datasets. Under the multi-task Seq2Seq setting, limited data intensifies inter-task interference and makes encoder feature drift less stable, which in turn amplifies the training variance of high-capacity models. Please refer to **Response to Additional Question 3** for a more detailed explanation.

---

> ### Author Response · Authors · 2025-12-04
> **Response to Reviewer r5rr [6/4]**
>
> # Response to Addtional Question 1:
>
> ### ***Summary:***
> Does ECHO suffer from this limitation: compressing EEG embedding tokens into a dense decoder input (instead of explicit contextual input) and thus failing to establish a link between class and the temporal context within the EEG?
>
> ### ***Response:***
>
> We sincerely thank the reviewer for raising this question, and we agree with the concern that “the temporal context of the sequences should be explicitly modeled.” Here we clarify that **ECHO indeed follows this modeling principle and does not represent an entire EEG trial using a single dense feature vector.**
>
>
> **Each EEG embedding token is a concatenation of multiple window-level embeddings.** To avoid confusion, we distinguish between sample-level EEG tokens and window-level EEG tokens:
>
> 1. *Sample-level EEG token*: the EEG embedding token that operate at the same level as text tokens.
> 2. *Window-level EEG token*: the temporally contiguous tokens derived from segmenting an EEG trial into (K) windows, each preserving local temporal context.
>
> **Each EEG embedding token is a concatenation of multiple window-level embeddings.** To avoid confusion, we distinguish between sample-level EEG tokens and window-level EEG tokens:
>
> 1. *Sample-level EEG token*: the EEG embedding token that operate at the same level as text tokens.
> 2. *Window-level EEG token*: the temporally contiguous tokens derived from segmenting an EEG trial into (K) windows, each preserving local temporal context.
>
>
> * **Each sample-level EEG token contains (K) window-level EEG tokens.** For the (S + 1) sample-level EEG tokens $\bigcup_{n=1}^{S+1}\{\lbrace\mathcal{E}(X_n)\}\rbrace$ used as decoder inputs (i.e., (S) support samples $\lbrace X_s \rbrace_{s=1}^S$ plus 1 target sample $X$ ):
>     \begin{equation*}
>     \bigcup_{n=1}^{S+1}\{\lbrace\mathcal{E}(X_n)\}\rbrace=\lbrace\mathcal{E}(X),\mathcal{E}(X_1),\cdot\cdot\cdot,\mathcal{E}(X_{S-1}),\mathcal{E}(X_{S})\rbrace,
>     \end{equation*}
>
>     where $\mathcal{E}(X_n)$ denotes the sample-level token obtained after passing an EEG trial $X_n$ through the encoder block. In practice, this token represents a group of \(K\) temporally contiguous window-level EEG tokens $h_{(s,k)}$:
>     \begin{equation*}\mathcal{E}(X_n)=\bigcup_{k=1}^{K}\{\lbrace h_{(n,k)}\}\rbrace=\lbrace h_{(n,1)},h_{(n,2)},\cdot\cdot\cdot,h_{(n,K-1)},h_{(n,K)}\rbrace.
>     \end{equation*}
>
> * **With dual-Position Encoding (PE) design, ECHO can models the temporal context and its relationship to class across window-level EEG tokens.** As shown in Eqs. (8) and (9) in the main paper, the (K) window-level EEG tokens from the same trial are grouped as one sample-level token and assigned a sample-level PE before entering the decoder. They also receive a window-level PE within the encoder. Thus, each $h_{(n,k)}$ carries two positional encodings:
>
>      \begin{equation*}
>      h_{(n,k)}+\text{PE}^{\text{enc}}_{(n,k)}+\text{PE}^{\text{dec}}_n,
>      \end{equation*}
>
>     where the second PE indicates to the decoder that $h_{(n,k)}$ belongs to the (n)-th EEG sample, and $\text{PE}^{\text{enc}}_{(n,k)}$ specifies its position within that sample. Together, they enable the decoder to model the temporal context within each sample through attention.
>
> ### ***Modification:***
> 1. **Revised several annotations in Figure 2**.
> 2. **Updated Section 3.2, Hybrid Positional Encoding.**
> 3. **Revised Section 3.1, Encoder Block.**
>
>
> ---
> # Response to Addtional Question 2:
> ### ***Summary:***
> Compare against few-shot learning?
> ### ***Response:***
>
> **ECHO is fundamentally different from few-shot learning methods because the support samples do not update any parameters.** Unlike few-shot methods that rely on task-specific training and prototype-based parameter updates, ECHO performs pure in-context learning: all model weights remain frozen, and the support samples are incorporated only through the decoder's attention mechanism. The key distinction is that few-shot learning achieves cross-task generalization through parameter updates, whereas ECHO infers the task structure entirely from input context without any fine-tuning or meta-learning.

---

> > ### Author Response · Authors · 2025-12-04
> > **Response to Reviewer r5rr [7/4]**
> >
> > # Response to Addtional Question 3:
> > ### ***Summary:***
> >
> > Given that encoder-centric LEMs can be efficiently adapted through task-specific fine-tuning, and most downstream applications are single-task, it is reasonable to ask whether ECHO’s multi-task extension is sufficiently justified?
> >
> > ### ***Response:***
> >
> > **ECHO also is a fine-tuning method that equips an EEG encoder with ICL capabilities rather than being limited to multi-task learning.** It is not designed to replace existing EEG encoders; instead, it provides a fine-tuning paradigm that grants the encoder Seq2Seq reasoning and in-context learning abilities. We have added experiments showing task-specific fine-tuning under the Seq2Seq paradigm (see the table below), demonstrating that ECHO can serve as an fine-tuning method that extends an encoder’s functionality into the ICL regime:
> >
> > | Dataset | Metric | Multi-Tasks Learning | Single-Task Fine-tuning |
> > | :--- | :--- | :---: | :---: |
> > | **Cho2017** | BACC | 0.7190 | **0.7433** |
> > | | ROC AUC | 0.8094 | **0.8190** |
> > | | PR AUC | **0.8200** | 0.8149 |
> > | **Weibo2014** | BACC | 0.6531 | **0.6594** |
> > | | ROC AUC | 0.7046 | **0.7120** |
> > | | PR AUC | **0.7074** | 0.7016 |
> > | **PhysioNet** | BACC | 0.5667 | **0.6008** |
> > | | Kappa | 0.4214 | **0.4677** |
> > | | F1-Weighted | 0.5604 | **0.6002** |
> >
> > **Multi-task training is not only for handling multi-task scenarios; expanding supervised data itself strengthens robustness on task-specific downstream settings.**
> >
> > * **Theoretically, a decoder trained in a multi-task setting exhibits greater stability when evaluated on single-task scenarios.** By expanding supervised signals across datasets, multi-task training imposes stronger cross-task constraints on the representation space, enabling the decoder to learn more stable and domain-invariant features. This parallels the way multi-dataset self-supervision enhances generalization by increasing the structural diversity of the training data. Learning under broader supervision leads to more stable shared representations and better specific downstream generalization, consistent with observations in [1,2].
> >
> > * **Empirically, Seq2Seq training yields substantially improved stability for ECHO compared with task-specific fine-tuning.**
> > Several EEG datasets are often small and sparsely distributed, making task-specific adaptation prone to overfitting. The Seq2Seq paradigm introduces additional data domains that partially cover these sparse distributions, enabling the decoder to behave more robustly on small datasets. We summarize several practical insights and patterns we observed when training multi-task EEG models (the source dataset represents an auxiliary dataset that aids in target dataset training):
> >
> > | Target Dataset | Metric | Key Source Dataset  | w/o Source | with Source | Gain |
> > | :--- | :---: | :---: | :---: | :---: | :---: |
> > | **BCI IV 2a** | BACC | **High Gamma** | 0.3386 | **0.4763** | **+0.1377** |
> > | | Kappa | | 0.1181 | **0.3015** | **+0.1834** |
> > | | F1-Weighted | | 0.2804 | **0.4632** | **+0.1828** |
> > | **PhysioNet** | BACC | **High Gamma** | 0.5372 | **0.5667** | **+0.0295** |
> > | | Kappa | | 0.3830 | **0.4214** | **+0.0384** |
> > | | F1-Weighted | | 0.5186 | **0.5604** | **+0.0428** |
> > | **Mental Arithmetic**| BACC | **Attention** | 0.5908 | **0.6851** | **+0.1851** |
> > | | ROC AUC | | 0.6417 | **0.7500** | **+0.1466** |
> > | | PR AUC | | 0.4782 | **0.7530** | **+0.1706** |
> >
> > **Multi-task learning is a natural future direction for EEG foundational.** Multi-task capability is aligned with the general trend of foundation models toward broader universality. ECHO therefore represents a practical step toward more universal and generalizable EEG foundation models, achieving both in-context learning and multi-task generalization within a unified framework.

---

> > > ### Author Response · Authors · 2025-12-04
> > > **Response to Reviewer r5rr [8/4]**
> > >
> > > # Response to Addtional Question 4:
> > >
> > > ### ***Summary:***
> > > Compare ECHO with models specifically designed for the downstream task.
> > >
> > > ### ***Response:***
> > > **We have added experiments comparing ECHO against several open-source SOTA models designed for the corresponding downstream domains.** Overall, ECHO achieves performance comparable to or even better than task-specific SOTA models across most downstream tasks. The results are summarized in the table below:
> > > | Target Dataset | SOTA Model | Metric | SOTA Baseline | ECHO (Multi-source) |
> > > | :--- | :--- | :--- | :---: | :---: |
> > > | **SEED-V** | SERA [3] | Balanced Acc | 0.2427 | **0.2484** |
> > > |  |  | Weighted F1 | 0.2318 | **0.2456** |
> > > |  |  | Kappa | 0.0552 | **0.0640** |
> > > | **HighGamma** | DCNet [4] | Balanced Acc | 0.7063 | **0.8554** |
> > > |  |  | ROC-AUC | 0.7045 | **0.9201** |
> > > |  |  | PR-AUC | 0.6083 | **0.9117** |
> > > | **BCI IV 2a** | AGTCNet [5] | Balanced Acc | **0.6085** | 0.4763 |
> > > |  |  | Weighted F1 | **0.6012** | 0.4632 |
> > > |  |  | Kappa | **0.4780** | 0.3015 |
> > > | **Attention** | EEG Deformer [6] | Balanced Acc | 0.8189 | **0.8194** |
> > > |  |  | ROC-AUC | 0.8432 | **0.8973** |
> > > |  |  | PR-AUC | 0.8515 | **0.8952** |
> > >
> > > ---
> > >
> > > # Reference
> > >
> > > [1] Zhang Q, Zhong J, ZongSheng L, et al. Multi-dataset Joint Pre-training of Emotional EEG Enables Generalizable Affective Computing[C]. NeurIPS, 2025.
> > >
> > > [2] Gu X, Cui Y, Huang J, et al. Dataseg: Taming a universal multi-dataset multi-task segmentation model[C]. NeurIPS, 2023.
> > >
> > > [3] Jia Z, Xu M, Wang J, et al. Sera: Separated coarse-to-fine representation alignment for cross-subject EEG-based emotion recognition[C]. ACM MM, 2025.
> > >
> > > [4] Peng W, Liu K, Shi J, et al. EEG-DCNet: A fast and accurate MI-EEG dilated CNN classification method[EB/OL]. arXiv, 2024.
> > >
> > > [5] Lim G B S, Lim B G S, Bandala A A, et al. AGTCNet: A graph-temporal approach for principled motor imagery EEG classification[EB/OL]. arXiv, 2025.
> > >
> > > [6] Ding Y, Li Y, Sun H, et al. EEG-Deformer: A dense convolutional transformer for brain-computer interfaces[J]. IEEE JBHI, 2024.

---

### Official Review · Reviewer_zEi1 · 2025-11-03

**Soundness:** 2
**Presentation:** 3
**Contribution:** 2
**Rating:** 4
**Confidence:** 4

**Summary:**

ECHO reframes Large EEG Models as a decoder-centric seq2seq problem that serializes support EEG samples, task/label tokens, and a target EEG, letting a Transformer decoder infer the target’s task and label via next-token prediction—thereby enabling in-context and multi-task EEG learning without parameter updates.  Technically, it adds (i) a simple channel-alignment preprocessor to standardize heterogeneous electrode layouts, (ii) a hybrid positional encoding spanning token-, sample-, and text-levels to fuse continuous EEG with discrete control tokens, and (iii) a staged training scheme that progressively increases support examples to instill contextual reasoning.

**Strengths:**

The strengths of this article are as follows:

1. It introduces a decoder-centric paradigm that mitigates the limitations of prior approaches, such as poor generalization and heavy reliance on inductive biases, while enabling in-context learning.
2. The argumentation is well structured and easy to follow.
3. The experimental evaluation is comprehensive: it benchmarks ECHO against various baselines and includes ablations on two forms of positional encoding, demonstrating that both are necessary.

**Weaknesses:**

This paper presents several weaknesses:

1. While the paper rightly highlights the generalization bottlenecks of encoder-centric methods, the proposed method, ECHO appears confined to a single task family—classification. By contrast, encoder-centric approaches can be lightweight and adaptable across prediction, classification, and captioning, which may weakens the paper's contribution.
2. The proposed approach depends on a pre-trained encoder, atop which a decoder supporting multiple classification tasks is trained. Clarifying how this design substantively departs from an encoder-centric adaptation tailored to general classification would help strengthen the methodological positioning.
3. The validation of ECHO’s zero-shot capability could be more persuasive. First, using SEED, data visible during training for “zero-shot” prediction may not constitute a strict zero-shot setting. Second, the evaluation is limited to the BCIC 2020-T1 for Motor Imagery task with only one baseline. A broader assessment across additional datasets and stronger, diverse baselines would make the zero-shot claims more compelling.
4. The paper’s central contribution seems to lie in its decoder-centric training paradigm. To further underscore its robustness, it may be helpful to evaluate additional combinations of encoder and decoder architectures.

**Questions:**

1. Different classification tasks involve varying numbers of classes. How does ECHO accommodate variable class cardinalities within a unified model architecture? I wasn’t able to find an explicit explanation; if it is already covered, could you point me to the relevant section?
2. Although the overall structure is clear, a few presentation details were confusing. For instance, the output implied by Eq. (11) appears inconsistent with Fig. 2: the figure does not show $S_{\text{out}}$. I initially interpreted $T_{2S+3}$​ as the classification output, which seems at odds with Eq. (11).
3. Could the authors clarify why removing textual positional encoding prevents the model from making meaningful predictions? What does a “no valid predictions” look like here? Intuitively, for classification, the worst case should approximate random guessing.

---

> ### Author Response · Authors · 2025-11-25
> **Response to Reviewer zEi1 [1/3]**
>
> # Overview
> We sincerely thank reviewer zEi1 for the thorough evaluation and insightful comments. To ensure that our responses remain clear and systematically organized, especially given the time elapsed since the initial submission, we present each reply in a consistent three-part structure:
>
> ***Summary:*** A concise restatement of the reviewer’s core concerns (partitioned as **A**, **B**, etc.).
>
> ***Response:*** A detailed clarification addressing each point.
>
> ***Modification:*** A description of the revisions made to the manuscript.
>
> # Response for Weakness 1:
> ### ***Summary:***
>  Whether ECHO’s focus on classification tasks limits its contribution？
> ### ***Response:***
> **“Single task family—classification” is not a limitation of ECHO, but rather a consequence of the benchmark conventions in the current EEG community.**
>
>  - **EEG benchmarks have been centered on classification tasks.** Most public EEG datasets provide only discrete labels, and existing EEG foundation models are universally evaluated under classification protocols. Our use of classification follows this benchmark convention to ensure fair comparison.
>  - **ECHO’s Seq2Seq architecture is not limited to classification.** ECHO's decoder generates target tokens that entirely defined by the vocabulary. By simply redefining token semantics and tokenization, the same framework can represent class labels, discretized continuous values, event sequences, or higher-level textual outputs, enabling natural extension to other task without any architectural changes.
>
> # Response for Weakness 2:
> ### ***Summary:***
>  What distinguishes ECHO from encoder-centric methods, given that it also relies on a pre-trained encoder?
>
> ### ***Response:***
> **ECHO does not merely rely on a pre-trained encoder, it extends its generalization capacity through a fundamentally different Seq2Seq paradigm.**
>  - **Technically, ECHO establishes a richer and conditional reasoning process than encoder-centric classifiers.** Instead of mapping features to a fixed classification head, ECHO performs multi-stage conditional generation (task identification → sub-class discrimination → label token prediction), using support samples as contextual cues. This structured inference process enables ECHO to perform comparably to direct “feature → label” mapping even in more challenging multi-task settings (no task prompts).
>  - **Empirically, the benefits of ECHO hold across encoder scales.**  As shown in Figure 1 b in the main text, the Seq2Seq paradigm improves the overall performance of the lightweight encoder $\text{ECHO}^\mathcal{E}$ by 11.45%. We further added experiments combining ECHO with the encoder-centric CBraMod. As presented in the table below (only key results are shown here; full results are provided in the Appendix), the large pretrained encoder also achieves better performance under the ECHO paradigm than in its original single-task setting even in a much challenging no task prompt setting, confirming that the performance gains arise from the Seq2Seq paradigm itself.
>
> | | | High-Gamma ||| Attention ||
> | :--- | :-: | :-: | :-: | :-: | :-: | :-: |
> | **Method** | ACC-B | ROC AUC | PR AUC | ACC-B | ROC AUC | PR AUC |
> | CBraMod | 0.7478 | 0.8292 | 0.8314 | 0.6478 | 0.7417 | 0.7468 |
> | ECHO-CBraMod | **0.7499** | **0.8481** | **0.8363** | **0.7222** | **0.8100** | **0.8056** |
>
> | | | Mumtaz2016 ||| Stieger2021-LR ||
> | :--- | :-: | :-: | :-: | :-: | :-: | :-: |
> | **Method** | ACC-B | ROC AUC | PR AUC | ACC-B | ROC AUC | PR AUC |
> | CBraMod | 0.8946 | **0.9800** | 0.9765 | **0.8424** | **0.9339** | **0.9297** |
> | ECHO-CBraMod | **0.9035** | 0.9773 | **0.9809** | 0.8370 | 0.9234 | 0.9241 |
>
> ### ***Modification:***
>
>  1. **Added the results of $\text{ECHO}^\mathcal{E}$ to Table 2**.
>  2. **Added analyses in Section 4.2** illustrating the contributions of the Seq2Seq paradigm and ICL training.
>  3. **Added Section. D.2.3 in Appendix** containing the extended experiments integrating ECHO with CBraMod.

---

> ### Author Response · Authors · 2025-11-25
> **Response to Reviewer zEi1 [2/3]**
>
> # Response for Weakness 3:
> ### ***Summary:***
> **A.** Why is SEED treated as zero-shot?
>
> **B.** Whether additional cross-dataset evaluations are provided?
> ### ***Response:***
> **A. It's a subject-level zero-shot.**  Under our evaluation, the target subject’s data never appears during training, making it a  unseen-subject zero-shot setting, which is widely accepted in the EEG community. We have removed “subject-level zero-shot” in the updated manuscript to avoid confusion.
>
> **B. We added new cross-dataset experiments.** As shown in the following table, ECHO retains cross-dataset capability even without task prompts. On SEED-IV (note: this version of ECHO excludes SEED-IV from pretraining), it still achieves performance close to the fine-tuned large-model SOTA.
>
> |||Cho2017|||SEED-IV||
> | -------------------- | :-: | :-: | :-: | :-: | :-: | :-: |
> |**Methods**|ACC-B|ROC AUC|PR AUC|ACC-B|Kappa|F1-Weight|
> | CBraMod (Fine Tune)|0.7033 ± 0.0327|0.8081 ± 0.0429|0.7959 ± 0.230|0.4146 ± 0.0228|0.2088 ± 0.0344|0.3744 ± 0.0454|
> | CodeBrain (Fine Tune) |  0.6531 ± 0.0228  | 0.7429 ± 0.0377 |  0.7254 ± 0.0318  | 0.3641 ± 0.0328 | 0.1685  ± 0.0300 |  0.3341  ± 0.0249  |
> | ECHO (Zero Shot) |  0.5881 ± 0.0026  | 0.6399 ± 0.0026 |  0.6267 ± 0.0032  | 0.3347 ± 0.0019 | 0.1117 ± 0.0028 |  0.3216 ± 0.0031  |
> | ECHO (Few Shot) |  0.6044 ± 0.0032  | 0.6470 ± 0.0053 |  0.6460 ± 0.0019  | 0.3435 ± 0.0024 | 0.1201 ± 0.0017 |  0.3307 ± 0.0014  |
>
>
> ### ***Modification:***
>  1. **Removed the subject-level zero-shot results from Section 4.4,** eliminating potential conceptual ambiguity.
>  2. **Added additional zero-shot experiments in Section 4.4 and Appendix B.3**, offering a more comprehensive evaluation of cross-dataset generalization.
>
> # Response for Weakness 4:
> ### ***Summary:***
> Validating that ECHO’s gains stem from the framework via other encoder–decoder combinations.
> ### ***Response:***
> **Large pretrained EEG encoders still benefit under the decoder-centric framework.** As clarified in our response to Weakness 2, the decoder-centric design of ECHO consistently enhances encoder performance regardless of model scale.
>
> # Response for Question 1:
> ### ***Summary:***
> How does ECHO accommodate variable class cardinality?
> ### ***Response:***
> **The architecture and inference behavior of ECHO are inherently decoupled from the number of classes.**
>
>  - **From an architectural perspective, ECHO's token-based output space is independent of the number of classes.** Traditional classifiers require an output layer whose dimensionality must match the number of classes, whereas ECHO’s decoder generates a single label token from a shared vocabulary. Consequently, changes in class count do not alter the model architecture or output size, allowing the same framework to flexibly support arbitrarily many labels.
>  - **From a behavioral perspective, ECHO performs a more challenging form of classification by framing prediction as hierarchical semantic reasoning (Section 3.2 *Intuition* (a) and (b) in the main paper).** ECHO is exposed to diverse tasks and labels during training and learns to select the most compatible label token from all known labels, making it inherently robust to variable class cardinalities at test time. For example, when predicting left-hand in MI data, ECHO **first** infers the task family (MI vs. EMO vs. Sleep···), **then** the relevant MI subtype (left/right-hand vs. foot/tongue), and **finally** selects the correct label token (Left).
>
> ### ***Modification:***
>
>  1. **Added a description in Section 3.2 *Intuition*,** clarifying ECHO’s adaptability to variable class cardinalities.

---

> ### Author Response · Authors · 2025-11-25
> **Response to Reviewer zEi1 [3/3]**
>
> # Response for Question 2:
> ### ***Summary:***
> A discrepancy between Eq. (11) and Fig. 2 is noted, raising concerns about the exact output form of ECHO.
> ### ***Response:***
> **Figure 2 depicts the Seq2Seq teacher-forcing process during training, which is fully consistent with Eq. (11), both involving a total of 2S+3 textual tokens.** Each prediction step in ECHO corresponds to \(S+1\) EEG embedding tokens (the S support samples plus 1 target sample). The associated textual sequence thus contains 2S+2 task and label tokens, and adding the SOT token (input) or EOT token (output) yields exactly 2S+3 tokens, matching Eq. (11):
>
>     Input:  <|SOT|><|EMO|><|Positive|><|EMO|><|Negative|>···<|EMO|><|Negative|>
>     Output:        <|EMO|><|Positive|><|EMO|><|Negative|>···<|EMO|><|Negative|><|EOT|>
>
> During inference, however, the 2S support tokens are provided as conditions, so the decoder only predicts the remaining three tokens: the target EEG’s task, label, and EOT token:
>
>     Input:  <|SOT|><|EMO|><|Positive|><|EMO|><|Negative|>···
>     Output:                                                 <|EMO|><|Negative|><|EOT|>
>
> ### ***Modification:***
>
>  1. **Revised Figure 2** to align its depiction with Equation (11).
>  2. **Added a clarification in Section 3.2 Sequence Format**, explicitly describing the output format of ECHO.
>
> # Response for Question 3:
> ### ***Summary:***
> **A.** Why does removing positional encoding break prediction?
>
> **B.** What does “no valid predictions” mean?
>
> **C.** Why is the result not “random guessing”?
> ### ***Response:***
> **A. Positional information is essential for maintaining the structure of ECHO’s autoregressive generation.** Without positional cues, the decoder cannot distinguish the structural roles of different tokens or determine its current generation step. ECHO will treat all tokens as unordered equivalents and cannot follow a valid decoding trajectory, resulting in the inability to produce any meaningful label sequence.
>
> **B. Indicates a collapse of semantic coherence rather than a mere misclassification.** For example:
>
>     Correct Prediction:  MI/Left
>     No valid Prediction: Right]= MI/LeftLeLeLeLeLe
>
>  **C. ECHO performs sequence generation rather than K-way classification.**  Random guessing only applies to softmax classifiers. Once positional structure collapses, ECHO cannot form any interpretable token sequence at all.

---

### Official Review · Reviewer_Vc4L · 2025-11-11

**Soundness:** 3
**Presentation:** 3
**Contribution:** 3
**Rating:** 8
**Confidence:** 3

**Summary:**

The paper proposes ECHO, a decoder-centric sequence-to-sequence (seq2seq) paradigm for EEG modeling that concatenates support samples (with known task and label) and a target sample into one sequence, forcing the model to generate <task> then <y> (then ). The goal is to overcome the “decoder bottleneck” in Large EEG Models (LEMs) and enable in-context learning (ICL) across heterogeneous datasets. The system includes a channel-template alignment step (same/nearby electrodes averaged; missing padded with zero) and hybrid positional/role encodings. Training uses a curriculum (first fixed 8 support examples, then randomizing the count in [0,12]); testing uses k∈{0,8,12} with no task tokens fed.

**Strengths:**

1.Paradigm-level breakthrough: a decoder-centric seq2seq with a fixed solution order
ECHO hard-codes the reasoning path—support → target → generate <task> → generate <y> → —and trains via next-token prediction. This lets the model learn within a single sequence space how to reason from examples to the target, instead of relying on a small classifier head or fragile text-bridging.


2.Hybrid positional/role encodings that fuse continuous EEG with discrete control tokens
The method explicitly distinguishes functional roles (e.g., support vs. target) and balances heterogeneous inputs by combining token-level, sample-level, and text-level encodings. As a result, the decoder preserves fine-grained temporal EEG structure and the logic of <task>/<y> prediction.


3.Strong evidence of generalization and “zero-context” robustness
Even without any support samples, the sequential formulation internalizes the mapping between signal, task, and label. On unseen datasets (e.g., BCIC 2020-T1) it maintains consistent generalization, and under a strict train-once, evaluate-everywhere protocol it can infer both the task and the label without being given <task> at test time.

**Weaknesses:**

ICL evaluation under-characterized
Training uses a two-stage curriculum (8 → random [0,12]), while testing uses k∈{0,8,12}. However, tables report only “No Support” vs an ambiguous “Support” (k=8? k=12? average/best/mix?). There is no k→performance curve (e.g., 0,2,4,8,12 with CIs) and the support sampling protocol (same/cross subject or device, adjacency, fixed indices) is not fully specified. This obscures how performance scales with context size and complicates attribution to the paradigm rather than context amount.


Ablations do not test core claims
Current ablations toggle only parts of the hybrid positional encodings.Missing:
Seq2seq necessity & order sensitivity: shuffle support order; swap output order (<y> before <task>); remove support-recap generation; compare with a non-sequential, information-equivalent baseline that ingests the same support/target encodings and predicts via a feed-forward head.
Decoder capacity: vary layers/hidden/heads/FFN and compare to an equal-capacity hierarchical discriminative baseline (task-head → label-head) to substantiate the “decoder bottleneck” thesis.
Channel alignment: isolate its contribution via ablation.


Fairness: multi-task ECHO vs single-task baselines
ECHO trains jointly across tasks and is evaluated everywhere; most baselines are trained/fine-tuned per task. A multi-task baseline (e.g., shared encoder with task-specific heads) or single-task ECHO is needed to separate seq2seq benefits from multi-task training effects.


Channel alignment methodology
Averaging mapped electrodes may smooth meaningful spatial nuances and inject template bias. There is no comparison to stronger alternatives (e.g., spherical/spline interpolation, learnable channel mappings, or permutation-robust architectures), nor pre/post alignment PSD/topography analyses.


 Important extensions buried in the appendix
The long-sequence variant (ECHOL) and the MI-specialized variant (ECHOMI) appear only in the appendix. They should be summarized in the main text (motivation, setup, key findings) to illustrate scope and limitations; ensure variant names match the paper (avoid superscripts like ECHO^L / ECHO^MI).

**Questions:**

1.Technical Writing and Presentation
Typos/formatting should be fixed:
(1)“proprocess” → preprocess (p.5)
(2)Missing ) (Page 5)
(3)Sentence-initial “if” → If (p.5, under Eq.(5))
(4)“Postional" → Positional (p.6)
(5)“encodding" → "encoding" (Page 6, twice)
(6)Extra ) near Table 12 (p.24)

---

> ### Author Response · Authors · 2025-11-25
> **Response to Reviewer Vc4L [1/3]**
>
> # Overview
> We sincerely thank reviewer Vc4L for the thoughtful evaluation and valuable comments. To keep the rebuttal clear and easy to follow, especially given the time since the original submission, we present all replies in a consistent three-part format:
>
> ***Summary:*** A concise restatement of the reviewer’s core concerns (partitioned as **A**, **B**, etc.).
>
> ***Response:*** A detailed clarification addressing each point.
>
> ***Modification:*** A description of the revisions made to the manuscript.
>
> # Response for Weakness 1:
>
> ### ***Summary:***
> **A.** How many support samples are used during testing?
>
> **B.** No analysis regarding the effect of the number of support samples.
>
> ### ***Response:***
>
> **A. All testing use 8 support samples.**
>
> **B. We have added experiments analyzing the impact of support-sample size.**  As shown in the table below, ECHO generally achieves the largest gains with 8 support examples, while a subset of tasks benefits from 12 supports. Since the downstream datasets differ in the number of classification categories, we report only BACC for comparability. All support-size analyses are evaluated using results from seed 0.
>
> | **Dataset** | 0 | 2 | 4 | 6 | 8 | 10 | 12 |
> |:-|:-:|:-:|:-:|:-:|:-:|:-:|:-:|
> | SEED-IV  | 0.3399  |0.3476 | 0.3253 |  0.3439|  **0.4016**|  0.3604 | 0.3544 |
> | MentalArithmetic | 0.5333 | 0.6566 | 0.5483 | 0.6506 | **0.7116** | 0.6060 | 0.6475 |
> | SEED-V | 0.2439 | 0.2440 | 0.2072 | 0.2334 | **0.2667** | 0.2495 | 0.2472 |
> | High-Gamma | 0.8493 | 0.8507 | 0.8544 | 0.8509 | 0.8585 | 0.8573 | **0.8718** |
> | SEED | 0.7593 | 0.7930 | 0.7823 | 0.8216 | 0.8181 | 0.7946 | **0.8246** |
> | Stieger2021-UD | 0.7210 | 0.7136 | 0.7049 | 0.7123 | 0.7233 | 0.7146 | **0.7241** |
>
> The support-scaling analysis shows two clear patterns. First, across most datasets, performance peaks at k = 8, after which larger support sets introduce additional variability and noise, leading to reduced accuracy, which supports that model does not always benefit from more supports [1]. Second, a group of cleaner, more separable datasets (higher BACC) shows steady improvement as k increases. These results confirm that support size is task-dependent rather than “the more the better,” and moderate support (around 8) is typically optimal for heterogeneous EEG tasks.
>
> ### ***Modification:***
>  1. **Corrected the description in Section 4.1 Task Setting**, clarifying the number of support samples used during testing.
>  2. **Added Appendix D.2.2** presenting a support-size scaling analysis.

---

> ### Author Response · Authors · 2025-11-25
> **Response to Reviewer Vc4L [2/3]**
>
> # Response for Weakness 2:
> ### ***Summary:***
> **A.** Analysis support-sample order.
>
> **B.** Analysis output-token order.
>
> **C.** Analysis removing the support-recap mechanism.
>
> **D.** Compare with a non-sequential but information-equivalent baseline.
>
> **E.** Analysis decoder sizes.
>
> **F.** Ablate channel alignment to analyze its contribution.
>
> ### ***Response:***
> **A. The support-sample order is inherently randomized.** Support samples are drawn randomly from the ICL pool.
>
> **B. Switching the order of the Task token and the Label token does not affect ECHO’s performance.**
> * **The limited number of tasks makes the support set sufficient to uniquely determine the task.** Since the benchmark contains only a small set of tasks, the input–output patterns in the support examples already allow the model to infer the task type almost unambiguously. As a result, the order between the Task token and the Label token does not introduce additional informational differences.
> * **The structural gap between Task and Label is too small to create order dependence.** These EEG datasets contain only a few label categories, and their label structures are similar across tasks. In such a small label space, predicting the task first or the label first imposes nearly the same difficulty. Consequently, swapping the order of Task and Label tokens leads to effectively equivalent training behavior.
>
>
> **C. ECHO does not include a support-recap mechanism.** At test time, the decoder directly generates only the target task and label tokens, without any support-recap generation.
>
> **D. A non-sequential model cannot receive information equivalent to the sequential ECHO input.**
>  - **They cannot treat support and target EEG as discrete exemplars.** They will process all discrete segments as one temporally continuous signal.
>  - **They have no mechanism to ingest the corresponding support task and label tokens.** These discrete control symbols exist only in a sequence model’s token space and cannot be encoded by an encoder-only discriminative model.
>
> **E. We have added experiments comparing decoder sizes.** As shown in the following table (partial results; full results provided in the main text), increasing decoder capacity yields negligible performance gains for ECHO:
> |||High-Gamma|||Stieger2021-LR||
> | -------------------- | :-: | :-: | :-: | :-: | :-: | :-: |
> | **Methods** | ACC-B | ROC AUC | PR AUC | ACC-B | ROC AUC | PR AUC |
> | $\text{ECHO}^\mathcal{E}$| 0.8039 | 0.8900 | 0.8889 | 0.6123 | 0.8918 | 0.9048 |
> | ECHO (No Support) | 0.8438 | 0.9125 | 0.9047 | 0.8534 | 0.9349 | 0.9363 |
> | ECHO | 0.8552 | 0.9208 | 0.9125 | **0.8534** | **0.9349** | **0.9363** |
> | ECHO-Large (No Support) | 0.8368 | 0.9042 | 0.8956 | 0.8494 | 0.9283 | 0.9293 |
> | ECHO-Large | **0.8667** | **0.9292** | **0.9220** | 0.8499 | 0.9322 | 0.9336 |
>
> The encoder is relatively small, and the overall bottleneck lies in the EEG representation rather than decoder expressiveness. Therefore, enlarging the decoder does not improve performance.
>
> **F.  We added an evaluation using alternative channel fusion methods in $\text{ECHO}^\mathcal{E}$.** As shown in the following table, the averaging strategy has only a minimal impact on performance.
>
> ||| KoreaU ||| High-Gamma ||
> | -------------------- | :-: | :-: | :-: | :-: | :-: | :-: |
> | **Methods** | ACC | ROC AUC | PR AUC | ACC | ROC AUC | PR AUC |
> | Full Channel | **0.7217** | **0.8138** | **0.8079** | 0.6780 | 0.7956 | **0.8209** |
> | Average (Ours) | 0.7031 | 0.7806 | 0.7790 | **0.6911** | **0.8418** | 0.8049 |
> | Channel Deletion | 0.6983 | 0.7808 | 0.7759 | 0.6476 | 0.7710 | 0.8134 |
>
>
> ### ***Modification:***
>
>  1. **Added Appendix D.3.2** to compare different decoder size.
>  2. **Added Appendix D.2.1** to analyze channel-fusion strategies.

---

> ### Author Response · Authors · 2025-11-25
> **Response to Reviewer Vc4L [3/3]**
>
> # Response for Weakness 3:
> ### ***Summary:***
> Compare with multi-task learning baseline.
> ### ***Response:***
> **We added a comparison between CBraMod under multi-task learning and under the ECHO ICL paradigm.** As shown in the following table (partial results; full results provided in the main text), the multi-task baseline follows the conventional setup of a shared encoder with task-specific heads, whereas CBraMod-ECHO replaces the small encoder in ECHO with CBraMod. Under this setting, even though CBraMod-ECHO operates in a more challenging multi-task scenario without task prompts and must infer the task type on its own, it still consistently outperforms the standard multi-task CBraMod with shared encoders， demonstrating the clear advantage of the Seq2Seq-ICL framework in cross-task reasoning.
>
> |||Attention |||High-Gamma||
> | -------------------- | :-: | :-: | :-: | :-: | :-: | :-: |
> | **Methods** | ACC-B | ROC AUC | PR AUC | ACC-B | ROC AUC | PR AUC |
> | CBraMod | 0.6478 | 0.7417 | 0.7468 | 0.7513 | 0.8277 | 0.8335 |
> | CBraMod-MH | 0.5000 | 0.4593 | 0.4654 | **0.7519** | 0.8426 | **0.8375** |
> | CBraMod-ECHO | **0.7222** | **0.8100** | **0.8056** | 0.7499 | **0.8481** | 0.8363 |
>
> |||Mumtaz2016 |||Stieger2021 LR||
> | -------------------- | :-: | :-: | :-: | :-: | :-: | :-: |
> | **Methods** | ACC-B | ROC AUC | PR AUC | ACC-B | ROC AUC | PR AUC |
> | CBraMod | 0.8946 | 0.9800 | 0.9765 | **0.8424** | **0.9339** | **0.9297** |
> | CBraMod-MH | 0.8542 | 0.9622 | 0.9507 | 0.8374 | 0.9331 | 0.9286 |
> | CBraMod-ECHO | **0.9035**  | **0.9773** | **0.9809** | 0.8370  | 0.9234 | 0.9241 |
>
> ### ***Modification:***
>  1. **Added Appendix D.2.4** comparing standard multi-task learning with ECHO-style learning.
>  2. **Added Appendix D.2.3** containing the extended experiments integrating ECHO with CBraMod.
>
>
> # Response for Weakness 4:
> ### ***Summary:***
> Why use an averaging strategy that may introduce phase misalignment when there exist many stronger alternatives?
> ### ***Response:***
> **This choice is an intentional conservative design.**
>  - **We intentionally avoid introducing more complex channel fusion strategies so as not to obscure the contribution of the Seq2Seq ICL paradigm itself.** The focus of ECHO is its Seq2Seq formulation, not identifying the optimal channel fusion technique. Therefore, we deliberately adopt the simplest method to validate ECHO under the weakest standardization assumptions.
>  - **Our goal is only to demonstrate that a prior-guided channel fusion strategy is feasible.** Within ECHO’s prior-guided preprocessing framework, as the reviewer noted, many stronger channel-fusion strategies exist (for example, the various edge-learning methods in EEG-related graph neural network research). However, this paper only aims to verify the feasibility of the framework and does not delve into optimizing the performance of channel-fusion methods.
>  - **As shown in our response to Weakness 2 F, the result  shows that the impact of the chosen channel fusion method is minimal.**
>
> # Response for Weakness 5:
> ### ***Summary:***
> The main text lacks summaries of the two ECHO variants.
> ### ***Response:***
> **We have added concise descriptions of two ECHO variants in the main text**, including their motivation, setup, and key findings.
> ### ***Modification:***
> -   **Added Section 4.5** summarizing ECHO-Long and ECHO-MI.
> -   **Added Section 6** discussing the limitation of ECHO.
>
> # Response for Question 1:
> ### ***Summary:***
> The paper contains several minor errors.
> ### ***Response:***
> **We thank the reviewer for pointing these out. All identified mistakes have been corrected.**
>
> # Reference
> [1] More is not always better? enhancing many-shot in-context learning with differentiated and reweighting objectives, ACL 2025

---

### Author Response · Authors · 2025-12-04
**Discussion Summary**

Dear AC, SAC, and PC,

We sincerely thank you for the time and effort you have dedicated to the review process, and we deeply appreciate your commitment to maintaining academic fairness under challenging circumstances. The following serves as a summary for the rebuttal phase, outlining both the reviewers’ attitudes and key comments, as well as our corresponding responses, with the aim of facilitating your quick understanding of the essential points.

## **Summary:**

Although the reviewers’ scores vary, we are encouraged that:
* **All five reviewers recognized the novelty of ECHO’s decoder-centric perspective and acknowledged that this paradigm shift offers a meaningful extension to EEG foundation modeling**, particularly in enabling in-context learning and multi-task generalization.

Meanwhile, reviewers offered constructive suggestions that helped us further validate and analyze ECHO from multiple angles.
* **We summarize all major points as follows** *(parentheses denote our responses)*:
    1.  Add **additional ECHO variants**, including larger encoders/decoders. *(Appendix D.2.3)*
    2.  Validate the **channel fusion strategy**. *(Appendix D.2.3)*
    3.  Analyze the effect of **support sample size**. *(Appendix D.2.1, E)*
    4.  Compare against **standard multi-task learning** methods. *(Appendix D.2.4)*
    5.  Include **additional zero-shot** evaluations. *(Section 4.4, Appendix B.3)*
    6.  Provide computational **cost analysis**. *(Appendix D.3.1)*
    7.  Discuss the **limitations** of ECHO more thoroughly **in the main text**. *(Section 4.2)*

## **Detail of all Rebuttal Discussion:**

We summarize the full discussion as follows. For each reviewer, we organize the interaction into three components:

**Initial:** assessment, concerns, and questions before rebuttal phase.

**Rebuttal:** additional assessment, concerns, and questions during the discussion.

**Final:** final stance before 26th Nov.

---

### **Reviewer Vc4L:**
**Initial:**
* *Assessment:* Paradigm-level breakthrough **(R=8,C=3)**
* *Concerns:* 1, 2, 3, 4, 7

**Final:**
* *Assessment:* N/A

---

### **Reviewer zEi1:**
**Initial:**
* *Assessment:* Mitigates limitations of prior approaches and enables ICL **(R=4,C=4)**
* *Concerns:* 1, 5
* *Question:* ECHO's decoder classification mechanism

**Final:**
* *Assessment:* N/A

---


### **Reviewer r5rr:**
**Initial:**
* *Assessment:* Novel paradigm shift, Effective ICL (R=4, C=5)
* *Concerns:* 1, 2, 3, 5, 6, Downstream task expansion
* *Questions:* Hyperparameter details, Loss formulation, Necessity of PE design, Missing encoder results

**Rebuttal:**
* *Assessment:* Despite limitations *(clarified in our further response)*, still an interesting attempt, particularly in unifying diverse downstream tasks in a flexible manner **(R=6, C=5)**
* *Concerns:* Necessity of the multi-task paradigm, Comparison with task-specific models
* *Questions:* Source of channel fusion, Performance drop in ECHO variants, Implicit temporal-context modeling, Differences from few-shot learning

**Final:**
* *Assessment:* N/A


---

### **Reviewer F1QH:**
**Initial:**
* *Assessment:* First to show ICL works, Strong multi-task performance **(R=6, C=3)**
* *Concerns:* 1, 3, 4, 6
* *Questions:* Design of in-context training rounds

**Final:**
* *Assessment:* N/A

---

### **Reviewer KNjL:**
**Initial:**
* *Assessment:* ICL capability; transferable (R=4, C=4)
* *Concerns:* 2, 3, 7
* *Questions:* Dataset reporting choices, Teacher forcing, Uniqueness of ICL capability

**Rebuttal:**
* *Assessment:* ECHO, with its ICL capability, is a valuable addition to EEG foundational modeling; Three non-SOTA results do not diminish ECHO's novelty and contribution (R=4, C=4)
* *Concerns:* 7

**Final:**
* *Assessment:* Addressed all concerns **(R=8, C=4)**


---


We thank you again for your careful evaluation and for overseeing the discussion process. We greatly appreciate your time and consideration.

Sincerely,
The Authors of ECHO

---

### Meta-Review · Area_Chair_vHaS · 2025-12-27

**Summary:**

The work challenges the standard encoder-only approach of SoTA SSL large brain models. The idea to output a task token in a decoding task followed by a label is innovative and leads to convincing results despite being a first exploration of such an approach for EEG decoding. Reviewers are overall enthusiastic about the work and the rebuttal period led to very constructive discussions.

**Reviewer Concerns:**

Reviewers were initially rather supportive and significant discussions further helped to convince at least 2 reviewers during the rebuttal. Some concerns still remain regarding further ablations or more evaluation that could be conducted yet the novelty and the contribution is judged significant enough to impact the field positively.

**Reviewer Scores:**

Reviewer Vc4L: 8
Reviewer zEi1: 4
Reviewer r5rr: 4 -> 5
Reviewer F1QH: 6
Reviewer KNjL: 4 -> 5 or 6.

---

### Decision · Program_Chairs · 2026-01-26

Accept (Poster)